# Genetic and environmental interactions contribute to immune variation in rewilded mice

Oyebola Oyesola [1,11] ✉, Alexander E. Downie [2,3,11], Nina Howard [1], Ramya S. Barre [2,4], Kasalina Kiwanuka[1], Kimberly Zaldana[1,5], Ying-Han Chen [6,7,10], Arthur Menezes [2], Soo Ching Lee [1], Joseph Devlin[6], Octavio Mondragón-Palomino[1], Camila Oliveira Silva Souza [1], Christin Herrmann[6,7], Sergei B. Koralov[5], Ken Cadwell [7,8], Andrea L. Graham [2,9] ✉ & P'ng Loke [1,8] ✉

The relative and synergistic contributions of genetics and environment to interindividual immune response variation remain unclear, despite implications in evolutionary biology and medicine. Here we quantify interactive effects of genotype and environment on immune traits by investigating C57BL/6, 129S1 and PWK/PhJ inbred mice, rewilded in an outdoor enclosure and infected with the parasite *Trichuris muris*. Whereas cellular composition was shaped by interactions between genotype and environment, cytokine response heterogeneity including IFNγ concentrations was primarily driven by genotype with consequence on worm burden. In addition, we show that other traits, such as expression of CD44, were explained mostly by genetics on T cells, whereas expression of CD44 on B cells was explained more by environment across all strains. Notably, genetic differences under laboratory conditions were decreased following rewilding. These results indicate that nonheritable influences interact with genetic factors to shape immune variation and parasite burden.

An individual's immune phenotype is shaped by some combination of genetic, maternal and epigenetic factors, and nonheritable influences such as environmental exposure (including infection history and the microbiome)[1–8]. However, the relative and potentially interacting contributions of heritable and nonheritable factors to interindividual immune variation remain controversial despite the importance of such variation for both medicine and evolutionary biology. For example, variation in immune responses can determine whether an individual will experience severe or asymptomatic infection[9,10], and whether severity arises due to failure to control pathogens or excessive collateral tissue damage following defective immune regulation[3].

Recent studies on the human immune system have sought to identify the relative contributions of genetic and environmental factors to variation in immune phenotypes among healthy individuals[1,6,7], as well as during infection[7] or inflammatory conditions[11]. Such studies draw upon analysis of immunological divergence between identical twins[7] or genetic heritability estimation for immune traits through functional genomics[1]. However, the design of these studies often makes quantifying the interactive effects of genetics and environment challenging and, generally, interacting effects have not been well examined in most immunological studies. For example, variation not attributable to genetics is generally attributed to environment alone rather than the possibility of genotype-by-environment interactions (Gen × Env), which are inferred if effects of environment are differentially amplified in different genotypes, or vice versa[12,13]. Important context dependency in immune function is thus often missing from these calculations.

For instance, what if the impact of environment upon memory T cell frequencies depends upon host genotype, or if, put differently, the impact of genotype upon memory T cell frequencies depends upon environment? Evolutionary biology is explicitly interested in such context dependencies because they provide the raw materials for adaptive evolution and diversification; Gen × Env interactions are common and substantial in effect for a variety of traits[14,15] and disease outcomes[12,13,16].

Controlled experiments with mice could help decipher effects of interactions between genetic and environmental effects on the immune system, but most studies in mice instead aim to reduce environmental variation to discover genetic factors regulating cellular and molecular components of immunity[3,17–19]. Most times, this approach ignores interactions and provides only partial insight into direct genetic effects by not investigating the extent of the measured genetic effect that is mediated by the environment. We have taken a decidedly different approach of using an outdoor enclosure system to introduce female laboratory mice of different genotypes—C57BL/6, 129S1 and PWK/PhJ—into a natural environment, in a process termed 'rewilding'[20]. We selected these genetically diverse nonalbino founder strains of the Collaborative Cross of mice[21] to enable more complex trait analysis in the future. C57BL/6J and 129S1/SvImJ are representative of classical laboratory inbred strains, whereas PWK/PhJ is a representative of a wild-derived strain[21]. We rewilded only female mice to prevent unintended breeding from male mice breaching the barriers in the rewilding environment. We have tracked behavior outdoors (revealing that social behavior was a key predictor of shared memory T cell and complete blood count (CBC) leukocyte differential profiles[22]), challenged the mice with *Trichuris muris* embryonated eggs 2 weeks after release, recovered the mice for analysis and then investigated genetic and environmental contributions to immune phenotypes[19,20,23]. Previously, using mice with mutant alleles in inflammatory bowel disease susceptibility genes (*Nod2* and *Atg16l1*), we found that the genetic mutations affected the production of cytokines in response to microbial stimulation, whereas immune cell composition was more influenced by environment[19,23]. We also found that rewilded C57BL/6 mice become more susceptible to infection with the intestinal nematode parasite *T. muris*[20]. However, those experiments explored limited genetic variation and did not examine whether interactions between genetics and environment would influence immune phenotype and helminth susceptibility[4,24].

Here, we quantify relative and interactive contributions of genetic and environmental influences on heterogeneity in immune profiles and helminth susceptibility. Our results demonstrate that interactions between genetics and environment are an important source of variation for specific immune traits, but there are also tissue-dependent differential effects of environment versus genetics on specific cellular compartments such as T cells and B cells. The effect of an extreme environmental shift on immune phenotype is modulated by genetics, and, in turn, the genetic differences among strains are modulated by the environment. Such interactions are an important source of interindividual immune variation and likely important in determining susceptibility to parasitic infections in humans and other natural mammalian populations.

## Results
### Experimental design
To quantify sources of heterogeneity in immune profiles and helminth susceptibility, we compared C57BL/6, 129S1 and PWK/PhJ mice housed in two different environments—a conventional vivarium but kept in summerlike temperatures and photoperiods (hereafter, 'Lab' controls) versus those that were outdoors (hereafter, 'Rewilded') (Fig. 1a). These strains differ by up to 50 million single nucleotide polymorphisms and short insertions/deletions (indels)[25] (the human population is estimated to contain approximately 90 million single nucleotide polymorphisms and indels[26]). Mice were randomly assigned into different groups for each experimental block. We rewilded mice (n = 72) or kept

them in laboratory housing (n = 63) for 2 weeks and then either infected them with approximately 200 eggs of the intestinal helminth *T. muris* (n = 61) or left them uninfected (n = 74), returning them to the outdoor or vivarium environment for a further 3 weeks. We conducted two replicate experiments across different periods during the summer months (Block 1, n = 61, ending in July; Block 2, n = 74, ending in August).

### Gen × Env interactions drive peripheral blood mononuclear cell immune variation
The immune cell composition of peripheral blood mononuclear cells (PBMCs) was analyzed by spectral cytometry with a lymphocyte panel (Supplementary Table 1). To quantify the relative contributions of genotype (that is, strain), environment (that is, Lab versus Rewilded), infection (that is, exposure to *T. muris*) and their interactions to the high-dimensional spectral cytometry data from the PBMC analysis, we used multivariate distance matrix regression (MDMR) analysis, a statistical approach used to identify factors contributing to variation in high-dimensional data[27,28]. The MDMR model we used incorporated genotype, environment and infection as fixed effects and the two independent experiments in July or August (denoted as 'Block') as a random effect to calculate the interactive and independent contributions of these factors to the outcomes.

The cellular composition data for the PBMCs of each individual mouse are determined by unsupervised *k*-means clustering to group cells into clusters based on similarities of cellular parameters (Extended Data Fig. 1). We calculate the composition of cells for each individual sample based on cluster membership established by *k*-means clustering, and these unbiased cluster composition data (Extended Data Fig. 1 and Supplementary Data 3) are then used as the outcome variable for the MDMR analysis.

MDMR analysis on PBMC cellular composition (Supplementary Data 3) showed that genotype and environment had a notable effect on variation in cellular composition, not only as independent variables (Fig. 1b, top) but also through interactions between genotype and the environment (Gen × Env) (Fig. 1b, bottom). These patterns can be visualized through a principal component analysis (PCA) on cellular composition data of individual mice (Fig. 1c,d and Extended Data Fig. 1b,c). The PCA plot indicated strong effects of environment on variation along the principal component 1 (PC1) axis (Fig. 1c,d) and of genetics on variation along the principal component 2 (PC2) axis (Fig. 1c,d), while infection displayed a minimal effect (Fig. 1b and Extended Data Fig. 1d,e). Variation along the PC1 axis for Rewilded mice is substantially greater than for Lab mice (Fig. 1c,d). The PCA plot also suggested that variance on the PC2 axis between mouse strains was greater in Lab mice than for Rewilded mice (Fig. 1c,d).

The loading factor in the PCA showed that Cluster C9, a TCRb⁻B220⁻Ki-67^hiCD44^hi population, might be driving the environment-related variation on PC1 axis (Extended Data Fig. 1c). While this population expands following rewilding regardless of the strain of mice, our limited markers prevent further characterization of this population. Interestingly, the loading factors in the PCA also showed that variation in expression of CD44 on CD4⁺ T cells is important for driving the genetic variation on the PC2 axis (Extended Data Fig. 1c). Although there is substantial difference in expression of CD44 on CD4⁺ T cells between the inbred strains for lab-housed mice, these differences were no longer present between the C57BL/6 mice and the PWK/PhJ mice following rewilding (Fig. 1e). Hence, the genetic differences seen in the clean laboratory environment can be reduced following rewilding. In contrast, rewilded C57BL/6 mice had more CD4⁺Tbet⁺ cells after infection compared with the PWK/PhJ and the 129S1 mice, while in laboratory uninfected conditions, there is no difference between these two strains of mice (Fig. 1f and Extended Data Fig. 2). These results indicate that a stronger T helper 1 cell ($T_H$1) response to *T. muris* in the C57BL/6 mice is observed in the rewilding condition compared with the other strains of mice. Hence, genetic

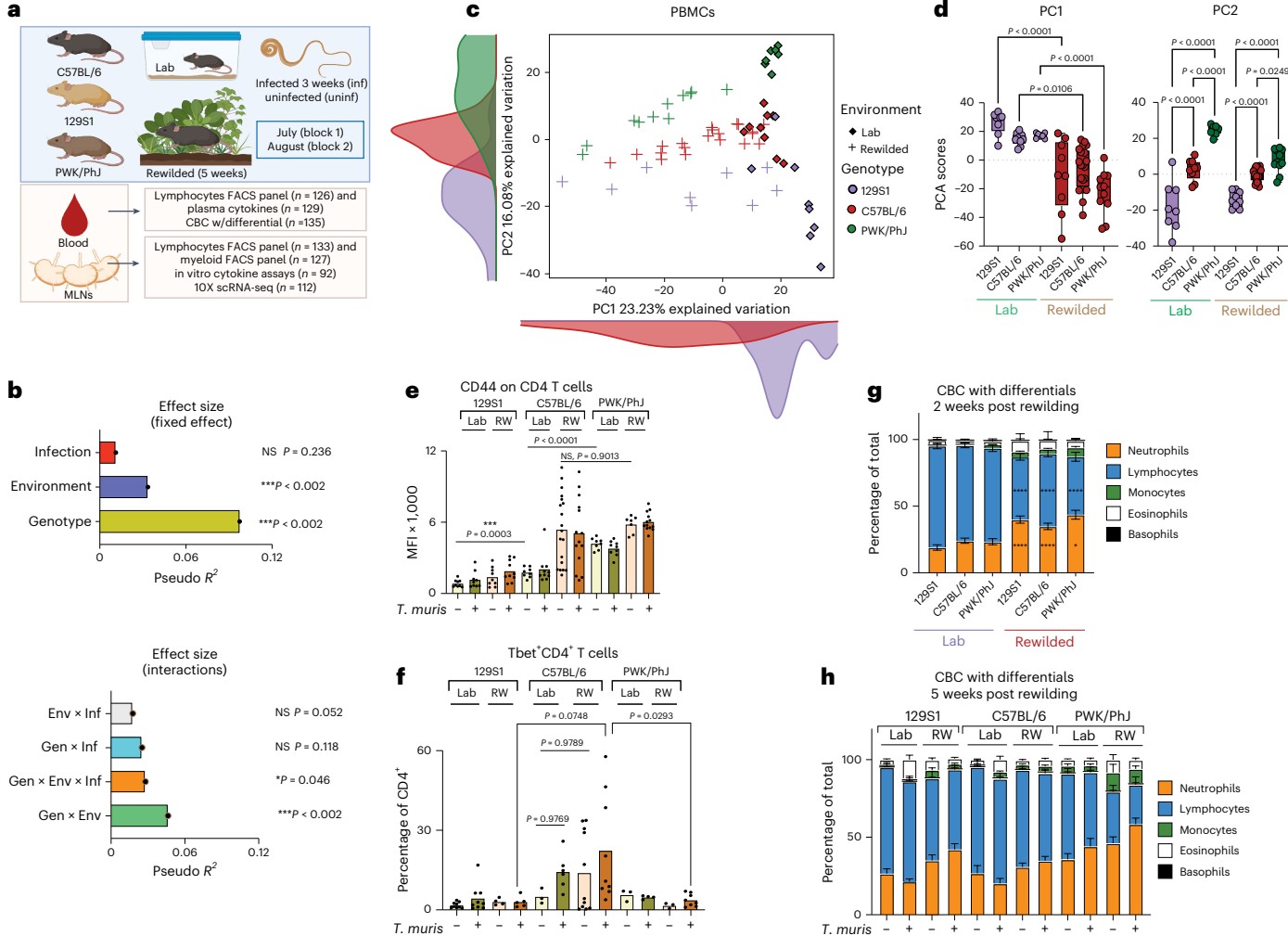

**Fig. 1 | Gen × Env interactions drive PBMC immune variation. a**, Experimental design. **b**, Bar plots showing the pseudo $R^2$ measure of effect size of predictor variables and interactions as calculated by MDMR ($n = 64$; 17 129S1, 29 C57BL/6 and 18 PWK/PhJ mice). **c**, PCA of immune cell clusters identified by unsupervised clustering ($n = 64$; 17 129S1, 29 C57BL/6 and 18 PWK/PhJ mice) in the blood. **d**, Box plot showing variance on PC1 and PC2 axes of PCA plots in **c**. The box plot center line represents median, the boundaries represent IQR, with the whiskers representing the upper and lower quartiles ±1.5 × interquartile range (IQR); all individual data points are shown (129S1 Lab = 8, C57BL/6 = 9, PWK/PhJ Lab = 7, 129S1 RW = 9, C57BL/6 RW = 20, PWK RW = 11). **e,f**, Bar plots showing GMFI of CD44 on blood CD4⁺ T cells (**e**), and percentage of Tbet⁺ CD4 T cells of Live, CD4⁺ T cells (**f**) (Block 2 only, $n = 64$). **g,h**, Bar plots showing percentages of neutrophils, lymphocytes, monocytes, eosinophils and basophils out of total

at 2 weeks post rewilding ($n = 139$, 40 129S1, 52 C57BL/6, 47 PWK/PhJ over two experimental blocks), *$P < 0.05$, ****$P < 0.0001$ (see details in the Source Data) (**g**), and 5 weeks post rewilding based on assessment by CBC with differentials ($n = 135$, 41 129S1, 51 C57BL/6, 43 PWK/PhJ over two experimental blocks) (**h**) (full raw dataset can be found in Supplementary Data 4). Statistical significance was determined based on MDMR analysis with R package (**b**) or based on one-way ANOVA one-tailed test between different groups with GraphPad software (**d**–**f**). For **e** and **f**, direct comparison was done between groups of interest with one-way ANOVA test. For **g**, two-way ANOVA with Tukey's multiple comparison was done to calculate column effect. Data are displayed as mean ± s.e.m. and for **d**, **e** and **f** bar plots dots represent individual mice. Not significant (NS) $P > 0.05$; *$P < 0.05$; **$P < 0.01$; ***$P < 0.001$; ****$P < 0.0001$. MFI, mean fluorescence intensity; RW, rewilded.

differences in response to infection can sometimes emerge only in rewilding conditions. These results illustrate how Gen × Env and genotype, environment and infection (Gen × Env × Inf) interactions affect specific immune traits.

CBC with differential (CBC/DIFF) is a standard clinical test used to assess inflammatory responses in patients and is suitable for longitudinal analyses to compare the acute effects of environmental change (at 2 weeks post rewilding) with when the immune system has acclimatized to the new environment (at 5 weeks post rewilding (Supplementary Data 4)). At 2 weeks post rewilding, there is a significant effect of rewilding on circulating neutrophils, lymphocytes and eosinophils across all genotypes (Fig. 1g). At 5 weeks post rewilding, we observed a trend towards more neutrophils in the rewilded PWK/PhJ mice (Fig. 1h), indicating a genotype effect on neutrophil abundance in the rewilding

environment. Eosinophils are more readily induced by *T. muris* infection in the Lab mice (Fig. 1h) than in the Rewilded mice on the 129S1 and C57BL/6 backgrounds. Together, CBC/DIFF data indicated that acute environmental change at 2 weeks had a bigger effect on total blood cell composition than at 5 weeks. Additionally, infection-induced responses in the laboratory setting can be altered during rewilding in specific genotypes. Hence, there are context-dependent effects of genotype, environment and infection on immune traits in the peripheral blood, depending on the timing of the environmental change.

### Gen × Env × Inf interactions drive mesenteric lymph node variation

In contrast to human studies, we can assess immune responses in secondary lymphoid organs focusing on the mesenteric lymph nodes

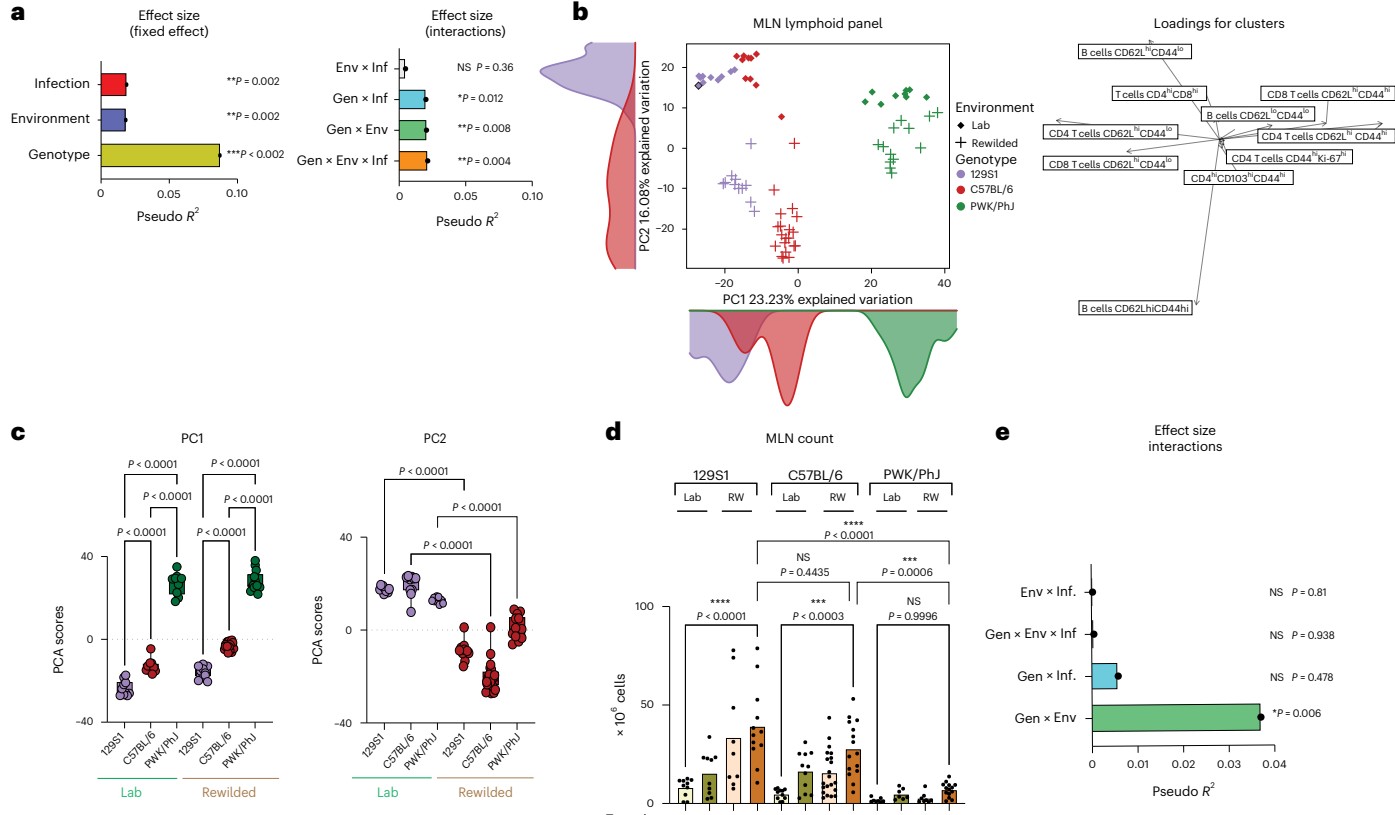

**Fig. 2 | Gen × Env × Inf interactions drive MLN variation. a**, Bar plots showing the pseudo $R^2$ measure of effect size of predictor variables and interactions as calculated by MDMR. Block 2 ($n$ = 73; 21 129S1, 30 C57BL/6 and 22 PWK/PhJ mice). **b**, PCA of immune cell clusters identified by unsupervised clustering in the MLN with the lymphoid panel and the loading factor of each population along the PCA. Block 2 ($n$ = 73; 21 129S1, 30 C57BL/6 and 22 PWK/PhJ mice). **c**, Box plot showing variance on PC1 and PC2 axes of PCA plots in **a**. The box plot center line represents median, the boundaries represent IQR, with the whiskers representing the upper and lower quartiles ±1.5 × IQR; all individual data points are shown (129S1 Lab = 10, C57BL/6 = 10, PWK/PhJ Lab = 9, 129S1 RW = 11, C57BL/6 RW = 20, PWK/PhJ RW = 13). **d**, MLN cell count from each mouse group, $n$ = 136, 129S1 Lab

Uninfected = 10, 129S1 Lab *T. muris* = 10, 129S1 RW Uninfected = 9, 129S1 RW *T. muris* = 12, C57BL/6 Lab Uninfected = 11, C57BL/6 Lab *T. muris* = 11, C57BL/6 RW Uninfected = 21, C57BL/6 RW *T. muris* = 15, PWK/PhJ Lab Uninfected = 8, PWK/PhJ Lab *T. muris* = 7, PWK/PhJ RW Uninfected = 8, PWK/PhJ RW *T. muris* = 14. **e**, Pseudo $R^2$ measure of effect size of predictor variables and interactions as calculated by MDMR analysis based on MLN cell count ($n$ = 136 over two experimental blocks, 129S1 = 41, C57BL/6 = 58, PWK/PhJ = 37). Statistical significance was determined based on MDMR analysis with R package for **a** and **e** or based on one-way ANOVA test between different groups with GraphPad software for **c** and **d**. Data are displayed as mean ± s.e.m. in bar plots and dots represent individual mice. NS $P$ > 0.05; *$P$ < 0.05; **$P$ < 0.01; ***$P$ < 0.001; ****$P$ < 0.0001.

(MLNs) because they drain the intestinal tissues, which are most affected by *T. muris* infection (in the cecum), as well as by alterations to the gut microbiota. In contrast with the blood, MDMR analysis of immune composition of the draining MLNs based on unsupervised clustering of cells, as explained above with the lymphoid panel (Supplementary Data 5), showed significant effects of genotype, environment and infection in determining immune variation (Fig. 2a, left). An interactive effect of genotype, environment and infection (Gen × Env × Inf) also contributed to the variation in immune composition in the MLN (Fig. 2a, right). We visualize the contribution of genotype, environment and infection with *T. muris* to MLN immune composition through PCA of immune cellular compositional data from individual mice (Fig. 2b and Extended Data Fig. 3a). The PCA plot showed prominent effects of genotype on variation along the PC1 axis (Fig. 2b,c), with effects of environment along the PC2 axis (Fig. 2b,c) and T. *muris* infection along the PC3 axis (Extended Data Fig. 3b,c).

To illustrate a Gen × Env × Inf interaction, we observed that *T. muris* infection had a significant effect on cellular composition of the draining MLNs with increased proportion and sometimes abundance of B cells, especially in the 129S1 and the C57BL/6 strains, and especially following rewilding (Extended Data Fig. 3a,b). The morphology of MLNs was quite different among mouse strains after rewilding, and this is reflected in the total cellular counts from the MLNs (Fig. 2d).

PWK/PhJ mice had smaller lymph nodes that were not expanded in size compared with C57BL/6 and 129S1 mice after rewilding and *T. muris* infection, illustrating a Gen × Env interaction that could be statistically quantified by MDMR (Fig. 2e).

Loading factors from the PCA (Fig. 2b, right) indicate that CD4 T and B cell populations in the MLNs showed differential effects of environment versus genotype in driving immune variation. As noted in the blood (Fig. 1e), expression of CD44 on CD4 T cells was influenced by genotype (Fig. 3a,b) in the MLNs, with highest expression of CD44 on PWK/PhJ mice across all environments. Expression of CD44 on B cells, which usually depicts antigen-experienced B cells[29], was predominantly influenced by environment and infection (Fig. 3c–e), with rewilded *T. muris*-exposed mice of all genotypes having more CD44-expressing B cells than their counterparts in the vivarium (Fig. 3c–e and Extended Data Fig. 4a–c). A similar genotype effect in the CD4 T cell compartment was also observed for other memory markers such as PD1, where expression of PD1 was also highest in the rewilded C57BL/6 and PWK/PhJ strain of mice (Extended Data Fig. 4d,e). Central memory CD4 and CD8 T cells expand following rewilding in the PWK/PhJ and C57BL/6 strains of mice (Extended Data Fig. 5a–e) as previous noted[19,23]. MDMR analysis of the different CD4 and CD8 T cell pools (Supplementary Data 6) shows that a Gen × Env interaction contributes to variation in the different T cell pools, with a residual fixed main genotype effect

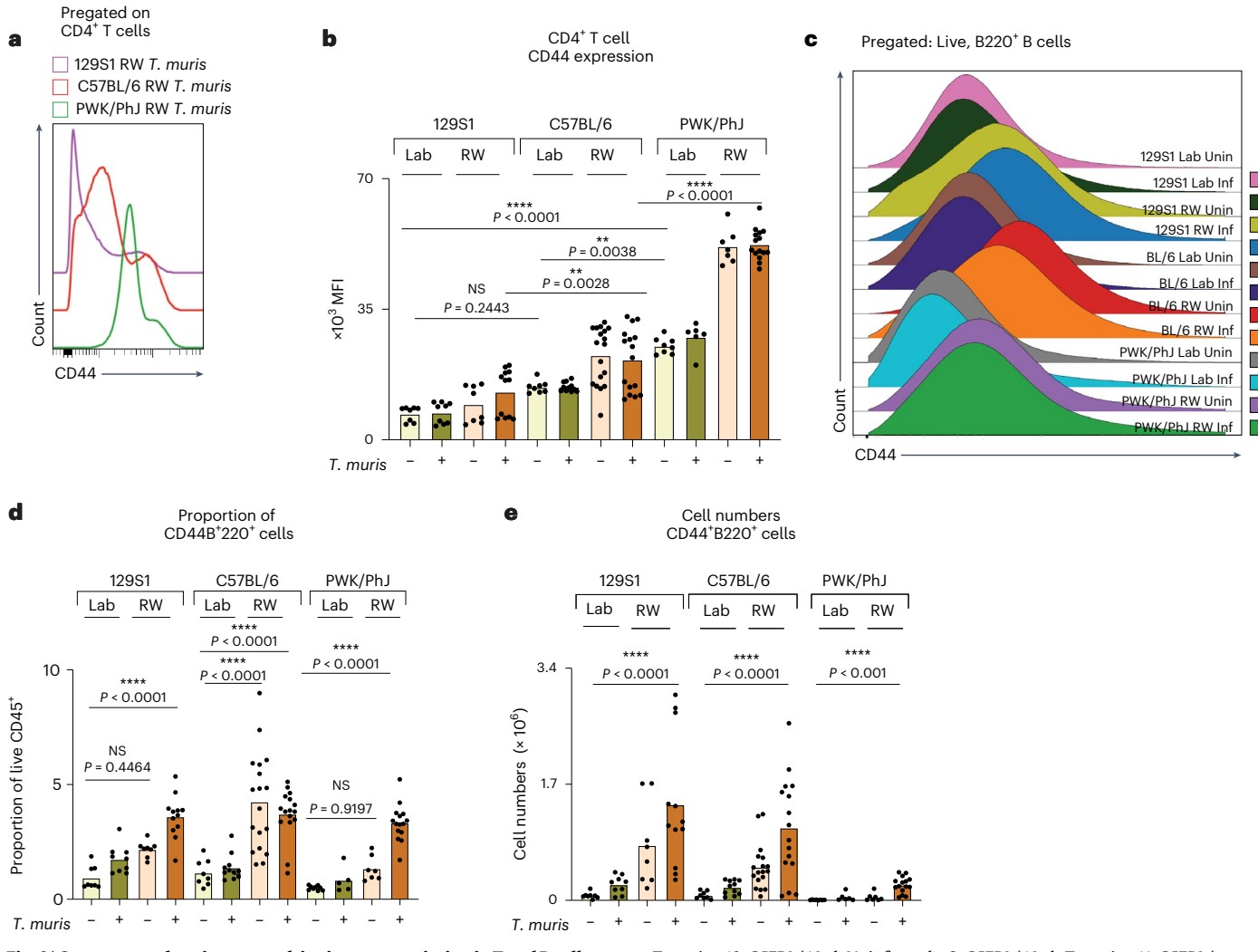

**Fig. 3 | Genotype and environment drive immune variation in T and B cell responses. a**, Representative histogram from Blocks 1 and 2 showing concatenated files from *T. muris*-infected and rewilded mice of each mice strain. **b**, Bar plots depicting MFI of CD44 on MLN CD4⁺ T cells. **c–e**, Representative histogram showing concatenated files from different groups of mice in Block 2 (**c**) with corresponding bar plots depicting proportion (**d**) and numbers (**e**) of B cells expressing CD44 on MLN cells. For **b**, **d** and **e**, *n* = 126; 129S1 Lab Uninfected = 8, 129S1 Lab *T. muris* = 9, 129S1 RW Uninfected = 8, 129S1 RW

*T. muris* = 12, C57BL/6 Lab Uninfected = 8, C57BL/6 Lab *T. muris* = 11, C57BL/ 6 RW Uninfected = 18, C57BL/6 RW *T. muris* = 16, PWK/PhJ Lab Uninfected = 8, PWK/PhJ Lab *T. muris* = 6, PWK/PhJ RW Uninfected = 7, PWK/PhJ RW *T. muris* = 15 over two experimental blocks. For **b**, **d** and **e** one-way ANOVA test was used to test statistical significance between the different groups of interest. Data are displayed as mean ± s.e.m. and for **b**, **d** and **e** bar plots dots represent individual mice. NS *P* > 0.05; *\**P* < 0.05; *\*\**P* < 0.01; *\*\*\**P* < 0.001; *\*\*\*\**P* < 0.0001. Inf, infected; RW, rewilded; Uninf, uninfected.

when proportions of cells were used for the analysis (Extended Data Fig. 5e). When analyzing absolute cell numbers, we found that Gen × Env similarly contributed to the variation in the T cell population. However, a residual main effect of environment was the predominant factor explaining the remaining variance, in contrast to the genotype effect that was prominent when we assessed cellular proportions (Extended Data Fig. 5f). The greater main fixed effect of environment on cell numbers might be due to environmentally acquired intestinal microbionts, meta organisms or food antigens increasing MLN numbers. Hence, genotype and infection (as well as Gen × Inf interactions) have a more substantial effect on immune phenotypes in the draining lymph nodes than in the peripheral blood. Differences in lymph node size between different genotypes, as well as the residence of *T. muris* in the cecum, illustrate why analyses of MLNs may reveal more Gen × Env × Inf interactions than the peripheral blood.

### Genotype has the biggest effect on cytokine levels

Cytokine response profiling is a common approach for immune phenotyping of patients to characterize immune responses. Supporting our

previous hypothesis[19], MDMR analysis of multiplex plasma cytokine data assessing systemic and circulating levels of IL-5, IL-17a, IL-22, IL-6, TNF and IFNγ (Supplementary Data 7) showed that there are no statistically significant interactions among genotype, environment and infection (Fig. 4a, right); and the main effect of genotype contributed to more variance than environment (Fig. 4a, left). However, there is a strong effect of the different experimental blocks (Fig. 4a), indicating that some unaccounted environmental or technical factors could also contribute to the variation. This genotype effect on plasma cytokines (circulating levels of IL-5, IL-17a, IL-22, IL-6, TNF and IFNγ, Supplementary Data 7) can be visualized on the PCA plot, where the C57BL/6 strain either in the laboratory or rewilded setting contributed to most of the difference on the PC1 axis (Fig. 4b). Assessment of the loading factors revealed that the IFNγ levels were important in driving this variance (Fig. 4c). Analysis of the individual cytokine data shows that the circulating IFNγ levels were especially high in infected C57BL/6 mice in both laboratory and rewilded settings (Fig. 4d).

When we characterized cytokine responses in the supernatant after in vitro stimulation of MLN cells, either with CD3/CD28 beads or

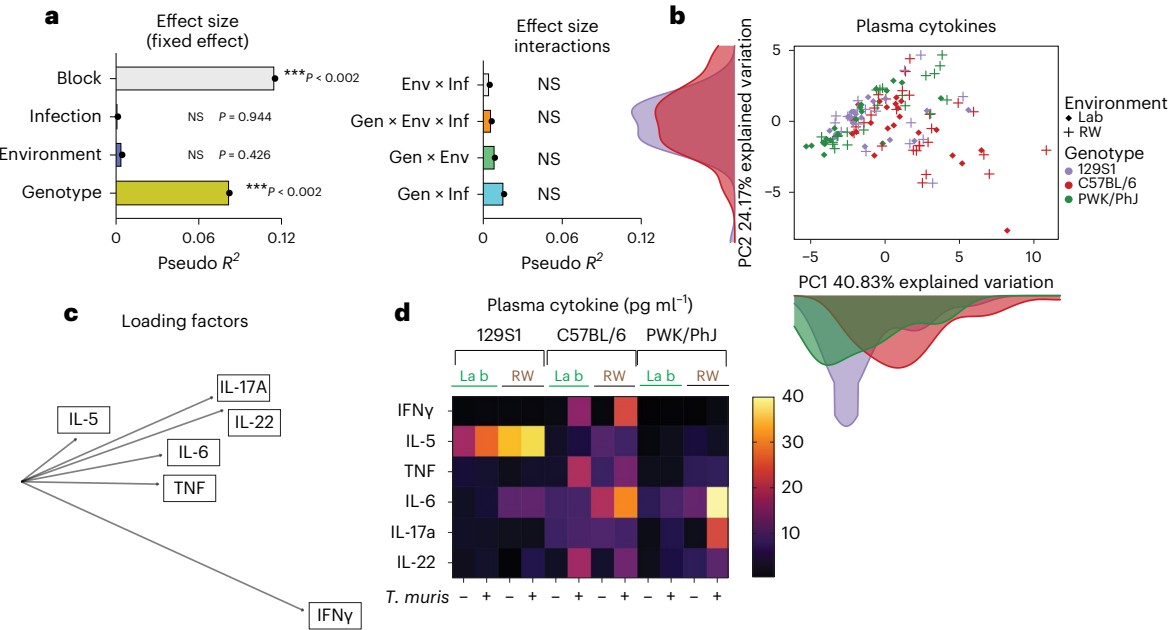

**Fig. 4 | Genotype has a bigger effect on plasma cytokine responses. a,** Bar plots showing the pseudo $R^2$ measure of effect size of predictor variables (left) and interactions (right) as calculated by MDMR from plasma cytokine data, $n = 129$; over two experimental blocks. **b,c,** PCA showing circulating plasma cytokine levels (**b**), raw data (Supplementary Data 6) and their loading factors (**c**).

**d,** Heatmap depicting circulating plasma cytokine levels in $n = 129$; 48 129S1, 39 C57BL/6 and 42 PWK/PhJ mice over two experimental blocks, Block 1 and Block 2. Data were transformed to ensure normality before analysis. NS $P > 0.05$; *$P < 0.05$; **$P < 0.01$; ***$P < 0.001$; ****$P < 0.0001$.

with other microbial stimulants (lipopolysaccharide, *Candida albicans*, *Clostridium perfringens*, *Bacteroides vulgatus* and *T. muris* antigens) (Extended Data Fig. 6a and Supplementary Table 4), MDMR analysis revealed genotype as having the biggest effect size on variation (Fig. 5a, left), which is consistent with the analysis of plasma cytokines. However, MDMR analysis of MLN cytokine responses (Supplementary Data 8) also showed that the effect of genotype on cytokine responses following stimulation of MLN cells with microbial antigens can be modulated by environment and infection (Gen × Env and Gen × Inf interactions) (Fig. 5a, right). When we focused our analysis on only cytokine responses to *T. muris*, we confirmed that genotype has the biggest effect on cytokine recall responses to *T. muris* antigen (Fig. 5b, left). However, this response also shows significant effects of Gen × Env, Gen × Inf and Gen × Env × Inf interactions (Fig. 5b, right). Analysis of the MLN supernatant cytokine data shows that consistent with the plasma cytokine data (Fig. 4d), production of IFNγ from MLN cells also tends to be higher in C57BL/6 mice compared with the 129S1 strain of mice following *T. muris* infection in the laboratory or the rewilded environment (Fig. 5c,d), demonstrating Gen × Env and Gen × Inf interactions. In addition, we observed similar Gen × Env and Gen × Inf interactions in other cytokine responses such as in production of IL-4 and IL-17. For example, we observed that IL-4 cytokine levels increase over baseline following exposure to *T. muris* in rewilded mice during recall responses only in the 129S1 mice and not in the other strains of mice (Extended Data Fig. 6b,c). On the other hand, responses to IL-17A expand over baseline only in the rewilded environment and following exposure to *T. muris* only in the C57BL/6 strain of mice and not in the other strains of mice (Extended Data Fig. 6d,e).

Together, these results support our previous observations that genetics influence cytokine responses more strongly than the environment[19]. However, here, we add evidence that the environment neither amplified nor eroded genetic effects on plasma cytokine levels, but that both environment and infection can modulate cytokine production in the antigen-stimulated MLNs, which are generally not accessible in human studies.

## Single-cell RNA sequencing validates Gen × Env interactions in immune variation

Single-cell RNA sequencing (scRNA-seq) is an unbiased approach to profile immune phenotypes without preselection for analytes and markers of interest. Here, we used scRNA-seq to examine effects of Gen × Env × Inf interactions on immune composition and cytokine responses in the MLN cells. MLN cells ($n = 49,727$) from individual mice ($n = 122$) identified 23 major immune cell subsets visualized by uniform manifold approximation and projection (UMAP) (Fig. 6a and Extended Data Fig. 7a). The cellular composition for each individual mouse based on cluster membership with these 23 major immune cell subsets (Extended Data Fig. 7b) is then used as the outcome variable for the MDMR analysis. In accordance with the cellular composition analysis with the flow cytometric data, MDMR analysis of the scRNA-seq compositional dataset (Supplementary Data 9) showed significant effects of genotype, environment and infection with *T. muris* in explaining immune variation as fixed predictor variables in addition to a substantial block effect (Fig. 6b, left). Genotype and environment (Gen × Env) interactions also contributed to significant variation in immune composition as assessed by scRNA-seq (Fig. 6b, right). PCA of the cellular composition from the single-cell sequencing analysis (Extended Data Fig. 7b and Supplementary Data 9) of the different individual mice reveals contributions of genotype and environment to the variation among individual mice along the PC1 and PC2 axes (Fig. 6c). An example of how genotype effects can be modulated by environment (Gen × Env interaction) can be observed in the increase of follicular B cells following rewilding which was especially heightened in C57BL/6 mice (Fig. 6d). In contrast, a trend towards a decrease in CD4 T cell abundance from rewilded naive mice occurred for both 129S1 mice and C57BL/6 mice in the laboratory environment, but not PWK/PhJ mice (Fig. 6e). Overall, examination of cellular composition in the MLNs by scRNA-seq resulted in a similar conclusion to spectral cytometry, in that Gen × Env interactions are particularly important. However, since we did not perform scRNA-seq on the peripheral blood, we could not directly compare if Gen × Env interactions are more important in

**a**  MLN re-stimulation assay supernatant cytokines (all antigens)

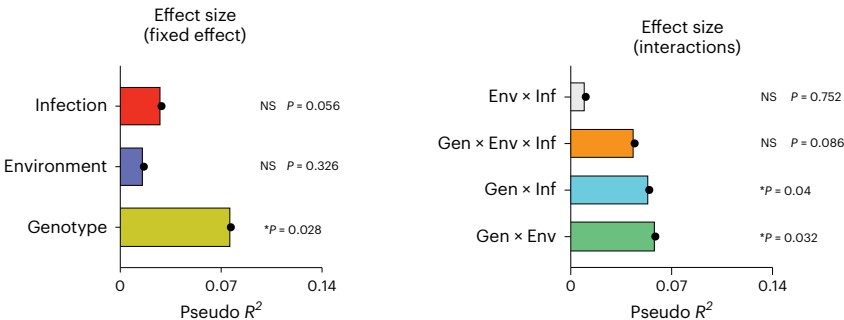

**c**  MLN supernatant -media alone (IFNγ)

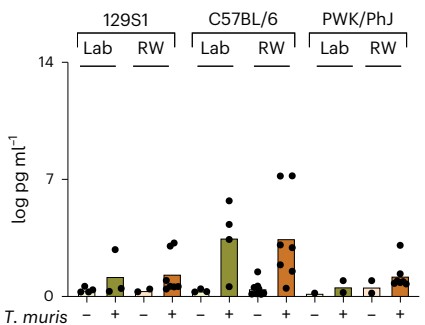

**b**  MLN re-stimulation assay supernatant cytokines  (*T. muris* antigen alone)

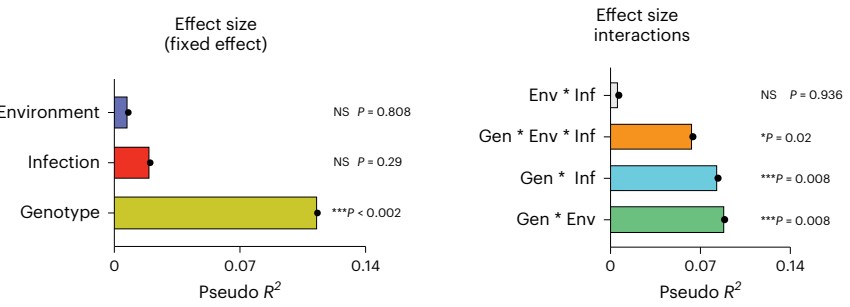

**d**  MLN supernatant - *T. muris* antigen (IFNγ)

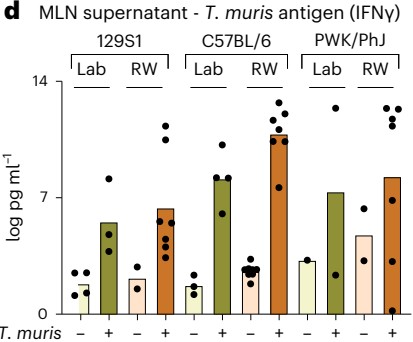

**Fig. 5 | Gen × Env interaction has a bigger effect on supernatant cytokine responses. a**, Bar plots showing the pseudo $R^2$ measure of effect size of predictor variables and interactions as calculated by MDMR from cytokine supernatant data of MLN cells stimulated with CD3/CD28 beads, lipopolysaccharide (LPS), *C. albicans*, *Clostridium perfringes*, *B. vulgatus* or *T. muris* antigens. **b**, Bar plots showing the pseudo $R^2$ measure of effect size of predictor variables and interactions as calculated by MDMR from cytokine supernatant data of MLN cells stimulated with *T. muris* antigen. For **a** and **b**, $n$ = 50; 16 129S1, 22 C57BL/6 and 15 PWK/PhJ mice in one experimental block, Block 2; raw data, Supplementary Data 8. **c,d**, Bar plots showing transformed IFNγ cytokine levels in the supernatant for

controls (**c**) as well as following stimulation with *T. muris* antigen (**d**). For **c** and **d**, $n$ = 50; 129S1 Lab Uninfected = 4, 129S1 Lab *T. muris* = 3, 129S1 RW Uninfected = 2, 129S1 RW *T. muris* = 7, C57BL/6 Lab Uninfected = 3, C57BL/6 Lab *T. muris* = 4, C57BL/6 RW Uninfected = 8, C57BL/6 RW *T. muris* = 7, PWK/PhJ Lab Uninfected = 1, PWK/PhJ Lab *T. muris* = 2, PWK/PhJ RW Uninfected = 2, PWK/PhJ RW *T. muris* = 7 over one experimental block, Block 2. Statistical significance was determined based on MDMR analysis with R package for **a** and **b**. Data included samples from Block 2 alone due to technical problems with stimulation assays from Block 1. Data below limit of detection were excluded. Data are displayed as mean ± s.e.m. and for **c** and **d** bar plots dots represent individual mice.

contributing to variance in the cellular composition of MLNs than in peripheral blood with this approach.

In addition to the interactive effects of genotype and environment, the compositional analysis based on scRNA-seq also identified independent effect of genetics, environment and infection with *T. muris* (Fig. 6b), which is consistent with the MLN spectral cytometry analysis (Fig. 2a). Hence, these factors can have independent effects on immune composition that are not dependent on other factors. Furthermore, in contrast with spectral cytometry, three-way interactions (Gen × Env × Inf) and other two-way interactions, Gen × Inf and Env × Inf, were not significant when immune composition analysis was done by scRNA-seq analysis (Figs. 2a and 6b). This difference may be driven by the determination of immune composition by protein markers compared with unbiased scRNA-seq, or by the total number of cells being analyzed. Nonetheless, the consistent conclusion of a significant Gen × Env interaction in both analyses suggests that this interaction is particularly critical in determining immune variation in the MLN.

### scRNA-seq validates Gen × Env interactions in cytokine variation

To characterize the functional activity of the MLN cells, scRNA-seq can identify the cells expressing cytokine-related genes. Based on Gene Ontology, we extracted data for 232 genes defined to have molecular function in cytokine activity (GO:0005125), of which expression of

123 genes could be identified in the scRNA-seq dataset (Supplementary Data 10). Expression levels of these genes ($n$ = 123) were used to subset and re-cluster the MLN cells, and they were visualized based on expression of cytokine activity genes and their original cellular identity (Fig. 7a). Notably, CD4 T cells and follicular B cells, which are the largest cellular populations in the overall dataset, had the smallest percentage of cells expressing cytokine genes (Fig. 7b), whereas CD8 effector cells, plasmablasts and dark zone germinal center B cells, which are less abundant in the total population, had higher proportions of cells expressing cytokine genes.

As described above, cells with cytokine activity were re-clustered based on their cytokine activity profiles (Fig. 7a), and cluster membership with these cytokine activity subsets (Supplementary Data 10) was then used as the outcome variable for the MDMR analysis. MDMR analysis showed that genotype had a significant effect on variation in cells with cytokine activity (Fig. 7c), which is consistent with our previous work[19] and with cytokine profiles described above. Also consistent with this analysis, other variables such as environment, infection with *T. muris* (Fig. 7c, left) and Gen × Env or Gen × Env × Inf interactions (Fig. 7c, right) had no significant effect on variation in the proportion of cells with cytokine activity as assessed by scRNA-seq. The genotype effect can be observed by plotting the percentage of MLN cells with cytokine activity for individual mice, with the 129S1 and PWK/PhJ mice having more cells expressing genes for cytokine activity than the C57BL/6 mice (Fig. 7d). PCA visualization

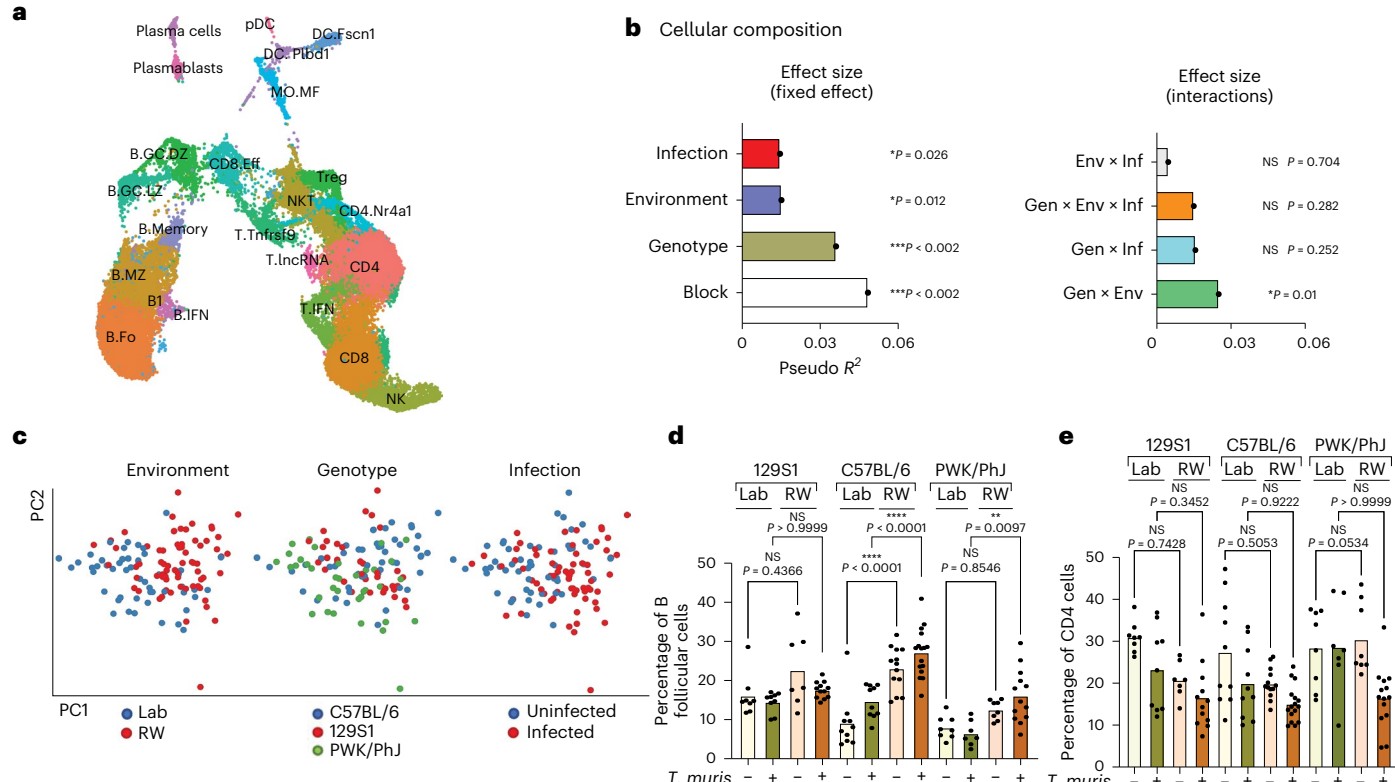

**Fig. 6 | Single-cell sequencing analysis for assessing immune variation in cellular composition. a**, UMAP visualization of scRNA-seq data identifying 23 major immune cell subsets, Block 1 and Block 2. **b**, Bar plots showing the pseudo $R^2$ measure of effect size of predictor variables and interactions as calculated by MDMR analysis based on cellular composition of cells identified in **a** in each mouse (Supplementary Data 7) (Block 1, $n = 51$, 17 129S1, 21 C57BL/6, 13 PWK/PhJ; Block 2, $n = 71$, 19 129S1, 28 C57BL/6 and 24 PWK/PhJ mice). **c**, PCA of MLN cellular compositional data as determined by scRNA-seq analysis. **d,e**, Bar plots showing percentages of B follicular cells (**d**) and CD4 T cells (**e**) based on the scRNA-seq identified in **a**. For **d** and **e**, $n = 122$; 129S1 Lab Uninfected = 8,

129S1 Lab *T. muris* = 9, 129S1 RW Uninfected = 7, 129S1 RW *T. muris* = 12, C57BL/6 Lab Uninfected = 10, C57BL/6 Lab *T. muris* = 10, C57BL/6 RW Uninfected = 13, C57BL/6 RW *T. muris* = 16, PWK/PhJ Lab Uninfected = 8, PWK/PhJ Lab *T. muris* = 7, PWK/PhJ RW Uninfected = 10, PWK/PhJ RW *T. muris* = 10 over two experimental blocks. Statistical significance was determined based on MDMR analysis with R package for **b**; for **d** and **e**, one-way ANOVA test with comparison by Tukey's multiple analysis was used to test statistical significance between the different groups of interest. Data are displayed as mean ± s.e.m. and for **d** and **e** bar plots dots represent individual mice. NS $P > 0.05$; *$P < 0.05$; **$P < 0.01$; ***$P < 0.001$; ****$P < 0.0001$.

of cellular composition based on cluster membership with cells of similar cytokine activity also showed distinct genotype differences along the PC1 axis (Extended Data Fig. 7c). MDMR analysis of the total number of cytokine-producing cells in the MLN (Supplementary Data 11) shows that Gen × Env interaction contributes the most to variation in number of cytokine-producing cells, with a residual fixed effect of environment contributing to the rest of the variation (Extended Data Fig. 7d). This can be observed by plotting the number of cells with cytokine activity for the individual mice with the genotype of the mice determining the magnitude of the effect of the environment on immune variation (Fig. 7e).

An unbiased scRNA-seq approach therefore supports the conclusion that genotype has the biggest effect on cytokine response heterogeneity based on proportion of cytokine-expressing cells, whereas cellular composition and numbers are driven more by interactions between genotype and the environment. The effect of genotype on cytokine response in the MLNs can be observed in feature plots where expression of IFNγ was examined (Extended Data Fig. 8a). Here, we noted that genotype influenced relative expression of IFNγ, with the greatest expression of IFNγ transcripts in the C57BL/6 strain of mice. There was also increased expression of IFNγ transcripts following rewilding and exposure to *T. muris* (Extended Data Fig. 8a). Examination of other cytokines and chemokines in various cell types, such as the CD8 effector cells, dendritic cells and monocytes/macrophage populations, also shows a genotype effect in differential expression

of transcripts of these inflammatory mediators between the different strains of mice (Extended Data Fig. 8b–d), supporting the importance of genotype on variation in functional response and cytokine activity.

## Gen × Env and immune variation contribute to *T. muris* worm burden

Ultimately, the question remains as to how the variance in these genetic, environmental and immunological factors influences susceptibility to subsequent infection. Therefore, we investigated predictors of worm burden (Fig. 8) and the contribution of genetics, environment and the different immunological factors to susceptibility to worm infection. Here, we observed that despite all 74 *T. muris*-exposed mice receiving approximately the same infectious dose (200 eggs), worm burden was negative binomially distributed among exposed mice (Fig. 8a and Supplementary Data 12). Analysis of worm burdens was done using generalized linear models with a negative binomial error distribution. We found a significant Gen × Env for worm burden (Fig. 8b, $P = 0.04015$), whereby C57BL/6 mice harbored more worms than the other genotypes in the vivarium, but rewilding was associated with higher worm burdens in all genotypes. In other words, the relative susceptibility of the different host strains to *T. muris* depended upon environment (paralleling[30]). When we used logistic regression to analyze worm presence/absence at the experimental endpoint (reported as prevalence of infection among exposed mice in Fig. 8b), significant effects in the best model included main effects of only genotype ($P = 0.0001221$)

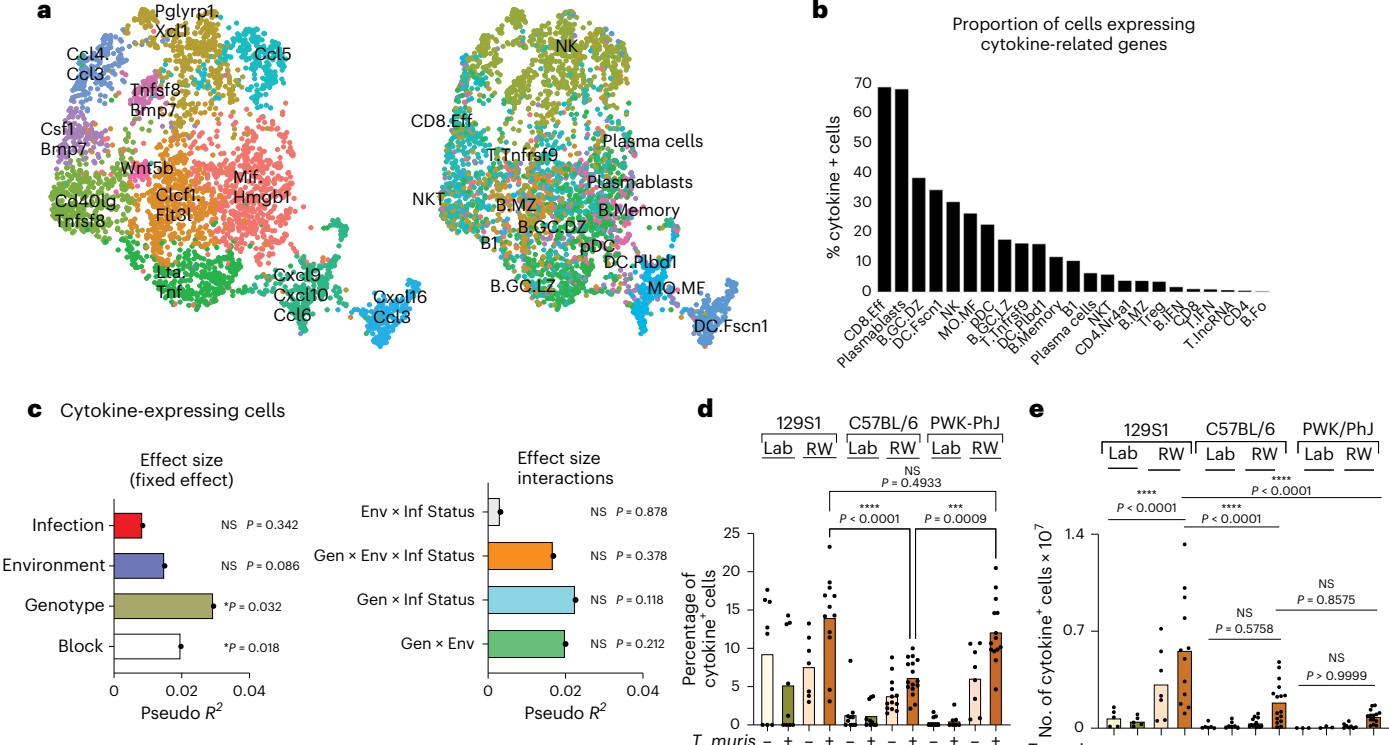

**Fig. 7 | Single-cell sequencing analysis for assessing immune variation in cytokine profiles. a**, Cytokine-expressing cell clusters. **b**, Proportion of cells expressing cytokine-related genes of those identified in **a**. **c**, Bar plots showing the pseudo $R^2$ measure of effect size of predictor variables and interactions as calculated by MDMR analysis based on data from proportion of cytokine-expressing cells identified in **a**. **d**,**e**, Bar plots showing proportion (**d**) and numbers (**e**) of cytokine-expressing cells identified in **a**, For **c**, **d** and **e**, n = 99; 129S1 Lab Uninfected = 5, 129S1 Lab *T. muris* = 5, 129S1 RW Uninfected = 7, 129S1 RW *T. muris* = 12, C57BL/6 Lab Uninfected = 6, C57BL/6 Lab *T. muris* = 7, C57BL/6

RW Uninfected = 13, C57BL/6 RW *T. muris* = 16, PWK/PhJ Lab Uninfected = 3, PWK/PhJ Lab *T. muris* = 3, PWK/PhJ RW Uninfected = 8, PWK/PhJ RW *T. muris* = 14 over two experimental blocks. Statistical significance was determined on MDMR analysis with R package for **c**. For **d** and **e**, one-way ANOVA test with comparison by Tukey's multiple analysis was used to test statistical significance between the different groups of interest. Data are displayed as mean ± s.e.m. and for **d** and **e** bar plots dots represent individual mice. NS $P > 0.05$; *$P < 0.05$; **$P < 0.01$; ***$P < 0.001$; ****$P < 0.0001$.

and environment ($P = 0.0044835$), plus a significant effect of replicate experiment (that is, Block; $P = 0.0329262$).

Interestingly, when we included PC1 and PC2 values from the MLN scRNA-seq analysis (Fig. 8b) as summary measures of immune variation among individual mice, significant effects in the best model of worm burden (Fig. 8c) included main effects of only genotype ($P = 0.0003322$), environment ($P = 0.0015615$) and PC2 scores ($P = 0.0108213$), which had a significant negative association with worm burden. Loading factors on the PC2 axis (Extended Data Fig. 9a) indicated that the dearth of T cells with an interferon signature (*T.IFN*) may be a driver of the relationship between high PC2 scores and decreased worm burden. Furthermore, the fact that PC2 explained more variance than Gen × Env suggests that environment-dependent differences in worm burdens among and within genotypes may hinge on immune factors captured on PC2 (Extended Data Fig. 9a). These results are consistent with increased differential expression of IFNγ transcripts in the C57BL/6 strain of mice based on scRNA-seq (Extended Data Fig. 8a), with reports of T$_H$1 responses being associated with increased susceptibility to helminth colonization[31,32], and suggest that despite complexities in how immune phenotype is influenced by genetics and environment, once that immune phenotype emerges, established 'rules' of infection susceptibility apply (as in ref. 20).

Quantification of goblet cell count as a measure of effector type 2 response[33–35] showed no significant differences between laboratory or rewilded environment in different strains of mice before exposure to *T. muris* (Extended Data Fig. 9b). Furthermore, flow cytometric analysis of cytokine production in the three different inbred strains

of mice under laboratory condition at day 14 post challenge with *T. muris* eggs from MLN cells and following in vitro stimulation with a cell activation cocktail (phorbol 12-myristate-13-acetate, ionomycin and protein transport inhibitor (Brefeldin A)) showed that increased levels of IFNγ, a type 1 cytokine in the CD4$^+$ T cells, rather than differences in production of type 2 cytokines IL-4 and IL-13 (Fig. 8d and Extended Data Fig. 9b), might explain variation in worm burden and prevalence, especially in the C57BL/6 strain of mice.

Together, these results suggest that genetic, environmental and individual immune variation as it relates to differential levels of type 1 immune responses and IFNγ is associated with varied infection burden.

## Discussion

Our results show that the effect of even an extreme environmental shift on immune traits is modulated by genetics. Interactions between environment and genotype are thus an important source of variation in immune phenotypes. While we previously proposed that the immune cell composition for an individual is primarily shaped by the environment[19], we find here that environmental effects on cellular composition are shaped by interaction with mouse genotype. The complexity of Gen × Env × Inf interactions has important ramifications for the course of natural selection on the immune system, immunogenetic diversity and efficacy of vaccines. For example, because any given genotype may produce different immune responses in different environments, environment can alter the ability of individuals to resist and tolerate infections; furthermore, natural selection operating on such variation is likely to generate divergent allele frequencies in different environments[36].

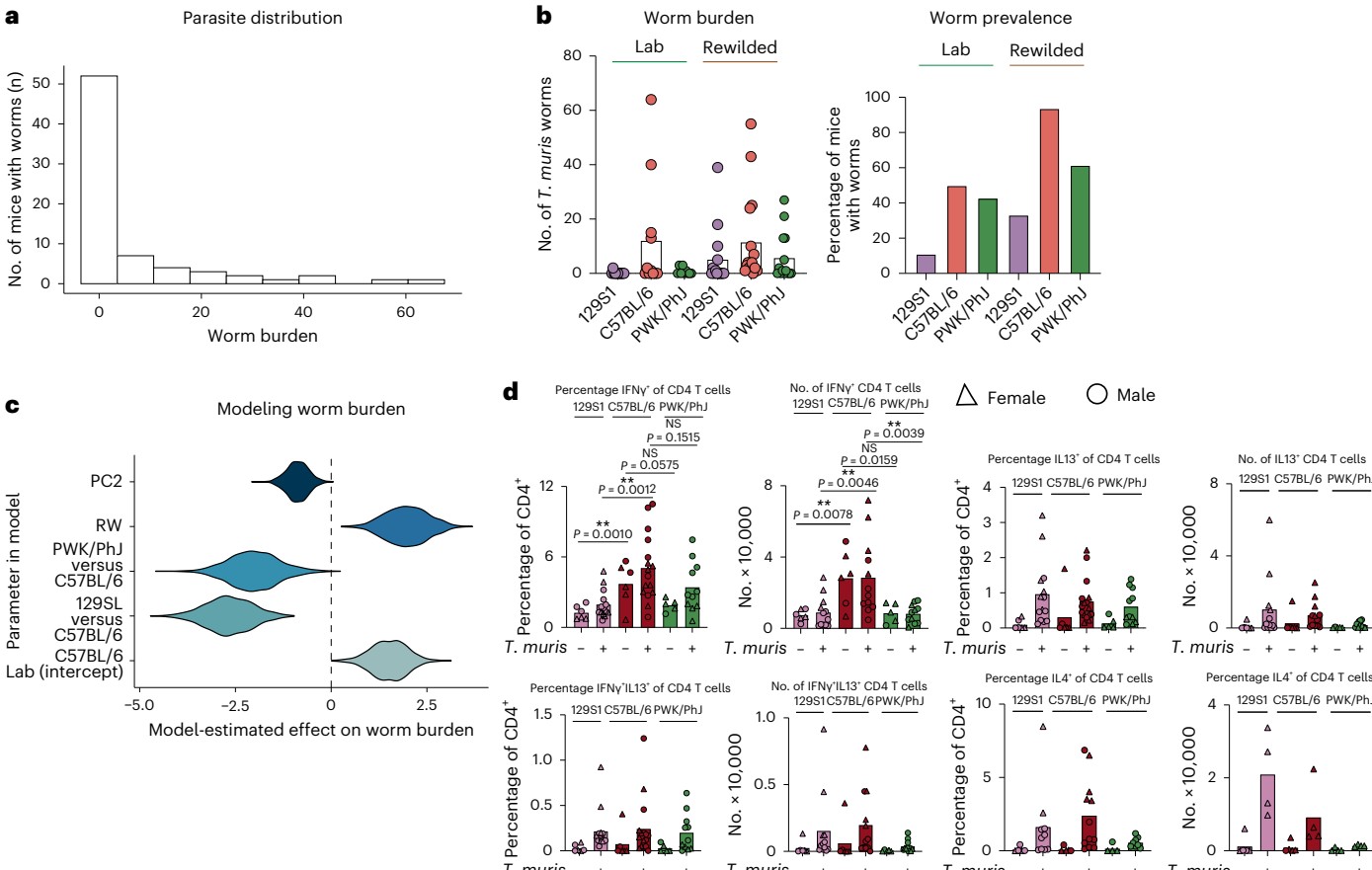

**Fig. 8 | Genetics and environmental factors predict outcomes during exposure with *T. muris* parasite.** Significant variation in worm burden among exposed mice, 3 weeks after inoculation with 200 eggs of *T. muris* per host, Block 1 and Block 2, Supplementary Data 12. **a**, Worm burden (number of nematodes remaining in the cecum at that timepoint) followed a negative binomial distribution. **b**, Worm burden depicted as number of worms per mouse (left, *n* = 75, C57BL/6 Lab = 11, 129S1 Lab = 11, PWK/PhJ Lab = 9, C57BL/6 RW = 16, 129S1 RW = 14, PWK/PhJ RW = 14, dots represent individual mice) and percentage of mice (Prevalence) still infected by worms (right). Each was predicted by a combination of genetic and environmental effects, including Gen × Env for worm burden (see text). **c**, When we used PC2 from the scRNA-seq data (Fig. 3b) as an index of immune variation among individuals in our statistical models, we found that Gen × Env was no longer significant. Instead, the best model included main effects of host strain (C57BL/6 versus 129SL versus PWK/PhJ), environment

(Lab versus RW) and PC2. The figure depicts 1,000 model-estimated values for the effect of each predictor on worm burden. The three different inbred strains of mice, 129S1, C57BL/6 and PWK/PhJ mice, were infected with *T. muris* under laboratory conditions, and at day 14 post infection, MLN cells were collected and stimulated with PMA/ION. **d**, The proportion and numbers of CD4+ T cells producing IFNγ, IL-13 and IL-4 at day 14 following infection with *T. muris* are calculated and displayed as mean ± s.e.m.; dots represent individual mice. NS *P* > 0.05; **P* < 0.01; females (triangular dots), males (circle dots). For IFNγ, IL-13, *n* = 48; 129S1 Uninfected = 6, 129S1 *T. muris* = 14, C57BL/6 Uninfected = 6, C57BL/6 *T. muris* = 16, PWK/PhJ Uninfected = 5, PWK/PhJ *T. muris* = 11 over 5 experiments, For IL-4, *n* = 44; 129S1 Uninfected = 5, 129S1 *T. muris* = 10, C57BL/6 Uninfected = 5, C57BL/6 *T. muris* = 12, PWK/PhJ Uninfected = 4, PWK/PhJ *T. muris* = 8 over 4–5 experiments. PMA/ION, phorbol myristate acetate/ionomycin.

Quantification of such interactions is rare in immunological studies, and this is a valuable step forward in understanding the evolution and function of the immune system. Rewilding can combine controlled experiments with the advantages of tissue accessibility and homozygous mouse genetics, with multi-dimensional immune phenotyping analyses applied in human immunology. We can identify specific traits for which main (that is, noninteraction) effects are dominant. Heterogeneity in proportions of cells producing cytokines shows a stronger influence of genetics, consistent with human studies[37], while heterogeneity in absolute cell numbers shows a stronger influence of the environment, consistent with studies of microbial exposure in mice[8]. The Human Functional Genomics Project also reported that variation in proportion of T cell phenotypes is more influenced by genetics, while B cell phenotypes are more influenced by nonheritable environmental factors[1]. While this observation remains unexplained, perhaps B cell responses are more influenced by the environment because their populations are driven more by microbial exposure as a result of direct activation through the B cell receptor, whereas underlying

genetic differences in the major histocompatibility complex (MHC) or human leukocyte antigen molecules that present antigen[38] have a larger effect on T cell phenotypes. However, the complexity and interdependence of B and T cell responses to infections makes it difficult to fully understand the differential contribution of environment and genetics to these adaptive immune cell populations. Different immunological readouts also are differentially impacted by genetics versus environment. We found here that genotype and infection explain more variation in lymph nodes than in peripheral blood. Since most human studies are restricted to peripheral blood, the effects of environment may appear more pronounced than if other tissue samples were analyzed. Our analysis and experimental design with the rewilding model is an opportunity to assess the contribution of Gen × Env interactions to various immune traits in different tissues, which is not feasible in human studies. This provides a bridge towards a better understanding of immune variation compared with specific pathogen-free mice[17].

In comparing our results with human studies, we note that human populations harbor greater heterozygosity and rarely undergo such

dramatic environmental shifts as the laboratory mice being released outdoors. Longitudinal studies on travelers, refugees or immigrants may perhaps reveal similar alterations in immune phenotypes driven by environmental changes. The effect size of genotype versus environment on immune profile might also be influenced by other factors such as the age of the individual when the environmental change occurs. A newborn might be more influenced by environmental factors than an adult[39,40]. We used only female mice between 5 and 10 weeks of age, reflecting a young adult population of a single sex[41]. Releasing younger mice or allowing sexual reproduction to occur in the rewilded environment is a subject for further investigation. Interindividual variation tends to accumulate with age[42] and sex affects susceptibility to *Trichuris* infections[31,43,44], but these questions remain open for future investigation. Using inbred strains of mice with homozygous alleles may also represent an extreme test of genetic influences on immune variation. Compared with inbred strains of mice, most human genomes exist in a predominantly heterozygous state. Hence, despite important differences between this study and studies on human populations, there is surprising consistency regarding the differential roles of environment and genetics in B and T cell traits as well as the role of genetics in explaining variation in cytokine responses[1,37].

These experiments were performed in two consecutive experimental blocks over one summer, which contributes substantially to variation in the data. By including and accounting for block effects statistically, we can quantify independent effects of genotype, environment and infection, as well as interactions between these variables, while excluding experimental variation. Block effects include technical variation, plus seasonal environmental differences, which may also explain differences in worm burden from our previous report. In general, the remarkable expansion of the neutrophil and eosinophil pool across all genotypes and the effect of rewilding on the proportion of mice infected with *T. muris* are consistent across different experiments over several years (refs. 20,23,45). However, other outcome measures, such as the number of worms recovered per mouse, were more affected by experimental block, further emphasizing effects of the environment on immuno–parasitological outcomes.

Variation in burden of soil-transmitted helminths between individuals typically follows a negative binomial distribution. While our results suggest that the immune consequences of Gen × Env interactions could contribute to the negative binomial distribution in worm burdens in natural populations, we cannot distinguish between worms that are in the process of being expelled and the ones that will survive till patency. Also, natural helminth infection occurs from trickle infection of multiple small doses of egg exposures; therefore, a high-dose *T. muris* infection may not be representative of real-world exposure. Nonetheless, in this system where interactions between genetics and environment can be quantified, the basic $T_H1$ versus T helper 2 cell immunological mechanisms that govern susceptibility to *T. muris* infection still predominate[20,31], highlighting how basic immunological mechanisms discovered in the specific pathogen-free facilities can be rigorously tested in a more naturalized system[17].

Unexpectedly, we did not observe a strong type 2 signature in the MLN and goblet cell responses despite type 2 responses being well documented in worm expulsion. Instead, variation in type 1 responses outdoors might tip the balance between type 1 and type 2 responses and thus differences in worm burden. Our previous rewilding study also found no difference in *T. muris* worm burden between rewilded C57BL/6 and STAT6KO mice[20] and IL-13+CD4+ T cells did not differ between laboratory and rewilded mice in the intestinal lamina propria at day 21 post infection[20]. The higher type 1 signature associated with increased worm burden in the C57BL/6 strain of mice is consistent with earlier studies describing the key role of type 1 cytokines, especially IFNγ, in suppressing the protective response during *T. muris* infection[46,47]. Type 1 cytokines might have other unexplored roles to play in intestinal helminth infections.

An interesting observation is the reduction in genetically driven immune phenotype differences in laboratory mice under rewilding conditions. Perhaps immune phenotypes may be more extreme in the absence of intensive microbial exposures and therefore have a greater impact in genetically susceptible individuals. One element of the hygiene or old friends hypothesis is that improved immune-regulatory responses through microbial exposure reduce the prevalence of inflammatory conditions[48–50]. Our results raise the possibility that increased microbial exposure may normalize or reduce the variation of immune phenotypes, hence reducing the number of individuals with extreme immune responses.

In conclusion, our results highlight how rewilding mice with controlled genetic backgrounds could be a bridge towards understanding causes, tissue specialization and consequences of immune variation between human individuals, and that quantification of the interactions at this interface may help elucidate the evolution of the immune system.

## Online content

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

[1]Laboratory of Parasitic Diseases, National Institute of Allergy and Infectious Diseases, National Institutes of Health, Bethesda, MD, USA. [2]Department of Ecology and Evolutionary Biology, Princeton University, Princeton, NJ, USA. [3]Department of Primate Behavior and Evolution, Max Planck Institute for Evolutionary Anthropology, Leipzig, Germany. [4]Department of Microbiology, Immunology, and Molecular Genetics, University of Texas Health Sciences Center at San Antonio, San Antonio, TX, USA. [5]Department of Pathology, New York University, Grossman School of Medicine, New York, NY, USA. [6]Kimmel Center for Biology and Medicine at the Skirball Institute, New York University, Grossman School of Medicine, New York, NY, USA. [7]Division of Gastroenterology and Hepatology, Department of Medicine, University of Pennsylvania Perelman School of Medicine, Philadelphia, PA, USA. [8]Department of Pathobiology, University of Pennsylvania School of Veterinary Medicine, Philadelphia, PA, USA. [9]Santa Fe Institute, Santa Fe, NM, USA. [10]Present address: Institute of Biomedical Sciences, Academia Sinica, Taipei, Taiwan. [11]These authors contributed equally: Oyebola Oyesola, Alexander E. Downie. ✉e-mail: oyebola.oyesola@nih.gov; algraham@princeton.edu; png.loke@nih.gov

## Methods

### Study design

**Mice and rewilding.** C57BL/6J, 129S1/SV1mJ and PWK/PhJ mice were purchased from The Jackson Laboratory and were housed under specific pathogen-free conditions with ad libitum access to food and water. All mouse lines were then bred onsite in a specific pathogen-free facility at the National Institutes of Health. The resulting littermates from the multiple breeding pairs were shipped to Princeton University where they were acclimated in a dedicated animal facility to temperatures and light cycles characteristic of summer in New Jersey (26 °C ± 1 °C, and a 15-h light/9-h dark cycle)[22]. Following this, mice were randomly assigned to either remain in the institutional vivarium (Lab mice) or be released into the outdoor enclosures (Rewilded mice) previously described[19,20,22,23]. For all rewilding experiments, only female mice were used to prevent unintended breeding in the rewilded environment, and mice were between 8 and 12 weeks at point of blood draw following rewilding. The protocols for mouse breeding were approved by the National Institute of Allergy and Infectious Diseases Animal Care and Use Committee, Protocol no. LPD 16E. The protocols for releasing the laboratory mice into the outdoor enclosure facility were approved by Princeton Institutional Animal Care and Use of Committee, Protocol no. 1982.

In total, 25–30 female mice of mixed strains and genotype, 129S1/SV1mJ, C57BL/6J and PWK/PhJ, were used for these experiments. Sample size was determined by logistical constraints and not by power calculations. For rewilding, 12–18 female mice of the different strains (129S1/SV1mJ, C57BL/6J and PWK/PhJ) were housed in different wedges in the enclosure for 5 weeks. In summary, for Block 1 we rewilded $n = 42$ mice (15 PWK/PhJ, 14 C57BL/6, 13 129S1) and in Block 2 we rewilded $n = 47$ mice (16 PWK/PhJ, 18 C57BL/6, 13 129S1), for a total of $n = 89$ rewilded mice. The rewilded enclosures, previously described in ref. 20 and used in refs. 19,22,23, are triangular wedges, ~180 m$^2$ in area and enclosed by 1-m-high walls of zinc-coated steel which penetrate into the ground by ~0.5 m. Concomitantly, ten mice of the different strains were left in the institutional vivarium (Lab mice) where the temperature and humidity were maintained as described above. Longworth traps baited with chow and peanut butter were used to catch the mice at approximately 2 weeks and 5 weeks after release[19]. At 2 weeks after release, 8–10 mice of each genotype were trapped, and blood and fresh stool were collected from the mice for longitudinal CBC analysis and microbiome analysis[22]. At the same time 2 weeks following rewilding, some of these mice, vivarium controls and the rewilded mice, were infected with 200 *T. muris* embryonated eggs by oral gavage. At approximately 5 weeks post rewilding and 19–21 d post *T. muris* infection, mice were recovered for analysis and worm count. We collected blood and MLNs for immune phenotyping and fecal samples for microbiota analyses. To assess immune cell composition, we analyzed CBC/DIFF values from total blood, PBMCs by flow cytometry with a lymphocyte panel (Supplementary Table 1) and MLN cells with both a lymphocyte and myeloid cell panel (Supplementary Tables 1 and 2). To assess cytokine responses, we measured plasma cytokine concentrations and stimulated MLN cells with microbial antigens and measured cytokines released in the supernatant. scRNA-seq of MLN cells enabled phenotyping of both immune cell composition and function. We also assessed worm burden and worm prevalence for *T. muris* at day 19–21 post infection before full worm maturation, as it was necessary to prevent shedding of *T. muris* eggs into the rewilded environment. Ceca were collected and the number of adult worms in each cecum were counted individually using an inverted microscope. Serology and PCR screening panels testing for over 30 pathogens indicated that the mice had no other detectable infections (Supplementary Data 1 and 2). For all analyses, samples that fail quality control, such as flow cytometry staining errors, high cell death and/or are under limit of detection such as for the ELISA assay, are not included in downstream statistical analyses.

Investigators were blinded to the experimental groups to which the mice belonged at the time of performing the different experimental assays but were unblinded at the point of statistical analysis and testing. For all analyses, samples that failed predetermined quality control such as flow cytometry staining errors, high cell death and/or are under limit of detection such as for the ELISA assay are not included in downstream statistical analyses. For the different measurements and assays, the same sample size was measured repeatedly except were mentioned in the figure legends. The number of mice per group, the number of experimental replicates, if any, and the statistical tests employed are reported in the figure legends. All data points represent biological replicates.

**CBC analysis.** Blood samples (approximately 30–50 µl) were collected from all mice at the endpoint via cheek bleeds using a Medipoint Golden Rod Lancet Blade 4MM (Medipoint NC9922361) into a 1.3-ml heparin-coated tube (Sarstedt, NC9574345). Blood samples were analyzed using the Element HT5 Veterinary Hematology Analyzer (Heska).

**PBMC preparation and isolation.** Heparinized whole blood collected via the cheek bleeds was mixed with blood collected via the cardiac puncture method. The combined blood samples were spun for 10 min at 1,500 rpm and plasma was collected and stored at −80 °C for further cytokine analysis. The cellular component re-suspended in PBS next underwent a density gradient separation process using the Lymphocyte Separation Media (LSM MP Biomedicals) according to the manufacturer's instructions. Isolated PBMCs were washed twice in PBS and then used for downstream spectral cytometric analysis. PBMC isolation was performed on 64 samples out of 74 mice in Block 2.

**Preparation of single-cell suspensions from MLNs.** Single-cell suspensions from the MLNs were prepared by mashing the tissues individually through a 70-µm cell strainer and washing with RPMI. Cells were then washed with RPMI supplemented with 10% FCS. Live cell numbers were enumerated using the Element HT5 Veterinary Hematology Analyzer (Heska). Samples with greater than 80% cell death were excluded for further analysis.

**Spectral cytometry and analysis.** Single-cell suspensions prepared from the PBMCs and MLNs were washed twice with flow cytometry buffer (FACS Buffer) and PBS before incubating with Live/Dead Fixable Blue (ThermoFisher) and Fc Block (clone KT1632; BD) for 10 min at 20–25 °C. Cocktails of commercially available manufacturer-validated fluorescently conjugated antibodies (listed in Supplementary Tables 1 and 2) diluted in FACS Buffer and 10% Brilliant Stain Buffer (BD) were then added directly to cells and incubated for a further 30 min at 20–25 °C. For the lymphoid panel, cells were next incubated in eBioscience Transcription Factor Fixation and Permeabilization solution (Invitrogen) for 12–18 h at 4 °C and stained with cocktails of fluorescently labeled antibodies against intracellular antigens diluted in Permeabilization Buffer (Invitrogen) for 1 h at 4 °C.

Spectral unmixing was performed for each experiment using single-strained controls using UltraComp eBeads (Invitrogen). Dead cells and doublets were excluded from analysis. All samples were collected on an Aurora spectral cytometer (Cytek) and analyzed using the OMIQ software (https://www.omiq.ai/), and data cleaning and scaling was done using algorithms such as FlowCut[20,23] within the OMIQ software. Subsampled cells including 10,000 live, CD45$^+$ cells were re-clustered in an unsupervised version using the JoesFlow software (GitHub: https://github.com/niaid/JoesFlow). In situations where traditional gating was done, an example flow plot depicting gating strategy is provided in Extended Data Fig. 10.

For flow cytometric analysis, due to batch effect and technical issues, samples from Block 1 and Block 2 were not combined. For PBMC analysis, 64 samples from Block 2 were processed for further

downstream analysis and for MLN analysis 73 samples of 74 were processed for further downstream analysis.

**MLN cell stimulation and cytokine profiling.** A single-cell suspension of MLN cells was reconstituted in RPMI at $2 \times 10^6$ cells per milliliter, and 0.1 ml was cultured in 96-well microtiter plates that contained $10^7$ colony-forming units per milliliter of UV-killed microbes, $10^5$ αCD3/CD28 beads (11456F) or lipopolysaccharide (100 ng ml$^{-1}$) (L2630), or PBS control. The stimulated microbes and antigens included were *B. vulgatus* (ATCC 8482), *C. albicans* (UC820), *C. perfringens* (NCTC 10240) and *T. muris* antigen, prepared in house as previously published[19,51–53]. Supernatants were collected after 2 d and stored at −80°C. Concentrations of IL-5, IL-6, IL-22, IL-17A, IFNγ, TNF, IL-2, IL-4, IL-10, IL-9 and IL-13 in supernatants were measured using a commercially available murine T helper cytokine LEGENDplex assay (Biolegend) panel (cat. no. 741044) according to the manufacturer's instructions. Plasma concentrations of IL-5, IL-6, IL-22, IL-17A, IFNγ and TNF were measured using the same panel according to the manufacturer's instructions. Cytokines levels that were lower than the limit of detection across samples were excluded from further analysis.

For intracellular staining, cells were stimulated with the Cell Activation Cocktail (Biolegend), a premixed cocktail with optimized concentration of phorbol 12-myristate-13-acetate, ionomycin and protein transport inhibitor (Brefeldin A), for 5 h at 37 °C. Cells were surface stained, fixed and permeabilized with BD Perm/Wash Buffer (BD Biosciences) according to the manufacturer's instructions followed by intracellular staining with monoclonal antibodies (Supplementary Table 1). Samples were acquired on the Aurora spectral cytometer (Cytek) and data were analyzed using the OMIQ software.

**scRNA-seq.** Single-cell suspensions were obtained from MLNs as described above. In total, 2,000 cells from each individual mouse (Block 1, $n = 51$; Block 2, $n = 71$) were labeled with the antibody hashtag oligonucleotides. These antibodies are a mix of anti-CD45 and anti-MHCI antibodies. TotalSeq-C antibodies are used with the Single Cell 5′ kit. Pooled samples from each group were then loaded on a 10X Genomics Next GEM chip and single-cell GEMs (gel beads in emulsion) were generated on a 10X Chromium Controller. Subsequent steps to generate complementary DNA and sequencing libraries were performed following the 10X Genomics' protocol. Libraries were pooled and sequenced using Illumina NovaSeq SP 100 cycles as per 10X sequencing recommendations.

The sequenced data were processed using Cell Ranger (v.6.0) to demultiplex the libraries. The reads were aligned to *Mus musculus* mm10 genomes to generate count tables that were further analyzed using Seurat (v.4.0). Sequencing data from the two blocks were integrated together before further downstream analysis. Data are displayed as UMAPs. The different cell subsets from each cluster were defined by the top 50 differentially expressed genes and identification using the SingleR sequencing pipeline[54,55]. Cell types with different cytokine expression were identified based on expression of genes related to cytokine function using the Gene Ontology Browser. The Seurat Analysis pipeline was used for comparisons between each of the different cell clusters of interest.

**Histopathological analyses of murine intestinal tissue.** At necropsy, the entire small intestine was excised and the cecum from *T. muris*-infected mice was saved for worm count. Following this, the distal ileum (6–8 cm) from representative uninfected mouse groups was then flushed with ice-cold PBS and 4% paraformaldehyde (PFA). Samples were then fixed in 4% PFA for 30 min at 20–25 °C, opened and rolled up into Swiss rolls. The Swiss rolls were fixed in 4% PFA overnight at 4 °C, washed with PBS and sequentially incubated first in 30% (w/v) sucrose overnight at 4 °C and then in fresh 30% sucrose for another 4–6 h. The samples were then embedded in Optimal Cutting Temperature

Compound (Tissue-Tek) and sectioned at 5 μm on a CM1950 Cryostat (Leica). Sections were stained with periodic acid–Schiff/Alcian blue (PAS/Alcian Blue). Images were acquired with a Hamamatsu Nano Zoomer S60 microscope (Hamamatsu) enabled with a ×40 objective. Images were viewed and goblet cell quantification per villi-crypt unit was performed in a blinded fashion using the Imagescope software.

### Visualization

scRNA-seq analysis data were visualized using Seurat (v.4.1.2) and R Studio (v.2022.07.1). Cartoons in Fig. 1a and Extended Data Fig. 6a were created using BioRender.com.

**Quantification and statistical analysis.** In all cases PCA was performed with R v.4.1.2. For cytokine data, log-transformed data were used for generation of PCA plots and for MDMR analysis. Biplots were constructed by projecting the weighted averages of each input feature (immune cell phenotypes, cytokine level, cellular composition and so on) along PC1, PC2 and/or PC3 derived from the biplot.pcoa function from the ape package, as done previously[19]. All data were assumed to be normally distributed but this was not formally tested except where mentioned. Effect size measures were determined using the MDMR v.0.5.1 (refs. 27,56,57) package in R and interactions were tested using the mixed effect analysis in the MDMR package. A significant effect size was said to be present if the $P$ value was less than or equal to 0.05 (*$P < 0.05$; **$P < 0.01$; ***$P < 0.001$; ****$P < 0.0001$).

MDMR analysis is a multivariate analog to the Fisher's $F$ ratio analysis that is rooted in traditional generalized linear models. This method provides an advantage over other standard multivariate procedures designed for use with small numbers of variables and other data reduction methods in that it combines the strengths of these two different approaches to test the association between a set of independent variables and high-dimensional data[27], such as those generated in this report. Further, use of MDMR provides us with an opportunity to be able to compare our current results with our previous report where we used a similar analytical toolset to assess the influence of genetic factors (mutant alleles in inflammatory bowel disease susceptibility genes) on immune variation[19]. The MDMR model calculates the effect size of each variable on the outcome measure to generate a pseudo $R^2$ value that quantifies the effect of the predictor variable on the dissociated outcome variable. Results in graphs and bar plots are displayed using Prism v.7 (GraphPad Software). Statistical analysis was performed using GraphPad Prism software (v.9). Right-skewed data were log- or square root-transformed. For analysis of the relationship between scRNA-seq cell composition and worm burden in Fig. 5, worm burden was modeled as following a negative binomial distribution. Predictor variables included in the regression model were mouse strain, mouse environment (that is, Lab or Rewilded) and loading on PC2 from analysis of scRNA-seq data. Then, 1,000 model-estimated coefficient values were plotted for each predictor variable. In some cases, data were analyzed by one-way analysis of variance (ANOVA) with Tukey's posttest when comparing three or more groups using GraphPad Prism software (v.9). Experimental group was considered statistically significant if the fixed effect $F$ test $P$ value was ≤0.05. Post hoc pairwise comparisons between experimental groups were made using Tukey's honest significant difference multiple-comparison test. A difference between experimental groups was taken to be significant if the $P$ value was less than or equal to 0.05 (*$P < 0.05$; **$P < 0.01$; ***$P < 0.001$; ****$P < 0.0001$).

### Inclusion and ethics statement

All collaborators of this study who fulfilled the criteria for authorship required by Nature Portfolio journals have been included as authors, as their participation was essential in the design and implementation of the study. This research does not result in stigmatization or discrimination of any of the participants. Local and regional research relevant to our study was considered in citations.

**Reporting summary**

Further information on research design is available in the Nature Portfolio Reporting Summary linked to this article.

## Data availability

Raw scRNA-seq data are deposited to the NCBI Sequence Archive (GSE236347). All other data needed to support the conclusions of the paper are present in the paper and associated Supplementary Data files. Further details regarding the dataset are available by request from P.L. Source data are provided with this paper.

## Code availability

All processing was performed in R and analysis scripts are available at https://github.com/oyeb2003/G-ERewilded-interaction-Project.

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

## Acknowledgements

We thank W. Craigens, C. Hansen and F. Rozenberg for invaluable assistance in the field and at Stony Ford. We also thank M. Zhao, K. Bledsoe and J. Randall for help with experiments in the lab and for help in maintaining the JoesFlow app software, respectively. We thank E. Tait Wojno for sharing some *Trichuris muris* parasite eggs with us. We thank M. Mahnaz and D. A. Alves of the Comparative Medicine Branch, NIAID/NIH, and S. Ganesan of the NIAID/NIH Research Technology Branch for help in imaging of slides. This research was supported by the Division of Intramural Research, National Institute of Allergy and Infectious Diseases, NIH, and K.C. is supported through NIH grant no. AI130945. A.E.D. acknowledges funding support from the National Science Foundation (award no. DGE-2039656). R.S.B. and A.L.G. acknowledge funding support from NJ ACTS (New Jersey Alliance for Clinical and Translational Science), which is supported in part by the New Jersey Health Foundation, Inc., and in part by a Clinical and Translational Science Award from the National Center for Advancing Translational Science of the National Institutes of Health, under award no. UL1TR003017. Y.-H.C. acknowledges funding from the Bernard Levine Postdoctoral Research Fellowship and the Charles H. Revson Senior Fellowship. S.B.K. and K.Z. acknowledge support from the National Cancer Institute (grant no. R01 CS271245). The content is solely the responsibility of the authors and does not represent the official views of the National Institutes of Health.

## Author contributions

O.O., A.E.D., K.C., A.L.G. and P.L. conceptualized the study. O.O., A.E.D., N.H., R.S.B., K.K., Y.-H.C., A.L.G., S.C.L., J.D. and O.M.-P. were responsible for the methodology. O.O., A.E.D., R.S.B., N.H., K.K., K.Z., Y.-H.C., A.M., S.C.L., O.M.-P., C.O.S.S., C.H. and P.L. performed investigations. O.O., N.H., A.E.D., A.L.G. and S.C. were responsible for data curation and analysis. O.O., A.E.D., N.H., A.L.G. and P.L. wrote the original draft of the paper. O.O., A.E.D., K.C., A.L.G. and P.L. reviewed and edited the paper. O.O., N.H., J.C., A.E.D. and A.L.G. performed visualizations. S.B.K., K.C., A.L.G. and P.L. supervised the project. K.C., A.L.G. and P.L. were responsible for funding acquisition.

## Competing interests

K.C. has received research funding from Pfizer, Takeda, Pacific Biosciences, Genentech and Abbvie, and P.L. has received research funding from Pfizer. K.C. has consulted for or received an honorarium from Puretech Health, Genentech and Abbvie. K.C. is an inventor on US patent 10,722,600 and provisional patents 62/935,035 and 63/157,225. S.B.K. acknowledges funding from Micreos and KymeraTx in the past 3 years. P.L. and O.O. are federal employees. The other authors declare no competing interests.

## Additional information

**Extended data** is available for this paper at https://doi.org/10.1038/s41590-024-01862-5.

**Correspondence and requests for materials** should be addressed to Oyebola Oyesola, Andrea L. Graham or P'ng Loke.

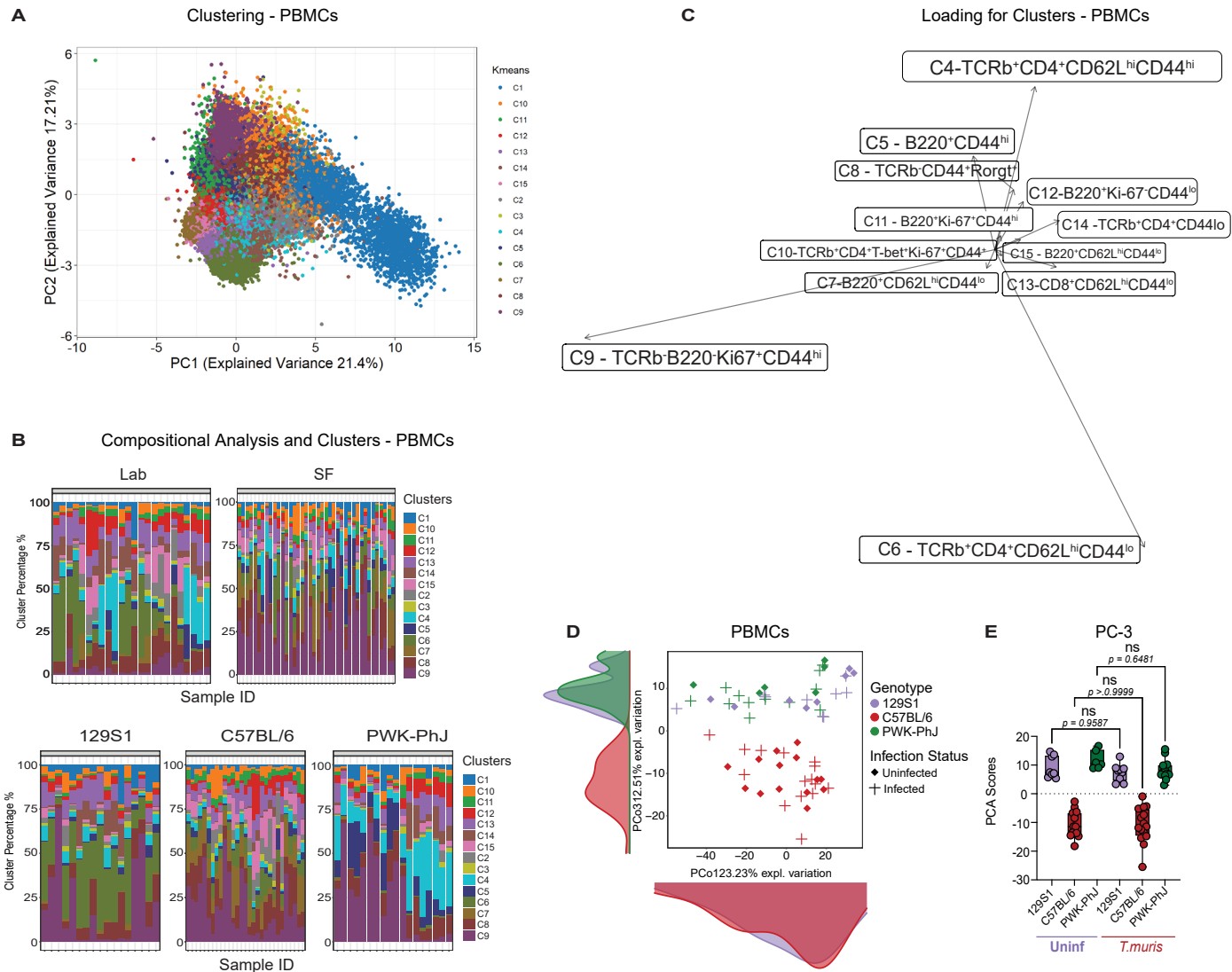

**Extended Data Fig. 1 | Unsupervised Clustering of murine PBMC cells. (A)** PCA Plot showing clusters generated following unsupervised clustering of PBMCs cells (n = 64; 17 129S1, 29 C57BL/6 and 18 PWK/PhJ mice) **(B)** Bar plot showing cluster percentage in different groups on a per mice basis. **(C)** the loading factors of immune clusters for PCA plot of PBMC cells showing PC1 and PC2 axis **(D)** PCA showing PC1 and PC3 axis of immune cell clusters identified by unsupervised clustering in the PBMCs cells and **(E)** Box plot showing variance on PC3 axis of

PCA plots in (D). The box plot center line represents median, the boundaries represent IQR with the whiskers representing the upper and lower quartiles ±1.5 Interquartile Range (IQR), all individual data points are shown (129S1 Uninf = 8, C57BL/6 = 14, PWK-PhJ Lab = 6, 129S1 RW = 9, C57BL/6 RW = 15, PWK RW = 12). Statistical significance in (E) was determined by one-way ANOVA test between different groups with Graph-Pad Software. *ns p > 0.05.*

**A** Pregated: Live, CD45⁺ TCRb⁺CD4⁺

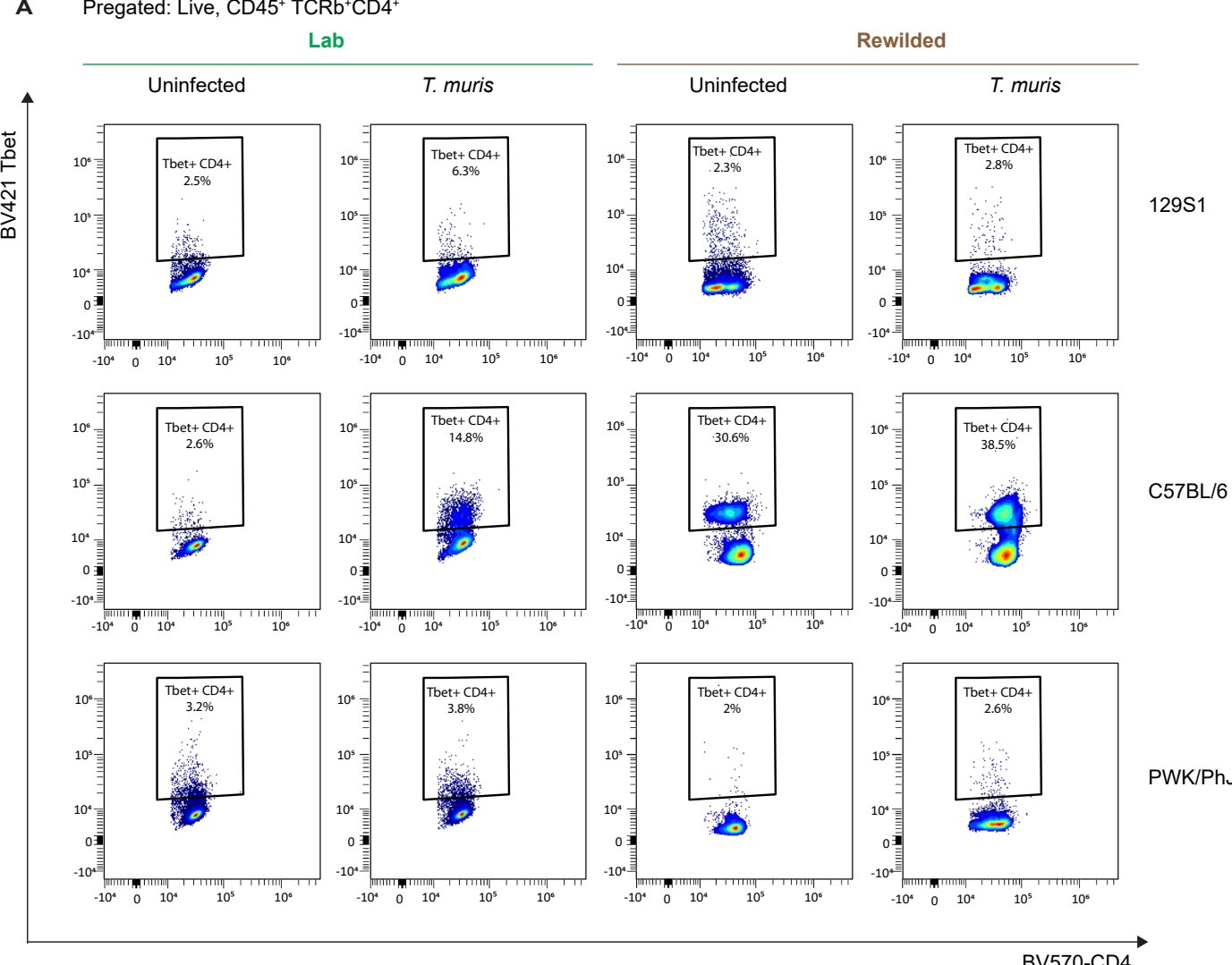

**Extended Data Fig. 2 | Gating strategy for identification of Tbet+ CD4+ T cells. (A)** Cells are pre-gated as singlets, Live+CD45+TCRb+CD4+ and representative contour lots showing percentage of Tbet⁺ CD4⁺ T cell of CD4+ cells in PBMCs are shown.

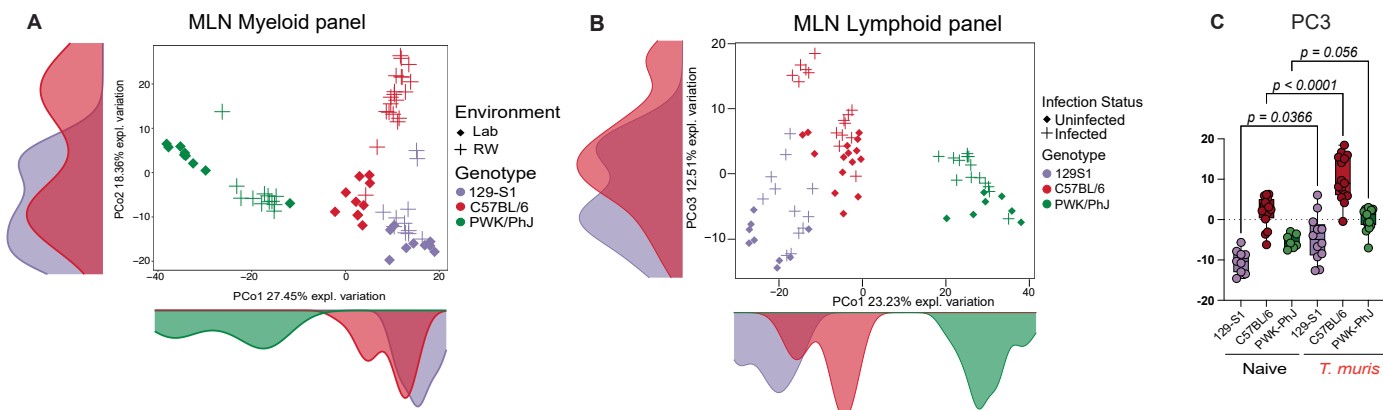

**Extended Data Fig. 3 | Genotype, Environment and Infection interactions determine immune composition in murine MLNs.** (**A**) PCA of immune cell clusters identified by unsupervised clustering in the MLN with the myeloid panel. (n = 73; 21 129S1, 30 C57BL/6 and 22 PWK/PhJ mice) (**B**) PCA of immune cell clusters identified by unsupervised clustering in the MLN (n = 73; 21 129S1, 30 C57BL/6 and 22 PWK/PhJ mice) with the lymphoid panel reflecting PC1 and PC3 with (**C**) Box plot showing the variance on PC3 axis of PCA plots. The box plot center line represents median, the boundaries represent IQR with the whiskers representing the upper and lower quartiles ±1.5 Interquartile Range (IQR), all individual data points are shown (129S1 Uninfected = 9, C57BL/6 Uninfected=16, PWK-PhJ Lab Uninfected = 7, 129S1 *T. muris* = 12, C57BL/6 *T. muris* = 15, PWK-PhJ *T. muris* = 15). Statistical significance was determined by one-way ANOVA one tailed test between different groups of interest with Graph-Pad Software (**C**).

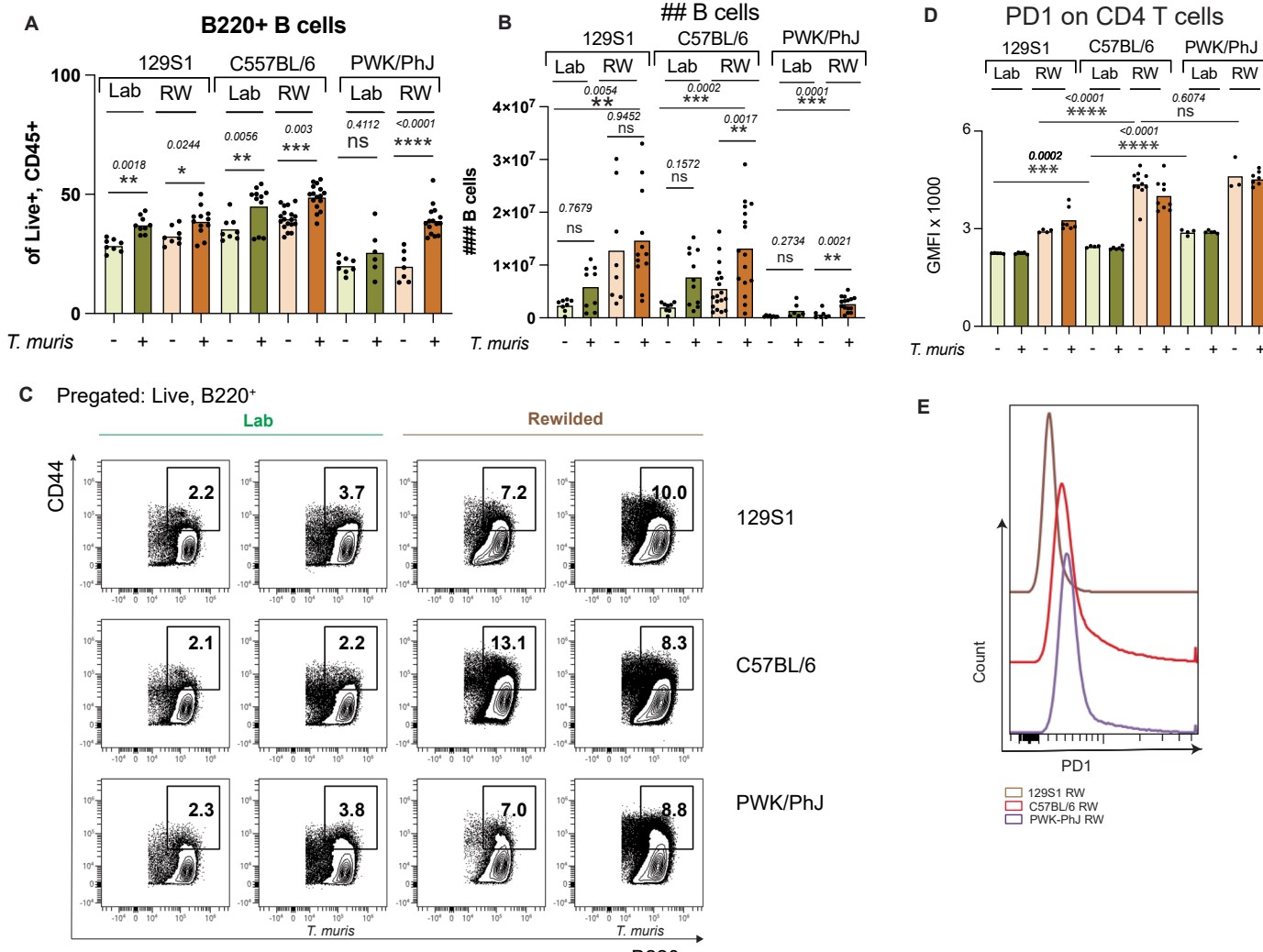

**Extended Data Fig. 4 | Genetics and Environmental effects on B cell and T cell phenotype.** Bar plots showing (**A**) frequencies and (**B**) numbers of B cells in the mesenteric lymph node, For (A) and (B) n = 126, 129S1 Lab Uninfected = 8, 129S1 Lab *T. muris* = 9, 129S1 RW Uninfected = 8, 129S1 RW *T. muris* = 12, C57BL/6 Lab Uninfected = 8, C57BL/6 Lab *T. muris* = 11, C57BL/6 RW Uninfected = 18, C57BL/6 RW *T. muris* = 16, PWK/PhJ Lab Uninfected = 8, PWK/PhJ Lab *T. muris* = 6, PWK/PhJ RW Uninfected = 7, PWK/PhJ RW *T. muris* = 15 over two experimental blocks; each dot represent individual mice. (**C**) Representative FACS plots showing percentage of CD44 high B cells in the mesenteric lymph node. (**D**) Bar plots depicting GMFI of PD1 on MLN CD4 + T cells,) n = 73, 129S1 Lab Uninfected = 5, 129S1 Lab *T. muris* = 5, 129S1 RW Uninfected = 4, 129S1 RW *T. muris* = 7, C57BL/6 Lab

Uninfected = 4, C57BL/6 Lab *T. muris* = 6, C57BL/6 RW Uninfected = 11, C57BL/6 RW *T. muris* = 9, PWK/PhJ Lab Uninfected = 4, PWK/PhJ Lab *T. muris* = 5, PWK/PhJ RW Uninfected = 3, PWK/PhJ RW *T. muris* = 10 over one experimental block, Block = 2 (**E**) Representative Histogram showing concatenated files from rewilded uninfected mice of each mice strain. Statistical significance was determined by one-way ANOVA one tailed test between different groups of interest with Graph-Pad Software (**A**), (**B**) and (**D**). For (**A**) and (**B**) direct comparison was done within each genotype; for (**D**), comparison was done between genotypes in the same environment with one-way ANOVA one tailed test. Data are displayed as mean ± SEM. *ns p > 0.05; *p < 0.05; **p < 0.01; ***p < 0.001; ****p < 0.0001.* RW, Rewilded.

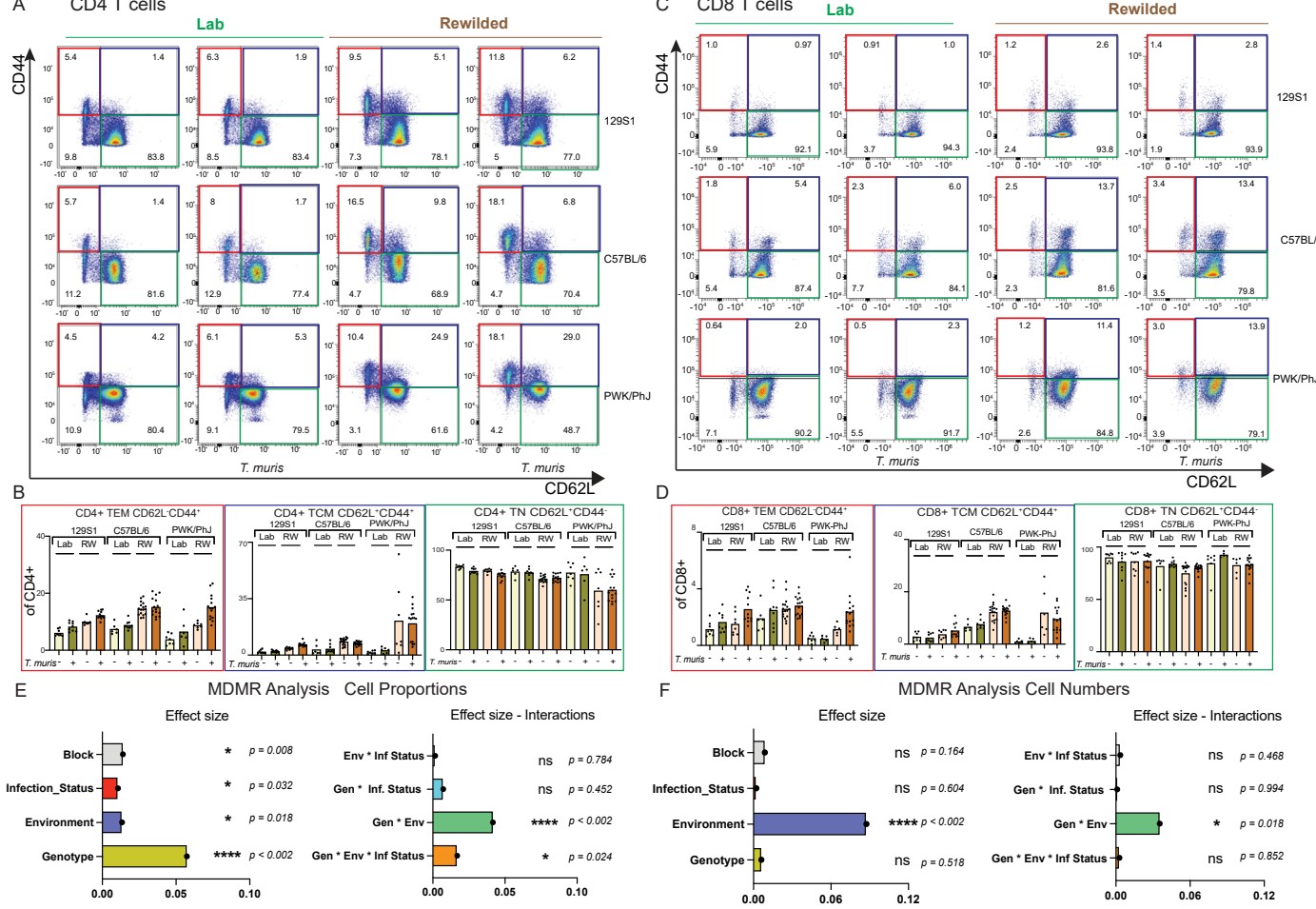

**Extended Data Fig. 5 | Genetics by Environment Interactions determine variation in T cell phenotype.** MLN cells from different strains of mice in the laboratory and rewilded conditions were harvested and the proportion of the Naïve and memory CD4 and CD8 T cell subset were determined by flow cytometric analysis. Representative flow cytometry plots and quantification of CD62L$^{hi}$CD44$^{lo}$, CD62L$^{hi}$CD44$^{hi}$, and CD62L$^{lo}$CD44$^{hi}$ CD4 T cells in CD4 T cells (**A**), (**B**) and the CD8 T cell (**C**) and (**D**) cellular population. For (**B**) and (**D**), n = 117; 129S1 Lab Uninfected = 8, 129S1 Lab *T. muris* = 9, 129S1 RW Uninfected = 8, 129S1 RW *T. muris* = 12, C57BL/6

Lab Uninfected = 6, C57BL/6 Lab *T. muris* = 9, C57BL/6 RW Uninfected = 17, C57BL/6 RW *T. muris* = 15, PWK/PhJ Lab Uninfected = 7, PWK/PhJ Lab *T. muris* = 5 PWK/PhJ RW Uninfected = 6, PWK/PhJ RW *T. muris* = 15 over two experimental blocks. Grubb's outlier test was done to remove outliers. Bar plots showing the pseudo $R^2$ measure of effect size of predictor variables and interactions as calculated by multivariate distance matrix regression analysis (MDMR) in based on (**E**) proportions and (**F**) cell numbers from the CD4 and CD8 T cell lymphoid cells CD4 T cell clusters harvested from the MLN. Raw data – Data File S6.

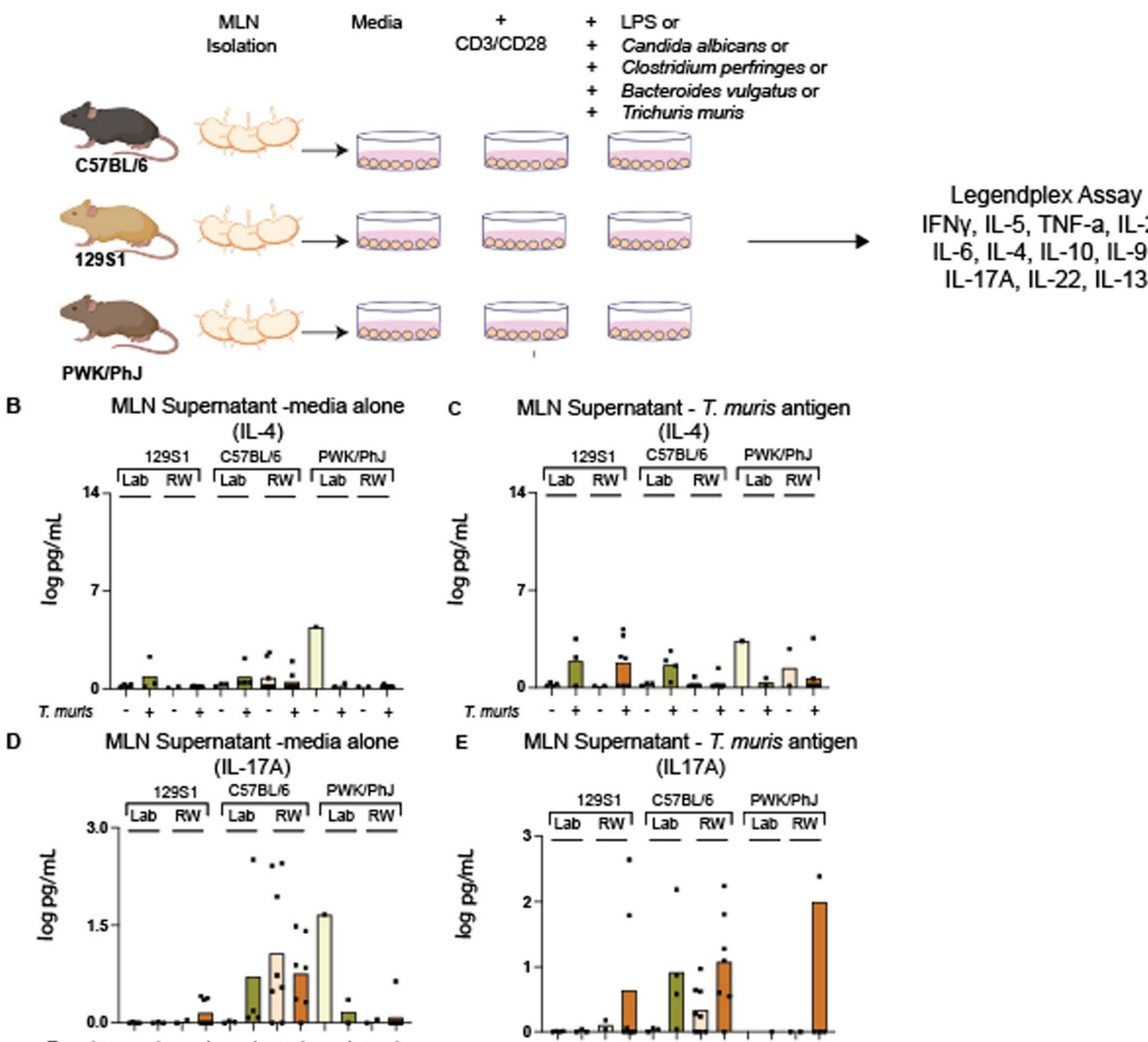

**Extended Data Fig. 6 | Schematic diagram of MLN *in-vitro* restimulation assay and cytokine levels with Legend plex cytokine assay.** (A) MLN cells from lab and rewilded of the different strains of mice were *ex-vivo* cultured with LPS, *C. albicans, C. perfringes, B. vulgatus, T. muris* or CD3/CD28 beads for 48 hours and supernatant was assayed for 11 cytokines IFN-γ, IL-5, TNF-α, IL-2, IL-6, IL-4, IL-10, IL-9, IL-17a, IL-22, IL-13. Bar plot showing transformed IL-4 and IL-17A cytokine levels in the supernatant for controls(**B, D**) as well as following stimulation with *T. muris* antigen (**C, D**), For (**B**), (**C**), (**D**) and (**E**), n = 50; 129S1 Lab Uninfected = 4, 129S1 Lab *T. muris* = 3, 129S1 RW Uninfected = 2, 129S1 RW *T. muris* = 7, C57BL/6 Lab Uninfected = 3, C57BL/6 Lab *T. muris* = 4, C57BL/6 RW Uninfected = 8, C57BL/6 RW *T. muris* = 7, PWK/PhJ Lab Uninfected = 1, PWK/PhJ Lab *T. muris* = 2, PWK/PhJ RW Uninfected = 2, PWK/PhJ RW *T. muris* = 7 over one experimental block, Block 2. Data were transformed to ensure normality before analysis. (Data File S8).

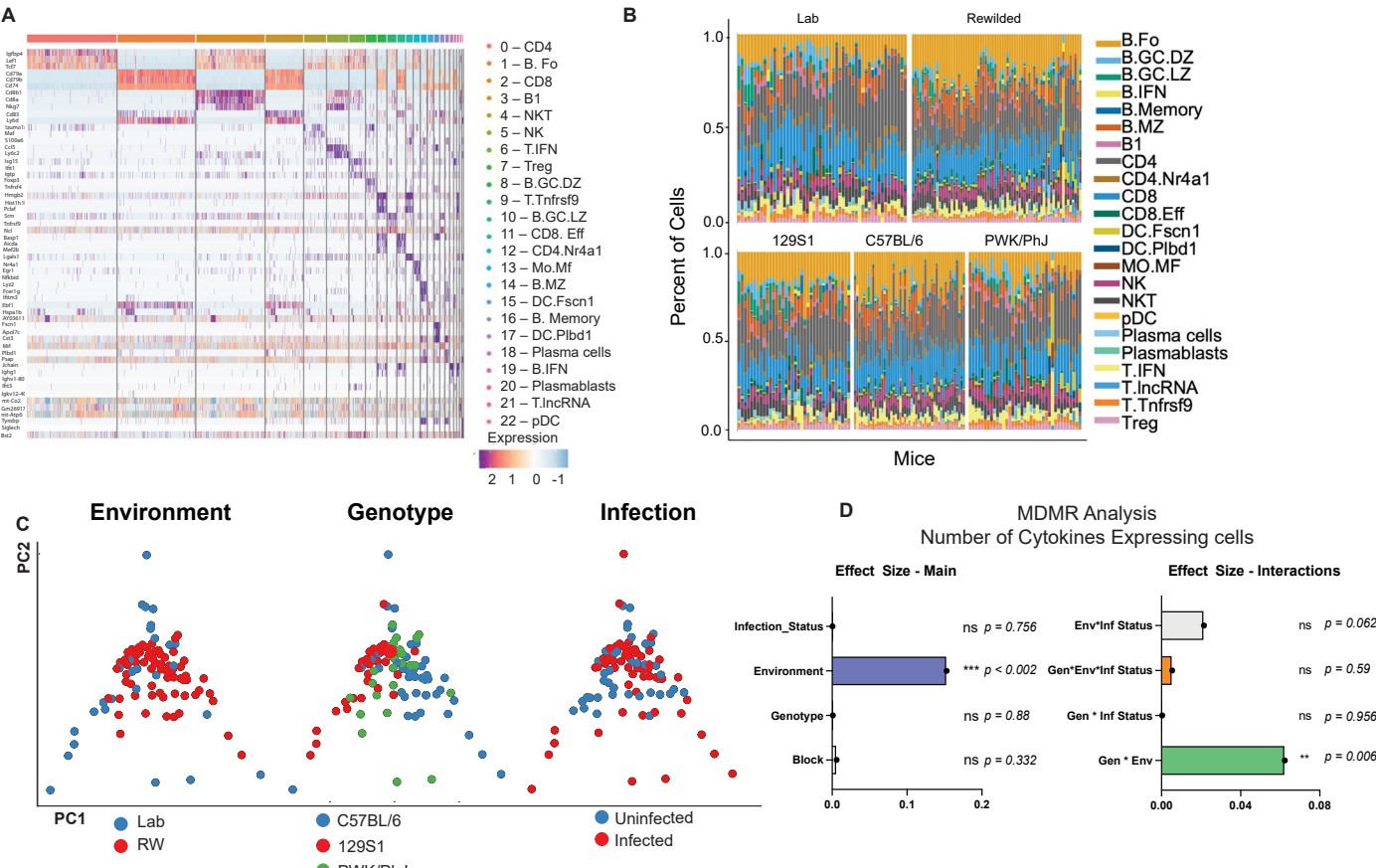

**Extended Data Fig. 7 | Single Cell Sequencing Analysis for assessing immune variation in cellular composition and cytokine profiles.** (**A**) Heat map depicting cluster defining genes used for cell type calling in Fig. 6a (**B**) Proportion of cell types identified in Fig. 3a on an individual mice basis (Block 1, n = 51, 17 129S1, 21 C57BL/6, 13 PWK/PhJ; Block 2, n = 71, 19 129S1, 28 C57BL/6 and 24 PWK/PhJ mice)

(**C**) PCA of proportion of cytokine expressing cells as determined by scRNAseq analysis (Data File S9). (**D**) Bar plots showing the pseudo $R^2$ measure of effect size of interactions as calculated by multivariate distance matrix regression analysis (MDMR) for number of cells with cytokine activity.

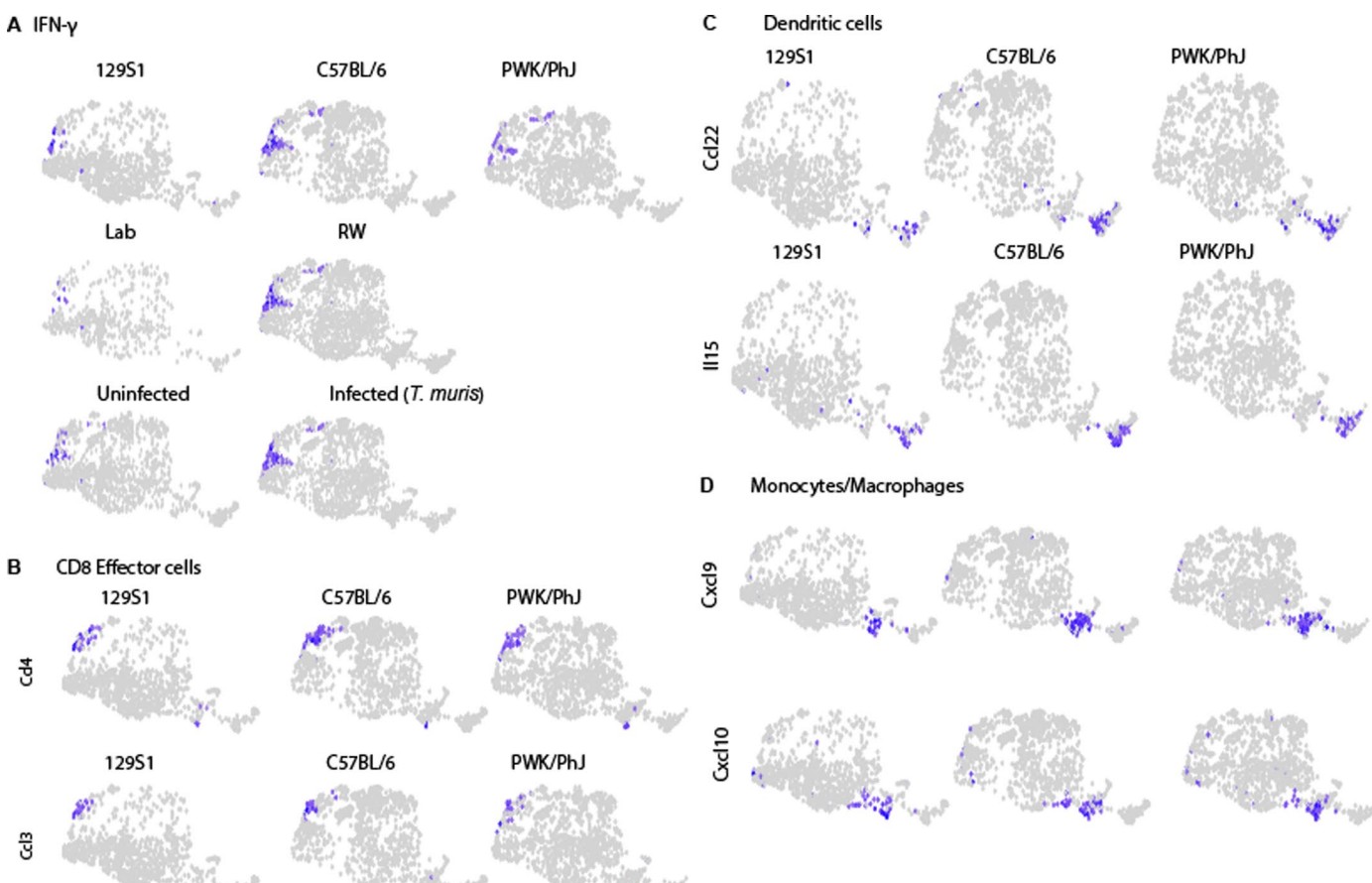

**Extended Data Fig. 8 | Genotype influences expression of genes with cytokine activity.** (**A**) Feature plot showing scRNA-seq IFN-γ transcripts based on mice strain, environment, and infection. Feature plot showing scRNA-seq cytokine transcripts based on mice genotype in different cellular clusters (**B**) CD8 Effector cells (**C**) Dendritic cells (**D**) Monocytes/Macrophages.

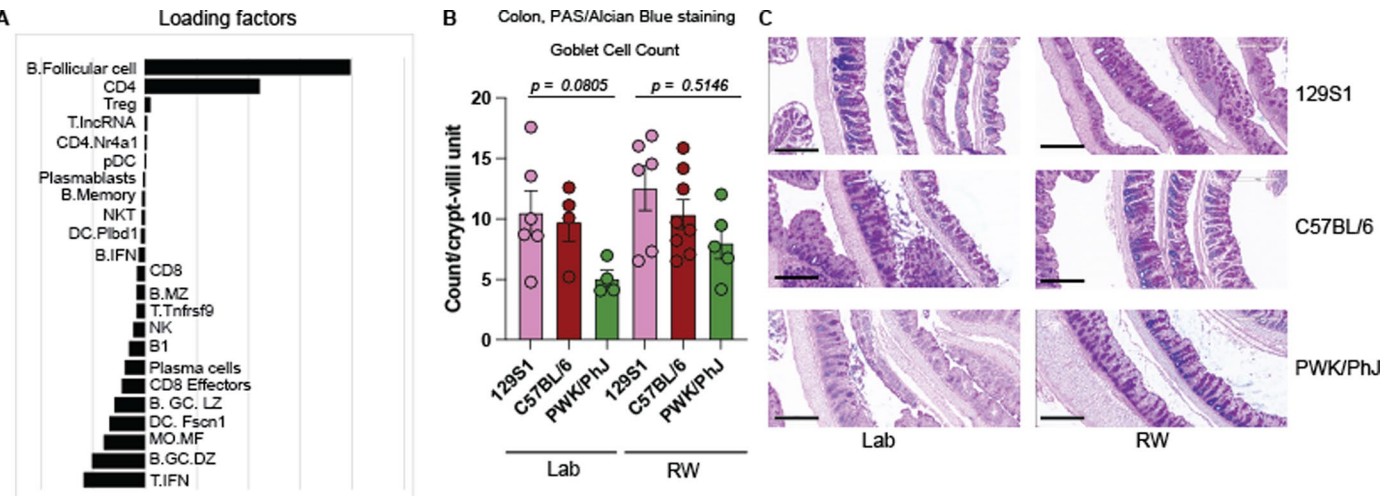

**Extended Data Fig. 9 | Genetics and Environmental factors predict outcomes during exposure with *T. muris* parasite. (A)** Loading Factors for PC2 of the scRNAseq dataset. **(B)** PAS/Alcian Blue-positive cells in the distal ileum were quantified from histological sections. n = 34, 129S1 Lab = 6, C57BL/6 = 4, PWK/PhJ = 4, 129S1 = 6, C57BL/6 = 8 and PWK/PhJ = 5. Data in

**(B)** is mean ± SEM. Statistical significance was determined by one-way ANOVA one tailed test between different groups with Graph-Pad Software **(C)** Representative images from PAS/Alcian Blue staining of histological sections from the distal ileum (200X), Scale bar = 300μM.

A. Representative Gating Strategy for PBMCs anaylsis

B. Representative Gating Strategy for Mesenteric Lymph node

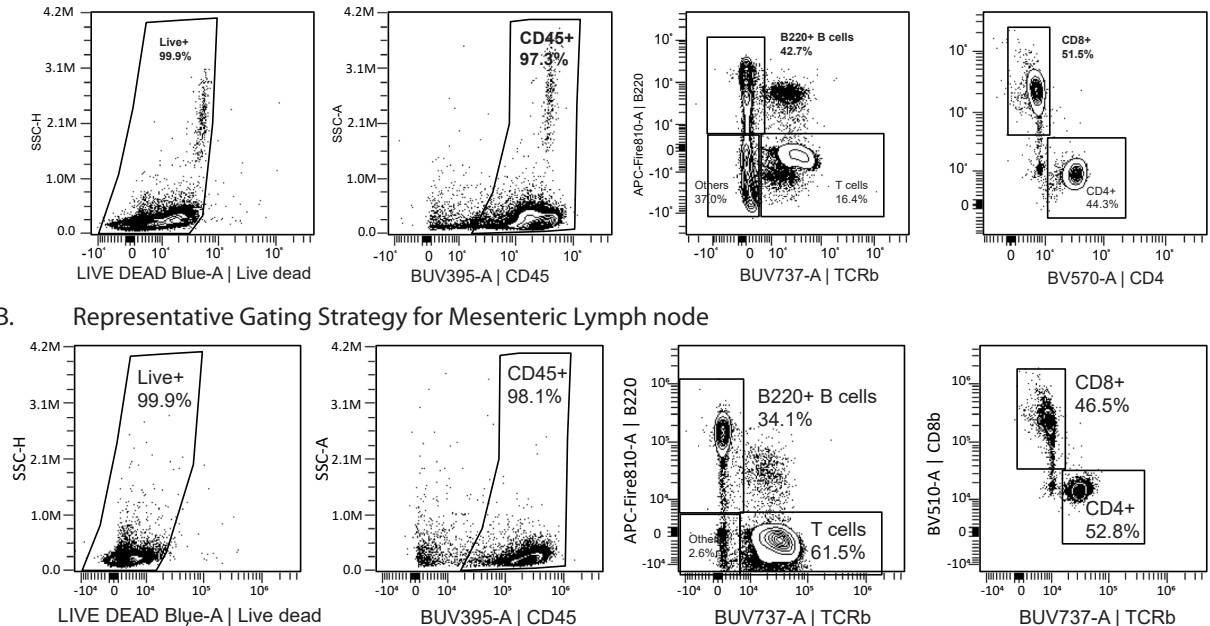

C. Representative Gating Strategy for Mesenteric Lymph node Intracellular Cytokine Staining

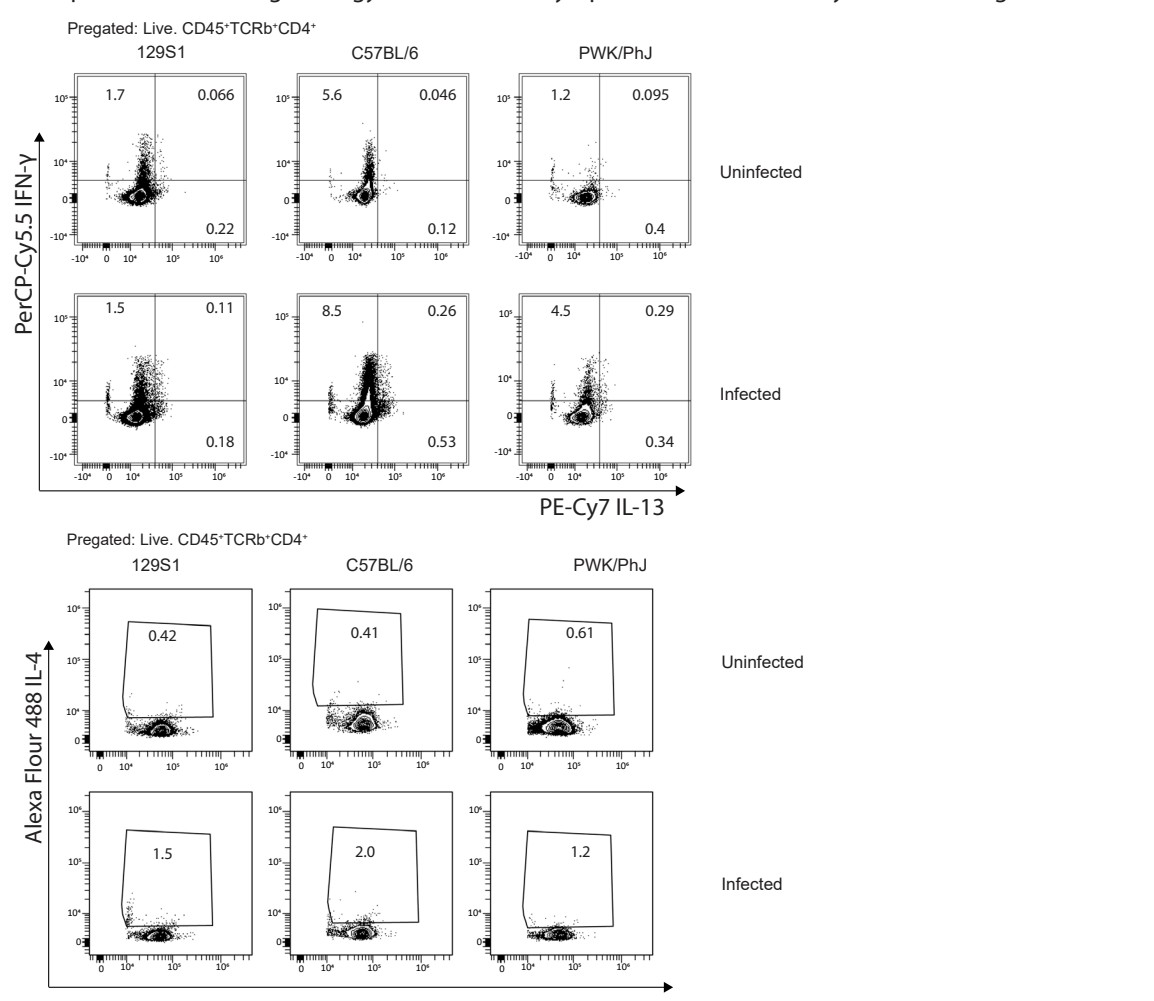

**Extended Data Fig. 10 | Gating Strategy for the different populations. (A)** Peripheral Blood Mononuclear cells (PBMCs), **(B)** Mesenteric Lymph node (MLN) and **(C)** MLN cells following PMA/ION stimulation.

# Reporting Summary

## Statistics

For all statistical analyses, confirm that the following items are present in the figure legend, table legend, main text, or Methods section.

| n/a | Confirmed | |
|---|---|---|
| ☐ | ☒ | The exact sample size (*n*) for each experimental group/condition, given as a discrete number and unit of measurement |
| ☐ | ☒ | A statement on whether measurements were taken from distinct samples or whether the same sample was measured repeatedly |
| ☐ | ☒ | The statistical test(s) used AND whether they are one- or two-sided *Only common tests should be described solely by name; describe more complex techniques in the Methods section.* |
| ☐ | ☒ | A description of all covariates tested |
| ☐ | ☒ | A description of any assumptions or corrections, such as tests of normality and adjustment for multiple comparisons |
| ☐ | ☒ | A full description of the statistical parameters including central tendency (e.g. means) or other basic estimates (e.g. regression coefficient) AND variation (e.g. standard deviation) or associated estimates of uncertainty (e.g. confidence intervals) |
| ☐ | ☒ | For null hypothesis testing, the test statistic (e.g. *F*, *t*, *r*) with confidence intervals, effect sizes, degrees of freedom and *P* value noted *Give P values as exact values whenever suitable.* |
| ☐ | ☒ | For Bayesian analysis, information on the choice of priors and Markov chain Monte Carlo settings |
| ☐ | ☒ | For hierarchical and complex designs, identification of the appropriate level for tests and full reporting of outcomes |
| ☐ | ☒ | Estimates of effect sizes (e.g. Cohen's *d*, Pearson's *r*), indicating how they were calculated |

*Our web collection on statistics for biologists contains articles on many of the points above.*

## Software and code

Policy information about availability of computer code

| | |
|---|---|
| Data collection | Blood samples (approx. 30-50𝜇L) were collected from all mice at endpoint via cheek bleeds using a Medipoint Golden Rod Lancet Blade 4MM (Medipoint NC9922361) into a 1.3mL heparin coated tube (Sarstedt Inc, NC9574345). Blood samples were analyzed using the element HT5, Veterinary Hematology Analyzer (Heska). <br><br> All Flow Cytometry samples were collected on an Aurora™ spectral cytometer (Cytek) and analyzed using the OMIQ software (https://www.omiq.ai/), data cleaning and scaling was done using algorithms like FlowCut56, 57 within the OMIQ software <br><br> Cytokine concentrations in plasma or MLN supernatants were measured also measured using the commercially available murine Th cytokine LEGENDplex assay (Biolegend) panel (Cat #741044) according to the manufacturer's instructions. <br><br> Single cell suspensions were obtained from MLN and pooled samples from each group were then loaded on a 10X Genomics platform, processed to generate cDNA and sequencing libraries were performed following 10X Genomics' protocol. Libraries were pooled and sequenced using Illumina NovaSeq SP 100 cycle as per 10X sequencing recommendations. |
| Data analysis | In all cases principal component analysis was performed with in R v4.1.2. For cytokine data, log transformed data were used for generation of PCA plots and for MDMR analysis. Biplots were constructed by projecting the weighted averages of each input feature (immune cell phenotypes, cytokine level, cellular composition etc.,) along PC1, PC2 and/or PC3 derived from the biplot.pcoa function from the ape package <br><br> The sequenced data were processed using Cell Ranger (version 6.0) to demultiplex the libraries. The reads were aligned to Mus musculus mm10 genomes to generate count tables that were further analyzed using Seurat (version 4.0). Sequencing data from the two blocks were |

integrated together prior to further downstream analysis.

Statistical analysis was performed using GraphPad Prism software (v9). Right-skewed data were log or square root transformed. For analysis of relationship between scRNAseq cell composition and worm burden in Fig. 5, worm burden was modeled as following a negative binomial distribution. Predictor variables included in the regression model were mouse strain, mouse environment (i.e., laboratory or rewilded), and loading on PC2 from analysis of scRNAseq data.

Code used for the analysis can be found here - https://github.com/oyeb2003/G-ERewilded-interaction-Project.

For manuscripts utilizing custom algorithms or software that are central to the research but not yet described in published literature, software must be made available to editors and reviewers. We strongly encourage code deposition in a community repository (e.g. GitHub). See the Nature Portfolio guidelines for submitting code & software for further information.

## Data

Policy information about availability of data

All manuscripts must include a data availability statement. This statement should provide the following information, where applicable:
- Accession codes, unique identifiers, or web links for publicly available datasets
- A description of any restrictions on data availability
- For clinical datasets or third party data, please ensure that the statement adheres to our policy

Raw scRNA sequencing data are deposited to the NCBI Sequence Archive (GSE236347). All other data needed to support the conclusions of the paper are present in the paper, Data Files or Extended Data Figures or can be accessed here -   doi:10.5061/dryad.ncjsxkt3w. If you need any more details about any of the data set, please contact lead corresponding author of the paper – Dr. P'ng Loke.  - png.loke@nih.gov

## Research involving human participants, their data, or biological material

Policy information about studies with human participants or human data. See also policy information about sex, gender (identity/presentation), and sexual orientation and race, ethnicity and racism.

| | |
|---|---|
| Reporting on sex and gender | N/A |
| Reporting on race, ethnicity, or other socially relevant groupings | N/A |
| Population characteristics | N/A |
| Recruitment | N/A |
| Ethics oversight | N/A |

Note that full information on the approval of the study protocol must also be provided in the manuscript.

# Field-specific reporting

Please select the one below that is the best fit for your research. If you are not sure, read the appropriate sections before making your selection.

☒ Life sciences ☐ Behavioural & social sciences ☐ Ecological, evolutionary & environmental sciences

For a reference copy of the document with all sections, see nature.com/documents/nr-reporting-summary-flat.pdf

# Life sciences study design

All studies must disclose on these points even when the disclosure is negative.

| | |
|---|---|
| Sample size | 25- 30 mice of mixed strains and genotype, 129S1/SV1mJ, C57BL/6J and PWK/PhJ female mice were used for these experiments. Sample size was determined by logistical constraints and not by power calculations. For rewilding, 12-18 female mice of the different strains (129S1/SV1mJ, C57BL/6J and PWK/PhJ) were housed in different wedges in the enclosure for 5 weeks. In summary, for Block 1 we rewilded, n = 42 mice (15 PWK/PhJ, 14 C57BL/6, 13 129S1); in Block 2 we rewilded, n = 47 mice (16 PWK/PhJ, 18 C57BL/6, 13 129S1) for a total of n =89 rewilded mice |
| Data exclusions | For all analysis, samples that failed pre-determined quality control such as flow cytometry staining errors, high cell death and/or are under limit of detection such as for the ELISA assay are not included in downstream statistical analyses. For the different measurements and assays, the same sample size was measured repeatedly except were mentioned in the Figure Legends. The number of mice per group, number of experimental replicates if any, and the statistical tests employed are reported in the figure legends. All data points represent biological replicates. |
| Replication | Experiments were repeated across two different experiment blocks with some immune phenotypes that replicates previously published rewilding studies. All data points represent biological replicates except where mentioned in the figure legend |

| Randomization | Mice were randomly assigned to either remain in the institutional vivarium (lab mice) or released into the outdoor enclosures (rewilded mice) previously described. For all experiments, only female mice were used to prevent unintended breeding in the rewilded environment and mice were between 8-12 weeks at point of blood draw following rewilding |
|---|---|
| Blinding | Investigators were blinded to the experimental groups the mice belong to at the time of the performing the different experimental assay but were unblinded at the point of statistical analysis and testing. |

# Reporting for specific materials, systems and methods

We require information from authors about some types of materials, experimental systems and methods used in many studies. Here, indicate whether each material, system or method listed is relevant to your study. If you are not sure if a list item applies to your research, read the appropriate section before selecting a response.

## Materials & experimental systems

| n/a | Involved in the study |
|---|---|
| ☐ | ☒ Antibodies |
| ☒ | ☐ Eukaryotic cell lines |
| ☒ | ☐ Palaeontology and archaeology |
| ☐ | ☒ Animals and other organisms |
| ☒ | ☐ Clinical data |
| ☒ | ☐ Dual use research of concern |
| ☒ | ☐ Plants |

## Methods

| n/a | Involved in the study |
|---|---|
| ☒ | ☐ ChIP-seq |
| ☐ | ☒ Flow cytometry |
| ☒ | ☐ MRI-based neuroimaging |

## Antibodies

| Antibodies used | Antibodies were used for the following antigens (clone) at the stated concentration from the stated vendor (catalog number)<br><br>Lymphoid Panel<br>Anti-mouse CD45 (30-F11) BUV395 BD Biosciences Cat#: 564279; 1:300<br>Rat Anti-mouse CD103 (M290) BUV496 BD Biosciences Cat#: 741083; 1:200<br>Rat Anti-mouse CD62L (MEL-14) BUV563 BD Biosciences Cat#: 741230; 1::100<br>Hamster Anti-mouse TCR-B chain (H57-597) BUV737 BD Biosciences Cat#: 612821; 1:200<br>Rat Anti-Mouse CD44 (IM7) BUV805 BD Biosciences Cat#: 741921; 1:100<br>Rat Anti-Mouse NKp46 CD335 (29A1.4), BV480 BD Biosciences Cat#: 746264 1:50<br>Anti-Mouse NK-1.1 (PK136), BUV661 BD Biosciences Cat#:74147; 1:200<br>Anti-mouse CD8β (H35-17.2) BV510 BD Biosciences Cat#: 740155; 1:200<br>Anti-mouse CD4 (RM4-5) BV570 BioLegend Cat#: 100542: 1:200<br>Hamster Anti-Mouse KLRG1 (2F1), BV750 BD Biosciences Cat#: 746972; 1:100<br>Anti-mouse TCRγδ (eBio-GL3) PECy5 Thermofisher Cat#: 15-5711-82 ; 1:50<br>Anti-mouse CD45R/B220 (RA3-6B2) APC/Fire810 BioLegend Cat#: 103278 1:200<br>Ant-mouse Tbet (4B10) BV421 Biolegend Cat#: 644816 1:50<br>Ki-67 Monoclonal Antibody (SolA15) AF532 Thermofisher Cat#: 58-5698-82; 1:100<br>Anti-mouse FoxP3 (FJK-16) AF700 Thermofisher Cat#: 56-5773-82 1:100<br>Anti-mouse RorgT (Q31-378) PE-CF594 BD Biosciences Cat#: 56284 1:50<br>Gata-3 Monoclonal Antibody (TWAJ), Alexa Fluor™ 488 Thermofisher Cat#: 53-9966-42; 1:50<br>Hamster Anti-Mouse CD69 (H1.2F3) BV650 BD Biosciences Cat#: 740460; 1:200<br>Anti-Mouse CD279 (PD-1) (29F.1A12) BV421 BioLegend Cat#:135218; 1:100<br>Anti-mouse IL-4 Clone 11B11 Alexa Flour 488 BD Biosciences Cat#: 557728; 1:50<br>Anti-mouse IFN-γ Antibody Clone XMG1.2 PerCP/Cy5.5 Biolegend Cat#: 505822; 1:100<br>Anti-mouse IL-13 Antibody Clone (eBio13A) PE-Cyanine 7 Thermofisher Cat#: 25-7133-82; 1:50<br><br>For Myeloid Panel.<br>Anti-mouse CD45 (30-F11) BUV395 BD Biosciences Cat#: 564279; 1:300<br>Rat Anti-mouse CD43 (S7) BUV563 BD Biosciences Cat#: 741238;1:100<br>Anti-mouse CD11b (M1/70) BUV615 BD Biosciences Cat#: 7511401:200<br>Anti-mouse TCRβ (H57-597) BUV661 BD Biosciences Cat#: 749914 1:200<br>Anti-mouse CD44 (IM7) BUV805 BD Biosciences Cat#: 741921 1:200<br>Anti-Mouse CD279 (PD-1) (29F.1A12) BV421 BioLegend Cat#: 135218; 1:150<br>MERTK Monoclonal Antibody (DS5MMER), Super Bright 436 Thermofisher Cat#: 62-5751-82 1:50<br>Anti-mouse CD49b (DX5), Pacific Blue Biolegend Cat #: 108918; 1:100<br>Rat Anti-mouse F4/80 Anti-Mouse (T45-2342) F4/80 BD Bioscience Cat #: 565635 1:100<br>Hamster Anti-Mouse CD27 (CD27) BD Bioscience Cat #: 563605; 1:100<br>Anti-mouse CD8α (5H10) Pacific Orange Thermofisher Cat#: MCD0830 1:200<br>Anti-mouse CD4 (RM4-5) Qdot800 Thermofisher Cat#: Q22165 RRID:AB_2556521 1:250<br>Anti-mouse CD64 (X54-5/7.1) PECy7 BioLegend Cat#: 139314  1:100<br>Anti1:-human CD278 (ICOS) (C398.4A) BV510 Biolegend Cat #: 313525 1:50<br>Anti-Mouse CD11c (N418) BV711 Biolegend Cat#: 117349 1:100<br>Hamster Anti-Mouse CD183 (CXCR3-173) BV750 BD Biosciences Cat#: 747298 1:!00 |
|---|---|

Anti-Mouse CX3CR1 (SA011F11) BV785 Biolegend Cat#: 149029 1:160
Rat Anti-mouse Siglec-F (E50-2440) BB515 BD Bioscience Cat#:564514  1:200
Anti-mouse TCRγδ (eBio-GL3) PECy5 Thermofisher Cat#: 15-5711-82 1:160
Anti-mouse CD301b/MGL2 (URA-1) PerCPCy5.5 BioLegend Cat#:146810 1:40
Anti-mouse CD273/PDL2 (B7-DC) PE BioLegend Cat#: 115565 1:20
Anti-mouse CD45R/B220 (RA3-6B2) APC/Fire810 BioLegend Cat#: 103278 1:200
Rat Anti-mouse CD62L (MEL-14) BUV563 BD Biosciences Cat#: 741230 1:200
Anti-mouse CD19 Antibody (6D5) Spark Blue 550 Biolegend Cat#: 115566 1:150
Mouse CXCR5 (614641) Alexa Fluor® 488 R&D Cat#: FAB6198G-100UG 1:100
Anti-mouse CD69 (H1.2F3) PE-Dazzle 594  Biolegend Cat#: 104535 1:100
Anti-mouse CD25 (PC61.5), PE-Cy5  Thermofisher Cat#: 15-0251-82 1:160
B7-2/CD86 (BU63) PE-Cy5.5 Novus Biological Cat#: NBP2-34569PECY55 1:100
Anti-mouse Tim-4 (RMT4-54) PECy7 Biolegend Cat#: 130010 1:50
Anti-mouse ST2 (DIH4) APC Biolegend Cat#: 146606 1:50
Anti-mouse CD206 (MMR) AF-647 Biolegend Cat#: 141712 1:100
Anti-mouse Ly-6C (HK1.4), PerCP Biolegend Cat#: 128028 1:!00
Anti-mouse Ly-6G (1A8), Spark NIR 685 Biolegend Cat#: 127666 1:200
Anti-mouse CD103, APC-R700 BD Biosciences Cat#: 565529 1:100
Anti-mouse KLRG1 (2F1/KLRG1), APC-Cyanine 7 Biolegend Cat#: 138436, 1:100

| | |
|---|---|
| Validation | The validation information of all commercially antibodies used in this study are available on the provider websites. |

# Animals and other research organisms

Policy information about <u>studies involving animals</u>; <u>ARRIVE guidelines</u> recommended for reporting animal research, and <u>Sex and Gender in Research</u>

| | |
|---|---|
| Laboratory animals | C57BL/6J, 129S1/SV1mJ and PWK/PhJ mice were purchased from The Jackson Laboratory (Bar Harbor, ME) and were housed under specific pathogen–free conditions with ad libitum access to food and water. All mouse lines were then bred onsite in a specific pathogen free (SPF) facility at National Institute of Health (NIH). The resulting littermates from the multiple breeding pairs were shipped to Princeton University where they were acclimated in a dedicated animal facility to temperatures and light cycles characteristics of summer in New Jersey (26℃ ± 1℃, and a 15-hour light/9-hour dark cycle). Following  this, mice were randomly assigned to either remain in the institutional vivarium (lab mice) or released into the outdoor enclosures (rewilded mice). |
| Wild animals | No wild animals were used in the study. |
| Reporting on sex | For all rewilding experiments, only female mice were used to prevent unintended breeding in the rewilded environment. However,some immune phenotypes  were repeated in laboratory conditions which included use of both male and female mice in the experiments. |
| Field-collected samples | No field collected samples were used in the study. |
| Ethics oversight | The protocols for mice breeding were approved by the NIAID Animal Care and Use Committee, Protocol Number, LPD 16E. The protocols for releasing the laboratory mice into the outdoor enclosure facility were approved by Princeton IACUC (protocol number 1982). |

Note that full information on the approval of the study protocol must also be provided in the manuscript.

# Plants

| | |
|---|---|
| Seed stocks | N/A |
| Novel plant genotypes | N/A |
| Authentication | N/A |

# Flow Cytometry

## Plots

Confirm that:

☒ The axis labels state the marker and fluorochrome used (e.g. CD4-FITC).

☒ The axis scales are clearly visible. Include numbers along axes only for bottom left plot of group (a 'group' is an analysis of identical markers).

☒ All plots are contour plots with outliers or pseudocolor plots.

☒ A numerical value for number of cells or percentage (with statistics) is provided.

## Methodology

**Sample preparation**

For PBMC
Heparinized whole blood collected via the cheek bleeds was mixed with blood collected via cardiac puncture method. The combined blood samples were spun for 10 minutes at 1500 rpm and plasma was collected and stored at -80oC for further cytokine analysis. The cellular component re-suspended in PBS next underwent a density gradient separation process using the Lymphocyte Separation Media (LSMTM MP Biomedicals, LLC) according to manufacturer's instruction. Isolated PBMCs were washed twice in PBS and then used for downstream spectral cytometric analysis

For MLN
Single cell suspension from the mesenteric lymph nodes (MLNs) were prepared by mashing the tissues individually through a 70μm cell strainer and washing with RPMI. Cells were then washed with RPMI supplemented with 10% FCS.

**Instrument**

All samples were collected and acquired on an Aurora™ spectral cytometer (Cytek) 5-Laser Configuration

**Software**

OMIQ software and unsupervised clustering using Joe's Flow software - https://github.com/niaid/JoesFlow were used. PCA anlaysis for flow analysis were analyzed using R studio (version 2022.07.1). Code available in Github - https://github.com/oyeb2003/G-ERewilded-interaction-Project.

**Cell population abundance**

All samples were collected on an Aurora™ spectral cytometer (Cytek) and analyzed using the OMIQ software (https://www.omiq.ai/), data cleaning and scaling was done using algorithms like FlowCut within the OMIQ software. Traditional gating strategy was used to determine cellular proportions and they were then multiplied by cell counts which was enumerated using the Element HT5, Veterinary Hematology Analyzer (Heska) for determination of counts for the different cell types

**Gating strategy**

Gating/Clustering was done through and unsupervised clustering method (https://github.com/niaid/JoesFlow). Doublets (identified by plotting the height versus the area for forward and side scatter) and dead cells (by plotting fixable viability dye against the SSC-Height). CD45+ cell were identified and samples were sub-sampled for unsupervised clustering.

In situations where traditional gating was done, an example flow plot depicting gating strategy was provided in Extended Data Set 10. B cells were considered single, Live, CD45+, B220+; CD4 T cells were identified as Live, CD45+TCRb+CD4+; CD8 T cells as Live, CD45+TCRb+CD8+,

☒ Tick this box to confirm that a figure exemplifying the gating strategy is provided in the Supplementary Information.

