## [Peer Review File · Nature Immunology]

Peer Review Information

Journal: Nature Immunology

Manuscript Title: Genetic and Environmental interactions contribute to immune variation in rewilded mice

Corresponding author name(s): Dr Oyebola Oyesola; Professor Andrea Graham; Dr Png Loke

Reviewer Comments & Decisions:

Decision Letter, initial version:
--

4th Aug 2023

Dear Bola,

Thank you for sending your point-by-point response to the referees' comments on your manuscript entitled "Genetic and Environmental interactions contribute to immune variation in rewilded mice". As mentioned previously, while they find your work of considerable potential interest, they have raised quite substantial concerns that must be addressed. In light of these comments, we cannot accept the current manuscript for publication, but would be very interested in considering a revised version that addresses these serious concerns.

We invite you to submit a substantially revised manuscript, however please bear in mind that we will be reluctant to approach the referees again in the absence of major revisions.

Specifically, the revision should include new experiments to address:

- (1) further interrogation of the available RNA-seq datasets
- (2) provide rationale for the choice of mouse strains and why only females were used in the study
- (3) discussion for why type 2 immune signatures were absent in the differential responses observed after *T. muris* infection and associated worm burden
- (4) Evaluate parameters of type 2 responses that might exist at baseline between the 3 mouse strains, both in under conventional laboratory housing and after rewilding conditions
- (5) Compare differences in immune responses to *T. muris* infection in laboratory mice at earlier time points than 21 day post-infection
{can this experiment be performed likewise with male mice?}

(6) Examine if differences exist for naive and memory T cell frequencies/marker expression in the rewilded mice

Please include the additional textual clarifications as indicated in your response letter.

When you revise your manuscript, please take into account all reviewer and editor comments, please highlight all changes in the manuscript text file in Microsoft Word format.

Please note, articles can be up to 4000-5000 words in length for the combined Introduction-Results-Discussion, 150 words for stand-alone Abstracts and have up to 8 main figures and 10 Extended Data figures.

* If you have not done so already please begin to revise your manuscript so that it conforms to our Article format instructions at <http://www.nature.com/ni/authors/index.html>. Refer also to any guidelines provided in this letter.

The Reporting Summary can be found here:

When submitting the revised version of your manuscript, please pay close attention to our [href="https://www.nature.com/nature-portfolio/editorial-policies/image-integrity">Digital Image Integrity Guidelines](https://www.nature.com/nature-portfolio/editorial-policies/image-integrity). and to the following points below:

[REDACTED]

If you wish to submit a suitably revised manuscript we would hope to receive it within 6 months. If you cannot send it within this time, please let us know. We will be happy to consider your revision so long as nothing similar has been accepted for publication at Nature Immunology or published elsewhere.

Nature Immunology is committed to improving transparency in authorship. As part of our efforts in this direction, we are now requesting that all authors identified as 'corresponding author' on published papers create and link their Open Researcher and Contributor Identifier (ORCID) with their account on the Manuscript Tracking System (MTS), prior to acceptance. ORCID helps the scientific community achieve unambiguous attribution of all scholarly contributions. You can create and link your ORCID from the home page of the MTS by clicking on 'Modify my Springer Nature account'. For more information please visit www.springernature.com/orcid.

Thank you for the opportunity to review your work.

Kind regards,

Laurie

Laurie A. Dempsey, Ph.D.
Senior Editor
Nature Immunology
l.dempsey@us.nature.com
ORCID: 0000-0002-3304-796X

Referee expertise:

Referee #1: helminth infections

Referee #2: helminth infections

Referee #3: helminth infections

Reviewers' Comments:

Reviewer #1:

Remarks to the Author:

This is an interesting body of work that seeks to quantify the contribution that genetics and

environment make to inter-individual murine immune response variation in the context of a bolus *T. muris* infection.

An impressive amount of data has been collected from the rewilded mice supporting the authors conclusions that the genetic differences in immune responses seen in SPF conditions are attenuated under rewilding conditions. Importantly, the variation in worm burden seen between the different strains of mice was greatly influenced by rewilding implying an environment-genetic influence. I cannot comment on the robustness of the authors statistical reasoning around the observation that PC2 outcompeting Gen*Env interactions (Figure 5D) implies that environment-dependent differences among genotypes in worm burdens may hinge on Th1 type immune factors. In conclusion the study reveals some interesting observations and approaches to the study of environment-gene reactions.

Reviewer #2:

Remarks to the Author:

The paper presents a comprehensive and in-depth study to discern the influence of host genotype, environment and genotype by environment interactions on phenotype using a well-established model of intestinal helminth infection, *Trichuris muris*. In order to tackle this a re-wilding approach has been employed, which has been used successfully by this team in several recent studies/publications. The advance in the present study makes use of three different inbred mouse strains and an extensive immunological and statistical analysis to a level which has not been utilised before in this arena to my knowledge. The experimental work has been carefully carried out using sound methodology and the data rigorously analysed. A number of key conclusions have been drawn, and for some, at one level, may not seem that surprising. However, the value of the work lies in the comprehensive and detailed analysis upon which the conclusions are based which adds considerable significance to the work. The paper has been clearly written and well- constructed. The figures are clear, if at times complex.

I have a number of comments which the authors might like to consider as follows. I think it would be useful to give a rationale for the choice of the three mouse strains used, e.g. I am assuming that the C57BL/6 was chosen because of its extensive use in immunological studies/transgenics and the PWK/PhJ because it was isolated from wild mice relatively (1970's) recently. There are some instances in the text where the analysis used is said to "determine" factors that are playing a role or the analysis "explains" an observation etc. I think, however, that the analysis identifies factors that may or are likely to determine or explain a particular role. I don't think they directly test the conclusions they make experimentally so the terms used should be a little more cautious.

The data from the PBMC phenotypic analysis is in depth and when compared to the mesenteric lymph node, the data confirm that the peripheral cellular response is less informative than the draining lymph node, which is not a new observation particularly with gut infections, although perhaps it has not been shown as elegantly before. Subsequent focus on the MLN provided further insight into the influence of genotype/environment and G*E. Whilst a IFN- γ signature was evident that correlated with the presence of parasites, are the authors surprised that no Type 2 immune signatures showed significant associations with absence of worms? The authors use a high dose infection which the literature from laboratory studies indicates drives Type 2 responses to *T. muris* in mice and most of the animals that were infected cleared most if not all their parasites. Have they any explanation for the lack of a type 2 signature? Is it related to the fact that the effector mechanisms that determine worm expulsion operate locally at the site of infection and that lymphocytes have to move from the MLN to

the intestine to mediate their effect, presumably switching on effector functions (e.g. cytokine production) at the intestinal site? The authors also state that there are differences in abundance/% cytokine producing cells between the different mouse strains used. I think that this needs to take into account the cell number of the MLN as there are marked differences between strains in this aspect too, which may change some of the conclusions drawn based upon abundance alone.

It is also noteworthy that the response in the MLN will be driven by the presence of the infection in the gut and there is significant variation in worm presence between strains on the day of sample collection. Day 21 post infection is a time point when, depending on mouse strain, worms will have been expelled or are in the process of being expelled. It may well be that some of the parasites observed in some of the mice are in the process of being expelled and would not survive until infection patency. There is little that can be done about this because of the practicalities of such an extensive and challenging study in a wild/natural environment but perhaps it should be borne in mind when forming some of the conclusions and perhaps discussed. In a similar vein, all mice used were female, which are known to be more resistant to *T. muris* than males from many laboratory studies in different inbred mouse strains and should be mentioned.

I am intrigued by the observations on different immune cell sub populations being primarily driven by different influences e.g. T cells by genetics and B cells by environment. Can the authors offer any insight into how this might be generated? The complexity of B and T cell responses, their interactions with each other and importance in protective immunity to

T. muris has been the subject of many studies. Conclusions from these would suggest that they are interdependent upon each other at multiple levels in the generation of the overall response so it is hard to understand how and why dominance of one immune factor or the other arises.

The section of the discussion dealing with comparison to human studies is interesting (line 371-414) and makes a compelling case for the use of the rewilding approach in mice in understanding human responses. It also considers the value of the mouse (both laboratory and rewilding) in gaining a greater depth and mechanistic understanding of phenotype and perhaps this conclusion could come across a little stronger in the text.

Lines 435-440. Based on the observation that the quantification of larval stages (worm burdens on day 21) the authors raise the possibility that the negative binomial distribution observed for worm burdens widely observed in natural populations infected by helminths may result as from immune consequences of G*E as opposed to differences in egg exposure, i.e. infection events. I am not sure if they are implying that the current view of negative binomial distribution in helminth infections is driven solely by egg exposure? I didn't think this was the case e.g. exposure to eggs and subsequent infection will drive the host immune response and this would be integral to the parasite distribution seen in natural populations that consider individuals of different ages and exposure/immune status. Perhaps, I am over-interpreting what the authors are saying; maybe the wording could be changed a little to clarify?

Overall, this is a fascinating study that applies an extensive and rigorous immunological analysis to a re-wilding study of infection unlike anything carried out previously. Clear conclusions have been generated which gives granularity to investigation of the relative effects of genetics and environment and their interplay in immune responses to parasitic worm infection.

Reviewer #3:

Remarks to the Author:

This is a conceptually exciting paper describing the analysis of 'rewilded' mice, comparing laboratory vivarium conditions to outdoor housing in a controlled pen. The authors have elaborated on their earlier studies by comparing 3 genetically distinct strains of mice in parallel, and conducted in depth analyses of plasma, blood and lymph node cell populations, and cytokine responsiveness in each setting. Superimposed on this, the effects of intestinal helminth infection with *Trichuris muris* are also analyzed.

The data are processed by sophisticated statistical techniques which distinguish the effects of mouse genetics, housing environment, and infection state. In Figure 1 they show that peripheral blood cellularity is primarily determined by genotype, albeit environment impacts on eosinophilia which is elevated in all strains when rewilded. Relevant to a later point, they also show increased Tbet expression in infected mice in a strain and environment-dependent manner. Figure 2 shows parallel data for mesenteric lymph nodes, and Figure 3 for cytokine levels in plasma, and for cytokine responses from cultured lymphocytes; again genotype is dominant but infection alters the profile substantially. In Figure 4 the authors perform single cell sequencing analyses on MLNs from the different groups, and here find that each parameter (infection, environment, genetics) independently exerts a significant effect. Finally, in Figure 5 the authors present the effects of the different conditions on *T. muris* infection, which as expected is strain-dependent but also is found to be amplified in the rewilded environment.

Overall, the approach is highly novel and the authors have collected an extraordinarily deep set of data. They use sophisticated statistical techniques, which is certainly a strength of the manuscript, but to some extent a weakness: relatively few individual parameters are shown to be influenced by each of the conditions so that it is not possible to infer mechanistic pathways that would explain the findings. The one exception is that IFN γ levels are raised in the rewilded environment, and are higher in the C57BL/6 mice; the rewilded C57BL/6 are also those with the highest worm levels. However, a positive correlation between Type 1 responses (also seen in the Tbet measures shown in Figure 1F) and *T. muris*, is only to be expected given the known dependence of immunity to this parasite on Type 2 responses.

The authors take a purely data-driven approach, deriving relationships which emerge from their statistical analysis. There is relatively little follow-through, for example if IFN γ is highly influenced by genes and environment, how does that compare with IL-4 and IL-17; and as CD44 expression is increased by rewilding, do the authors have data on other memory B cell and T cell markers? These analyses may add depth and finer detail to the manuscript.

While there is an absence of mechanistic relationships, the authors do highlight which outcomes are most determined by genetic factors versus environmental, together with the very interesting examples of those governed by gene-by-environment interactions.

Specific Comments

1. I found the Abstract a little underwhelming and overgeneralized ("T cell markers are driven more by genetics, B cell markers more by environment"). Is it possible to parameterize some of the statements here?

2. Introduction, lines 38-40. One could add epigenetic and maternal factors into the mix here.
3. Lines 99-101: what was the rationale for selecting these particular strains of mice, and what has previously been established about their comparative immune phenotypes?
4. Lines 128-138. It is not clear from the text whether MDMR is the only applicable approach to analyze the data, or what specific advantages it offers over alternatives.
5. In Figure 1 C (PBMC), there is a positive correlation between PC1 and PC2 in rewilded, but not lab housed, mice. Does this signify that a factor (ie a cell type) in PC1 correlates with a different factor/cell type in PC2, in a relationship not seen in vivarium mice (or with MLN as there is no similar correlation in Figure 2B).
6. In Figure 1G, eosinophilia is seen in rewilded but not lab housed mice, yet in Figure 1H, the highest eosinophilia is seen in lab mice given T muris; hence the effect of infection is lost in the wild. While this validates the authors' contention of context dependent interactions, how can this be mechanistically explained?
7. In Figure 2D, the MLN cell numbers are sharply increased in the rewilded mice, presumably due to environmentally-acquired intestinal infections or microbionts. In the analyses shown elsewhere in the Figure, are cell numbers or percentages used? The results may be very different in each case.
8. Why in Figure 4B does each individual parameter have a significant effect size, but not when in combination?
9. Figure 4 G-I present proportions of all cells expressing any cytokine; this would seem very broad-brush and difficult to interpret. Can a more granular analysis be presented?
10. In Figure 4 the Legend for (C) and (D) is erroneous and (E) is missing.
11. Figure 5 D indicates factors that correlate positively or negatively with worm burden; these could be strengthened by showing direct correlations with statistical support. What does "Lab Intercept" in panel C mean?

Author Rebuttal to Initial comments

See inserted PDF

We appreciate the constructive comments and suggestions from the reviewers and the editors, which have substantially improved our revised manuscript. Additional experiments and analyses have been performed and we have now added a substantial amount of new information with additional clarification for the study. These changes have been highlighted in the revised manuscript. We believe we have addressed most of the reviewers' concerns.

Our new experiments and analyses further indicate that increased susceptibility to worm burden is associated with a stronger Type 1 immune response which might have an influence on the net balance of the Th1 vs Th2 immune response, instead of differences in the strength of Type 2 responses. However, it is important to point out that the main conclusion of this study centers around the contribution of interactions between genetic and environmental factors to immune variation.

Editor Specific Response

Specifically, the revision should include new experiments to address:

(1) further interrogation of the available RNA-seq datasets

In response to the comment by Reviewer 3 that a more granular analysis be presented for Figure 4G – I, we have (1) generated a heatmap which illustrates genes that are being expressed by the different cell clusters shown in Figure 4A. Results are shown in Fig S6A. Additionally, we have (2) highlighted expression of IFN γ between the genotypes, environment and infection status in Fig. S7A and (3) highlighted expression of cytokines that differ between genotypes in different cell types, now shown in Fig. S7B, S7C and S7D respectively.

(2) provide rationale for the choice of mouse strains and why only females were used in the study

We have now explained in the introduction the selection of genetically diverse founder strains of the Collaborative Cross that are non-albino (as required by the Princeton IACUC for outdoor experiments), which may enable complex trait analysis in the future. Females were used because several male mice were able to bypass barrier separation in previous experiments, leading to unwanted complexities of female pregnancies.

(3) discussion for why type 2 immune signatures were absent in the differential responses observed after *T. muris* infection and associated worm burden

We have now discussed possible explanations for why differential Type 2 signatures were not detected. Notably, altered balance between Type 1 and Type 2 responses may be more important in determining infection susceptibility. These results can also be explained by earlier studies describing the key role Type 1 cytokines, especially, IFN γ can play either directly or indirectly in the suppression of protective responses during *Trichuris muris* infection (Else, Finkelman et al. 1994, Artis, Potten et al. 1999, Cliffe, Humphreys et al. 2005, Hurst and Else 2013).

(4) Evaluate parameters of type 2 responses that might exist at baseline between the 3 mouse strains, both in under conventional laboratory housing and after rewilding conditions.

For additional parameters of the effector Type 2 response, we have now examined goblet cells in the intestinal tissues of the rewilded mice (SFig 8) and have also performed additional experiments at baseline between the 3 mouse strains under laboratory conditions (Fig 5). These results indicate that differences in Type 1 responses varies more significantly between the different mouse genotypes, being higher in the C57BL/6 mice, which may have a net effect on increased worm burden.

(5) Compare differences in immune responses to *T. muris* infection in laboratory mice at earlier time points than 21 day post-infection {can this experiment be performed likewise with male mice?}

We have performed additional experiments at day 14 post infection in both male and female laboratory mice and found that the biggest difference between the genotypes is in the Type 1 response as described above. Results are shown in Figure 5D and Fig. S8.

(6) Examine if differences exist for naive and memory T cell frequencies/marker expression in the rewilded mice

We have performed additional analyses of naïve and memory T cell frequencies and examined different memory marker expression. These new results are shown in SFig 3G-H and SFig 4. We show that there are differences in the memory T cell frequencies in the rewilded mice as previously reported (Lin, Devlin et al. 2020, Yeung, Chen et al. 2020) and that a Genotype by Environment (Gen* Env) interaction explains significant variation in the naïve and memory T cell frequencies and numbers.

Reviewer 1:

(Remarks to the Author)

This is an interesting body of work that seeks to quantify the contribution that genetics and environment make to inter-individual murine immune response variation in the context of a bolus *T. muris* infection.

An impressive amount of data has been collected from the rewilded mice supporting the authors conclusions that the genetic differences in immune responses seen in SPF conditions are attenuated under under rewinding conditions. Importantly, the variation in worm burden seen between the different strains of mice was greatly influenced by rewinding implying an environment-genetic influence.

I cannot comment on the robustness of the authors statistical reasoning around the observation that PC2 outcompeting Gen*Env interactions (Figure 5D) implies that environment-dependent differences among genotypes in worm burdens may hinge on Th1 type immune factors.

In conclusion the study reveals some interesting observations and approaches to the study of environment-gene reactions.

Response to Reviewer

We thank the Reviewer for the enthusiasm for our approach and the conclusions that we have drawn. The Reviewer also identifies the important relationship between Type 1 immune factors and worm burden outcomes as being an important observation that we should further substantiate.

We performed additional experiments in laboratory mice for the different genotypes and observed a higher Th1 responses in the C57BL/6 strain and not differences in Th2 response is associated with differences in worm burden (Fig 5D). We have now added more text in the discussion and more data in our results section including further analysis of our scRNAseq results (SFig7A) to support this observation.

Figure 5D

SFig7A

IFN- γ

129S1

C57BL/6

PWK/PhJ

Lab

RW

Uninfected

Infected (*T. muris*)

Reviewer 2:

(Remarks to the Author)

The paper presents a comprehensive and in-depth study to discern the influence of host genotype, environment and genotype by environment interactions on phenotype using a well-established model of intestinal helminth infection, *Trichuris muris*. In order to tackle this a re-wilding approach has been employed, which has been used successfully by this team in several recent studies/publications. The advance in the present study makes use of three different inbred mouse strains and an extensive immunological and statistical analysis to a level which has not been utilised before in this arena to my knowledge. The experimental work has been carefully carried out using sound methodology and the data rigorously analysed. A number of key conclusions have been drawn, and for some, at one level, may not seem that surprising. However, the value of the work lies in the comprehensive and detailed analysis upon which the conclusions are based which adds considerable significance to the work. The paper has been clearly written and well-constructed. The figures are clear, if at times complex.

We thank the Reviewer for their very positive remarks and comment on our manuscript. We especially appreciate the comment that we were able to convey what is quite a complex study in a clear and well-constructed manner.

I have a number of comments which the authors might like to consider as follows. I think it would be useful to give a rationale for the choice of the three mouse strains used, e.g. I am assuming that the C57BL/6 was chosen because of its extensive use in immunological studies/transgenics and the PWK/PhJ because it was isolated from wild mice relatively (1970's) recently.

In response to Editor Comment 2 above and to both Reviewer 2 and Reviewer 3, we have now described in the Introduction (Page 4 Line 77-82) the rationale for the choice of C57BL/6, 129S1 and PWK/PhJ strains. The following changes have been made to the text.

“We selected these genetically diverse non-albino founder strains of the Collaborative Cross of mice (27) to enable more complex trait analysis in the future. C57BL/6J and 129S1/SvImJ are representative of classical laboratory inbred strains, whereas PWK/PhJ is a representative of a wild-derived strain (27). Furthermore, we rewilded only female mice to prevent unintended breeding from male mice breaching the barriers in the rewilding environment.”

There are some instances in the text where the analysis used is said to “determine” factors that are playing a role or the analysis “explains” an observation etc. I think, however, that the analysis identifies factors that may or are likely to determine or explain a particular role. I don't think they directly test the conclusions they make experimentally so the terms used should be a little more cautious.

We thank the reviewer for keenly reading our manuscript and for these helpful suggestions to the text. We have now made necessary amends to the text as follows.

Page 7, Line 144 – “determine” changed to “identify”
Page 11, Line 277 - from “explained” to “contributed to”;
Page 11, Line 282 – from “explained” to “contributed to”;
Page 13, Line 328 – from “explained” to “contributed”;
Page 13: Line 339 – from “explaining” to “contributing to”;
Page 15, Line 394-395 – from – Variation in *Trichuris muris* worm burden is determined by genetic, environmental, and immunological factors to Genetic, environmental and immunological factors contribute to variation in *Trichuris muris* worm burden.

The data from the PBMC phenotypic analysis is in depth and when compared to the mesenteric lymph node, the data confirm that the peripheral cellular response is less informative than the draining lymph node, which is not a new observation particularly with gut infections, although perhaps it has not been shown as elegantly before. Subsequent focus on the MLN provided further insight into the influence of genotype/environment and G*E. Whilst a IFN-g signature was evident that correlated with the presence of parasites, are the authors surprised that no Type 2 immune signatures showed significant associations with absence of worms? The authors use a high dose infection which the literature from laboratory studies indicates drives Type 2 responses to *T. muris* in mice and most of the animals that were infected cleared most if not all their parasites. Have they any explanation for the lack of a type 2 signature? Is it related to the fact that the effector mechanisms that determine worm expulsion operate locally at the site of infection and that lymphocytes have to move from the MLN to the intestine to mediate their effect, presumably switching on effector functions (e.g. cytokine production) at the intestinal site?

Yes, we agree with the Reviewer, and we were also surprised by the lack of a stronger Type 2 signature and speculated that the Type 2 signature could be more likely to be detected in the intestinal tissues. Unfortunately, it was not possible to isolate the cells from the intestinal tissue for detailed characterization for all the mice in this experiment for practical and logistical reasons associated with working in the rewilding system.

However, it is important to note that in a previous experiment (Leung, Budischak et al. 2018), whereby we could examine intestinal T cell responses because of the smaller numbers of only C57BL/6 mice involved, we still did not find a clear role of Type 2 immune response in explaining differences in worm burden in rewilded mice.

For instance, there was no significant difference in worm burdens between C57BL/6 and STAT6KO mice following rewilding, which suggested that in the rewilded environment, the Stat6 signaling pathway plays a more limited role in the resulting differences in worm burden. Furthermore, in that experiment (Leung, Budischak et al. 2018), we found that there was a bigger difference in IFN γ +CD4+ T cells than in IL13+ CD4 T cell response, at day 21 post infection between laboratory and rewilded mice following *Trichuris muris* infection in the intestinal lamina propria (tissue site) (Leung, Budischak et al. 2018).

Together with the results from this current experiment, we now favor the hypothesis that a stronger Type 1 response in the rewilded C57BL/6 mice is associated with increased susceptibility, and this might influence the balance with Type 2 immunity.

Furthermore, we analyzed goblet cell numbers as a representation of effector Type 2 responses from stored representative distal large intestinal tissues. Although the PWK-PhJ strain trended towards having fewer goblet cells in the lab, overall, there is no significant genotype associated differences in goblet cell numbers, as well as no significant environment related difference in goblet cell counts in the vivarium and following rewilding in all the three different strains of mice (Fig S8B).

Overall, our new results support data from our previous report (Leung, Budischak et al. 2018), that differences in Type 2 responses might not explain the higher worm burdens in rewilded C57BL/6 strain of mice. Instead, differences in Type 1 response associated with increased production of IFN gamma in C57BL/6 mice, support our hypothesis that an increase Type 1 signature which might consequently have a net effect on the balance of the Th1 and Th2 response might explain the increased worm burden observed in the C57BL/6 strain of mice, and indeed in rewilded mice of all strains. These results can also be explained by earlier studies describing the key role Th1 cytokines, especially, IFN gamma can play either directly or indirectly in the suppression of protective responses during *Trichuris muris* infection (Else, Finkelman et al. 1994, Artis, Potten et al. 1999, Cliffe, Humphreys et al. 2005, Hurst and Else 2013). We have now expanded our discussion of the role of Type 1 cytokines in responses to *Trichuris muris* infection in the discussion section.

Changes to Results: Page 16, Line 425-433

*“Quantification of goblet cell count as a measure of effector Type 2 response (45-47) showed no significant difference between lab or rewilded environment in different strains of mice prior to exposure to *T. muris* (SFig. 8B). Furthermore, flow cytometric analysis of cytokine production in the three different strains of mice at day 14 post challenge with *T. muris* eggs from MLN cells following in vitro stimulation with cell activation cocktail (phorbol 12-myristate-13-acetate, ionomycin and protein transport inhibitor (Brefeldin A)) showed that increased levels of IFN γ , a Type 1 cytokine in the CD4⁺ T cells rather than differences in production of Type 2 cytokines, IL-4 and IL-13 (Fig. 5D) might explain variation in worm burden and prevalence, especially in the C57BL/6 strain of mice”.*

Modifications to the discussion is as below: Page 20-21, Line 570-588

*“Furthermore, we did not observe a strong Type 2 signature in the mesenteric lymph nodes nor in goblet cell responses (Figure 5D, SFig 5B), despite Type 2 responses being well documented to play significant roles in worm expulsion following infection with a high bolus of *Trichuris muris* eggs (43, 79, 80). Instead, we observed differences in Type 1 responses between the different strains of mice (Figure 1F, S2, 3C, 3D, S7A, 5D, S7A) that might have a net effect on the balance between Type 1 and Type 2 responses to be associated with differences in worm burden. Notably, our previous rewilding study found no difference in *Trichuris muris* worm burden between rewilded C57BL/6 and STAT6KO mice (28) and IL-13⁺ CD4⁺ T cells were also not significantly different between laboratory and rewilded mice following *Trichuris muris* infection in the intestinal lamina propria at day 21 post infection (28). Instead, an increase Type 1 signature is associated with increased worm burden in the C57BL/6 strain of mice. This report fits into earlier studies describing the key role Type 1 cytokines, especially IFN-gamma, can play in suppressing the protective response either directly or indirectly during *Trichuris muris* infection (80-83). Although, rewilded mice of all genotypes had higher eosinophilia at two weeks post-release (Fig 1G), likely due to increased granulopoiesis from exposure to wild fungi (52), there was no additional expansion following exposure to *T. muris* in the rewilded mice as opposed to laboratory mice (Fig 1H). While the mechanism for this observation is unclear and may be related to interactions with hematopoiesis, it further validates the concept of context dependent interactions on the immune phenotype and functional outcomes”.*

The authors also state that there are differences in abundance/% cytokine producing cells between the different mouse strains used. I think that this needs to take into account the cell number of the MLN as there are marked differences between strains in this aspect too, which may change some of the conclusions drawn based upon abundance alone.

The Reviewer makes another excellent point that we should consider both the total cell number in the MLNs, as well as the % of cytokine producing cells. We have now shown the results for the number of cells with cytokine activity cells based on the scRNA seq in Figure 4J.

Furthermore, our MDMR analysis of the cell numbers rather than cell proportions demonstrate a key effect of Gen* Env interaction in determining variation in number of cells expressing cytokine activity genes with a residual fixed effect of the environment when assessed by the scRNAseq. We have now shown this result in an updated Fig 4 and SFig6. These results are now described in Page 14, Line 374-379 and below

*“MDMR analysis of the total number of cytokine producing cells in the MLN (Data File S11) shows that Gen*Env interaction contribute the most to variation in number of cytokine producing cells, with a residual fixed effect of environment contributing to the rest of the variation (Fig S6E). This can be observed by plotting the number of cells with cytokine activity for the individual mice with the Genotype of the mice determining the magnitude of the effect of the environment on immune variation (Fig. 4I, right panel)”.*

It is also noteworthy that the response in the MLN will be driven by the presence of the infection in the gut and there is significant variation in worm presence between strains on the day of sample collection. Day 21 post infection is a time point when, depending

on mouse strain, worms will have been expelled or are in the process of being expelled. It may well be that some of the parasites observed in some of the mice are in the process of being expelled and would not survive until infection patency. There is little that can be done about this because of the practicalities of such an extensive and challenging study in a wild/natural environment but perhaps it should be borne in mind when forming some of the conclusions and perhaps discussed

The Reviewer makes an excellent point that response in the MLN can be driven by the presence of infection in the gut and that significant variation in worm presence between strains on the day of sample collection could influence variation in MLN response. Therefore, we have examined the immune response at Day 14 post infection (Fig 5D), when the worms have not yet been cleared and would be similar between the different strains of mice. Here, we observed that there are significant differences in Type 1 immune responses even at day 14 post infection when worms have not yet been cleared. Furthermore, the reviewer correctly notes that it is not possible for us to determine at the Day 21 time point if the *Trichuris* worms are in the process of being expelled and would not have survived until infection patency or have already been expelled. This has now been mentioned in the discussion (Page 20, Line 548-554).

*“It is interesting to note that the quantification of larval stages in the process of being expelled also follow a negative binomial distribution. This analysis is limited by the difficulty in distinguishing between worms that are in the process of being expelled and the ones that will survive till patency. Further analysis of worm burden at a later timepoint may determine whether variation in worm burden persist till day 35 post infection. This suggests that the immune consequences of Gen*Env interactions might contribute to the negative binomial distribution in worm burdens in natural population.”*

In a similar vein, all mice used were female, which are known are known to be more resistant to *T. muris* than males from many laboratory studies in different inbred mouse strains and should be mentioned.

The use of female mice is for purely logistical reasons. We have now highlighted this point in the Introduction (Page 15-16, Line 419-423) and discussed the concept of female mice being more resistance to *T. muris* than male mice in other laboratory studies (Page 18, Line 497-499). Additionally, in our new experiments analyzing the immune response at an earlier time point, we have used both males and female mice and found the effect of sex to be negligible in immune responses at day 14 post infection (Fig. 5D).

“Furthermore, we rewilded only female mice to prevent unintended breeding from male mice breaching the barriers in the rewilding environment”.

*“Furthermore, future work on contributions of Genotype and Environment to immune heterogeneity in both sexes would be of value, especially in the context of known sex related susceptibility during *Trichuris* infections (43, 59, 60)”.*

I am intrigued by the observations on different immune cell sub populations being primarily driven by different influences e.g. T cells by genetics and B cells by environment. Can the authors offer any insight into how this might be generated? The complexity of B and T cell responses, their interactions with each other and importance in protective immunity to *T. muris* has been the subject of many studies. Conclusions from these would suggest that they are interdependent upon each other at multiple levels in the generation of the overall response so it is hard to understand how and why dominance of one immune factor or the other arises.

We thank the reviewer for an opportunity to discuss this further. While this observation is poorly understood, we have now provided additional context and added the following to our discussion (Page 18-19, Line 507-517).

The differential contribution of environment versus genetics on the B cell versus T cell subset is interesting. The Human Functional Genomics Project (HFGP) had observed that the percentage of immune variation that is explained by genetics for T cells (~30%) is greater than for B cells (<~18%), which also suggest that T cells could be influenced more by genetics than B cells(1). This may reflect underlying genetic differences in the MHC or HLA molecules that present antigen (61-63). Perhaps B cell responses are more influenced by the environment because their populations are driven more by microbial exposure as a result of direct activation through the BCR. However, the complexity and interdependence of B and T cell responses to infections makes it difficult to fully understand the differential contribution of environment and genetics to these adaptive immune cell populations.

The section of the discussion dealing with comparison to human studies is interesting (line 371-414) and makes a compelling case for the use of the rewilding approach in mice in understanding human responses. It also considers the value of the mouse (both laboratory and rewilding) in gaining a greater depth and mechanistic understanding of phenotype and perhaps this conclusion could come across a little stronger in the text.

We thank the reviewer for their point of making the conclusions about use of environmental models such as the rewilding approach for studies of human disease stronger. We also agree that it validates the value of mouse experiments in gaining greater mechanistic understanding of immune phenotypes that are not possible in human studies. We have now edited this text thereby making our conclusions stronger and more encompassing (Page 19, Line 518-524). The paragraph reads as follows:

“Our analysis and experimental design with the rewilding model present us an opportunity to assess the contribution of Gen Env interactions to various immune traits and in different tissue sites, which is not feasible in human studies. Together, this provides a bridge towards a better representation of variation in human immune phenotypes compared to the use of specific pathogen free mouse models(25). Further, such mouse experiments can provide greater mechanistic understanding of immune phenotypes that are not possible in human studies, whereby access to tissue samples or during infectious challenges might be limited”.*

Lines 435-440. Based on the observation that the quantification of larval stages (worm burdens on day 21) the authors raise the possibility that the negative binomial

distribution observed for worm burdens widely observed in natural populations infected by helminths may result as from immune consequences of G*E as opposed to differences in egg exposure, i.e., infection events. I am not sure if they are implying that the current view of negative binomial distribution in helminth infections is driven solely by egg exposure? I didn't think this was the case e.g., exposure to eggs and subsequent infection will drive the host immune response and this would be integral to the parasite distribution seen in natural populations that consider individuals of different ages and exposure/immune status. Perhaps, I am over-interpreting what the authors are saying; maybe the wording could be changed a little to clarify?

We appreciate the reviewer for the suggestion about clarification of our text around the binomial distribution of worm burdens. We have now edited the text (Page 20, Line 547-563) to make our discussions surrounding this observation clearer as well as included other factors such as the age of the host not previously considered before into the text as potential sources of variation in worm count in natural populations. The paragraph reads as follows:

*“Variation in adult worm burden of soil transmitted helminths between individuals typically follow a negative binomial distribution in humans (64) and wild animals (65). It is interesting to note that the quantification of larval stages in the process of being expelled also follow a negative binomial distribution. This analysis is limited by the difficulty in distinguishing between worms that are in the process of being expelled and the ones that will survive till patency. Further analysis of worm burden at a later timepoint may determine whether variation in worm burden persist till day 35 post infection. This suggests that the immune consequences of Gen*Env interactions might contribute to the negative binomial distribution in worm burdens in natural population. Hence, in addition to studies demonstrating that host genetics (66-68) and environmental factors (28) could influence susceptibility to helminth infection, we show here that interactions between genetics and environmental factors could also influence helminth infection outcomes. Variation in outcome to helminth infection is complex and maybe influenced by various other factors such as heterogeneity in parasite genetic factors (67-69), parasite dose and frequency of parasite exposure (70-73), host microbiome factors (74-76), age(77, 78) as well as an individual's infection history(4, 73). Importantly, natural helminth infection typically occurs from trickle infection of multiple small doses of egg exposures; therefore, the experimental system here of a high dose *T. muris* infection may not be representative of a real-world exposure.”*

Response to Reviewer 3
Reviewer #3

(Remarks to the Author)

This is a conceptually exciting paper describing the analysis of 'rewilded' mice, comparing laboratory vivarium conditions to outdoor housing in a controlled pen. The authors have elaborated on their earlier studies by comparing 3 genetically distinct strains of mice in parallel, and conducted in depth analyses of plasma, blood and lymph node cell populations, and cytokine responsiveness in each setting. Superimposed on this, the effects of intestinal helminth infection with *Trichuris muris* are also analyzed.

The data are processed by sophisticated statistical techniques which distinguish the effects of mouse genetics, housing environment, and infection state. In Figure 1 they show that peripheral blood cellularity is primarily determined by genotype, albeit environment impacts on eosinophilia which is elevated in all strains when rewilded. Relevant to a later point, they also show increased Tbet expression in infected mice in a strain and environment-dependent manner. Figure 2 shows parallel data for mesenteric lymph nodes, and Figure 3 for cytokine levels in plasma, and for cytokine responses from cultured lymphocytes; again genotype is dominant but infection alters the profile substantially. In Figure 4 the authors perform single cell sequencing analyses on MLNs from the different groups, and here find that each parameter (infection, environment, genetics) independently exerts a significant effect. Finally, in Figure 5 the authors present the effects of the different conditions on *T. muris* infection, which as expected is strain-dependent but also is found to be amplified in the rewilded environment.

Overall, the approach is highly novel and the authors have collected an extraordinarily deep set of data. They use sophisticated statistical techniques, which is certainly a strength of the manuscript, but to some extent a weakness: relatively few individual parameters are shown to be influenced by each of the conditions so that it is not possible to infer mechanistic pathways that would explain the findings. The one exception is that IFN γ levels are raised in the rewilded environment, and are higher in the C57BL/6 mice; the rewilded C57BL/6 are also those with the highest worm levels. However, a positive correlation between Type 1 responses (also seen in the Tbet measures shown in Figure 1F) and *T. muris*, is only to be expected given the known dependence of immunity to this parasite on Type 2 responses.

We thank the reviewer for their comment and enthusiastic remarks about our paper. We note that we have now provided additional data to support the hypothesis that increased susceptibility is associated with a stronger Type 1 response, more so than differences in the Type 2 response. Additionally, we have now addressed the other points raised by the reviewer in the comments below.

The authors take a purely data-driven approach, deriving relationships which emerge from their statistical analysis. There is relatively little follow-through, for example if IFN γ is highly influenced by genes and environment, how does that compare with IL-4 and IL-17; and as CD44 expression is increased by rewilding, do the authors have data on

We have examined additional T cell activation markers and have quantified the expression of PD1 on CD4 T cell. Our results show that there are some strain dependent differences in expression of PD1 marker in the lab controls and which becomes activated following rewilding with some reduction of genotype related differences following rewilding. However, some of the genotype related difference in expression of PD1 between the 129S1 and the PWK-PhJ strain of mice persist even following rewilding (SFig3G-H). This result is now described in Page 10, Line 246-248 and shown below.

“A similar genotype effect in the CD4 T cell compartment was also observed for other memory markers such as PD1, where expression of PD1 was also highest in the rewilded C57BL/6 and PWK/PhJ strain of mice (Fig. S3G and Fig, S3H).”

Additionally, we provide more detailed analyses of TCM, TEM and Naïve T cell populations in the MLN, and similar to our previous results, we demonstrate that again the genotype by environment interactions has the biggest effect on determining variation in the different population of T cells with a residual fixed effect of genotype on the proportions of cell and of environment on the number of cells. These results are now reflected in SFig 4 and described in Page 10, Line 249-261 as well as in the text below

*“In addition, when we examined the proportions of the different naïve and memory T cell populations as we had previously examined in rewilded C57BL/6 mice (27, 31), we found that the central memory CD8 and CD4 T cells expand following rewilding in the PWK/PhJ and C57BL/6 strain of mice (Fig. S4A, B, C, D) as previous noted. An MDMR analysis of the different CD4 and CD8 T cell pools (Data File S6) reveals again that a Gen*Env interaction contributes to variation in the different T cell pools with a residual fixed main Genotype effect when proportions of cells were used for the analysis (Fig S4E). However, when analyzing absolute cell numbers, we found that Gen*Env similarly contributed to the variation in the T cell population. In this analysis, a residual main effect of Environment was the predominant factor explaining the remaining*

variance, in contrast to the Genotype effect that was prominent when we assessed cellular proportions (Fig. S4F). This differential main fixed effect of Genotype and Environment to cell proportion versus cell numbers might be due to the huge effect of environmentally-acquired intestinal microbiota, metaorganisms or food antigens on the MLN numbers(40, 41)”.

We also provided a more encompassing discussion of variation in cellular proportion and cell number in Page 17, Line 463-474

“Despite the complexities, we do also identify traits for which main (i.e., non-interaction) effects are dominant. For instance, heterogeneity in proportions of cell with cytokine responses show a stronger influence of genetics, consistent with human studies (48), while heterogeneity in absolute cell numbers shows a stronger influence of the environment, consistent with results from previous microbial exposure mice studies(49). Furthermore, the Human Functional Genomics Project produced reported results, that variation in proportion of T cell phenotypes are relatively more influenced by genetics, while B cell phenotypes are relatively more influenced by non-heritable environmental factors (1). However, variation in absolute numbers of T cell phenotypes is still relatively more influenced by the environment, suggesting that an individual’s microbial history and experience plays a large role in determining absolute cell numbers present in different tissue sites through effects of microbial exposure on immune cell differentiation, activation, proliferation and maturation(31, 50-52).”

Specific Comments

1. I found the Abstract a little underwhelming and overgeneralized (“T cell markers are driven more by genetics, B cell markers more by environment”). Is it possible to parameterize some of the statements here?

We apologize to the reviewer for this point, but we were unable to make the abstract more specific due to the current restrictions on word count. We would be happy to expand this further with some editorial approval.

We have now made the following change to the abstract, Page 2, Line 36-37:

“Notably, genetic differences under laboratory conditions can be decreased following rewilding, and expression of CD44 on T cells is driven more by genetics, whereas on B cells more by the environment.”

2. Introduction, lines 38-40. One could add epigenetic and maternal factors into the mix here.

We thank the reviewer for raising this point. We have now edited the text and the references to include these factors (Page 3, Line 42-44).

“An individual’s immune phenotype is shaped by some combination of genetic, maternal, epigenetic factors, and nonheritable influences such as environmental exposure (including infection history and the microbiome) (1-13)”

3. Lines 99-101: what was the rationale for selecting these strains of mice, and what has previously been established about their comparative immune phenotypes?

We thank the reviewer for raising this point which is similar to one raised by Reviewer 2 above. We have now addressed this point in Page 4 Line 77-81.

“We selected these genetically diverse non-albino founder strains of the Collaborative Cross of mice (29) to enable more complex trait analysis in the future. C57BL/6J and 129S1/SvImJ are representative of classical laboratory inbred strains, whereas PWK/PhJ is a representative of a wild-derived strain (29).”

4. Lines 128-138. It is not clear from the text whether MDMR is the only applicable approach to analyze the data, or what specific advantages it offers over alternatives.

We thank the reviewer this point, we have now expanded this text in Page 7, Lines 145-154 to offer an explanation to the advantages of MDMR in comparison to other alternative approaches and our interest in the use of this approach in comparison to others.

MDMR analysis is a multivariate analogue to the Fisher’s F ratio analysis that is rooted in traditional generalized linear models. This method provides an advantage over other standard

multivariate procedures that are designed for use with small number of variables and other data reduction methods, in that it combines the strengths of these two different approaches to test the association between a set of independent variables and high-dimensional data (36) such as those generated in this report. Further, use of MDMR provides us an opportunity to be able to compare our current results with our previous report where we used similar analytical tool set to assess the influence of genetic factors (mutant alleles in inflammatory bowel disease susceptibility genes) on immune variation(27).

5. In Figure 1 C (PBMC), there is a positive correlation between PC1 and PC2 in rewilded, but not lab housed, mice. Does this signify that a factor (i.e. a cell type) in PC1 correlates with a different factor/cell type in PC2, in a relationship not seen in vivarium mice (or with MLN as there is no similar correlation in Figure 2B).

As the Reviewer noted, there are cell types driving differences in PBMCs as opposed to the MLN in Figure 1C. While we could determine that these cells are lymphocytes that are CD44+ and Ki67+ but are not T cells or B cells, we don't have sufficient markers in the flow cytometry panel to further define this population. While they could be some activated and proliferating innate cell type, the lack of clarity prevents us from discussing them in detail. We have added the following sentence to the text so as not to ignore them - Page 8, Line 176-179.

“The loading factor in the PCA analysis showed that Cluster C9, a TCRb⁺B220⁻Ki-67^{hi}CD44^{hi} population, might be driving the environment related variation on PC1 axis (Fig. SIC). While this population expands following rewilding regardless of the strain of mice, our limited markers prevent further characterization of this population.”

6. In Figure 1G, eosinophilia is seen in rewilded but not lab housed mice, yet in Figure 1H, the highest eosinophilia is seen in lab mice given *T muris*; hence the effect of infection is lost in the wild. While this validates the authors' contention of context dependent interactions, how can this be mechanistically explained?

We thank the reviewer for highlighting this observation. The increase in granulocytes and specifically, neutrophils and eosinophils in the rewilded mice has been investigated in detail and was recently published by our colleagues (Chen, Yeung et al. 2023). In summary, bone marrow granulopoiesis is critical and the expansion of the eosinophils and neutrophils was linked to fungal colonization. We have now included this new report in our discussions – Page 19, Line 540-541.

*However, we cannot provide a mechanistic explanation for why the effect of *T. muris* infection on eosinophilia is lost in rewilded mice since we don't have any data on this. We provide some context in discussion section on this. See Page 21, Line 583-588.*

*“Although, rewilded mice of all genotypes had higher eosinophilia at two weeks post-release (Fig 1G), likely due to increased granulopoiesis from exposure to wild fungi (52), there was no additional expansion following exposure to *T. muris* in the rewilded mice as opposed to laboratory mice (Fig 1H). While the mechanism for this observation is unclear and may be related to*

interactions with hematopoiesis, it further validates the concept of context dependent interactions on the immune phenotype and functional outcomes”

7. In Figure 2D, the MLN cell numbers are sharply increased in the rewilded mice, presumably due to environmentally-acquired intestinal infections or microbiota. In the analyses shown elsewhere in the Figure, are cell numbers or percentages used? The results may be very different in each case.

Reviewer 2 also raised the concern of using cell numbers compared to cell percentages in our analyses. In the rest of Figure 2, we have used cell percentages as a measure of cellular composition. This has now been clarified and we have included the associated cell numbers for our analysis of the corresponding proportions of CD44+ B cell that was previously reported in Figure 2.

When we use cell numbers, as noted by the reviewer, analysis of the different memory and naïve population (SFig4) as well as the cytokine data (SFig6) shows that the genotype by environment interactions has the biggest effect on determining variation in the different population of T cells with a residual fixed effect of genotype on the proportions of cell and of environment on the number of cells. These results are now described in Page 10, Line 249-261 and discussed in Page 17, Line 463-474.

Results

*“In addition, when we examined the proportions of the different naïve and memory T cell populations as we had previously examined in rewilded C57BL/6 mice (27, 31), we found that the central memory CD8 and CD4 T cells expand following rewilding in the PWK/PhJ and C57BL/6 strain of mice (Fig. S4A, B, C, D) as previous noted. An MDMR analysis of the different CD4 and CD8 T cell pools (Data File S6) reveals again that a Gen*Env interaction contributes to variation in the different T cell pools with a residual fixed main Genotype effect when proportions of cells were used for the analysis (Fig S4E). However, when analyzing absolute cell numbers, we found that Gen*Env similarly contributed to the variation in the T cell population. In this analysis, a residual main effect of Environment was the predominant factor explaining the remaining variance, in contrast to the Genotype effect that was prominent when*

we assessed cellular proportions (Fig. S4F). This differential main fixed effect of Genotype and Environment to cell proportion versus cell numbers might be due to the huge effect of environmentally-acquired intestinal microbiots, metaorganisms or food antigens on the MLN numbers(40, 41).”

Discussion, Page 17, Line 463-474

Despite the complexities, we do also identify traits for which main (i.e., non-interaction) effects are dominant. For instance, heterogeneity in proportions of cell with cytokine responses show a stronger influence of genetics, consistent with human studies (48), while heterogeneity in absolute cell numbers shows a stronger influence of the environment, consistent with results from previous microbial exposure mice studies(49). Furthermore, the Human Functional Genomics Project produced reported results, that variation in proportion of T cell phenotypes is relatively more influenced by genetics, while B cell phenotypes are relatively more influenced by non-heritable environmental factors (1). However, variation in absolute numbers of T cell phenotypes is still relatively more influenced by the environment, suggesting that an individual’s microbial history and experience plays a large role in determining absolute cell numbers present in different tissue sites through effects of microbial exposure on immune cell differentiation, activation, proliferation and maturation(31, 50-52)

8. Why in Figure 4B does each individual parameter have a significant effect size, but not when in combination?

When individual parameters have a significant effect size, but not their interactions, this indicates that the effects of the parameters are independent of each other. In this case,

Genetics and Environment interact to influence cellular composition in the scRNA-seq data, but the effects of infection on cellular composition does not significantly interact with Genetics and the Environment. Indeed, the results for Figure 4B is not the same as the results shown in Figure 2A, whereby G*E*I interactions were detected, but cellular composition was determined based on Flow Cytometry. The much greater number of cells analyzed by flow cytometry may provide more power to detect interactions that cannot be detected by ScRNA-seq analysis. Alternatively, an unbiased analysis by RNA-seq provides a different picture of immune composition from defined usage of cell surface markers. We have now added additional context to the text below:

*“In addition to the interactive effects of genotype and environment, the compositional analysis based on scRNAseq also identified independent effect of Genetics, Environment, and Infection with Trichuris muris (Figure 4B), which is consistent with the MLN spectral cytometry analysis (Figure 2A). Hence, these factors can have independent effects on immune composition that are not dependent on other factors. Furthermore, in contrast with spectral cytometry, three-way interactions (G*E*I) and other two-way interactions - G*I and E*I, were not significant when immune composition analysis was done by scRNAseq analysis (Figure 4B) (Figure 2A). This difference may be driven by the determination of immune composition by protein markers compared to unbiased scRNAseq, or by the total number of cells being analyzed. Nonetheless, the consistent conclusion of a significant Gen*Env interaction in both analyses suggests that this interaction is particularly critical in determining immune variation in the MLN.”*

9. Figure 4 G-I present proportions of all cells expressing any cytokine; this would seem very broad-brush and difficult to interpret. Can a more granular analysis be presented?

This an excellent suggestion by the Reviewer. For Figure 4G – I, we have now delved deeper into the analysis to (1) examine expression of IFN γ as a feature plot between the different strains of mice, environment and infection status in SFig7 (2) examine expression of some key cytokines in different cell types which the expression varies in the different strains of mice examined. These results are shown in SFig7 and described in Page 14-25, Lines 380-392 to further emphasize our conclusion that genotype contribute to variation in proportion of cell with cytokine activity as examined by scRNAseq.

S Fig 6

*“An unbiased scRNAseq approach therefore supports the conclusion that Genotype has the biggest effect on cytokine response heterogeneity based on proportion of cytokine expressing cells, whereas cellular composition and numbers are more driven by interactions between Genotype and the Environment. The effect of genotype on cytokine response in the MLN can be observed in feature plots when expression of IFN γ was examined as a feature plot (S Fig 7A). Here, we noted that genotype had an effect on relative expression of IFN γ with the greatest expression of IFN γ transcripts in the C57BL/6 strain of mice, though there was also increased expression of IFN γ transcripts following rewilding and exposure to *T. muris* (S Fig. 7A). Examination of other cytokines and chemokines in various cell types such as the CD8 effector cells, dendritic cells and monocytes/macrophage populations also shows a Genotype effect in differential expression of transcripts of these inflammatory mediators between the different strains of mice (S Fig. 7B, C and D). Together, these results suggest that the Genotype of the mice contribute to variation in functional response and cytokine activity”.*

10. In Figure 4 the Legend for (C) and (D) is erroneous and (E) is missing.

We apologize for this error, which has now been fixed.

11. Figure 5 D indicates factors that correlate positively or negatively with worm burden;

these could be strengthened by showing direct correlations with statistical support. What does “Lab Intercept” in panel C mean?

We now show a direct correlation between some of the loading factors in PC2 and Worm burden. For example, we had previously indicated that a component of the PC2 loading factor such as a dearth of T cells with interferon signature (T.IFN) may be a driver of the relationship between high PC2 scores and decreased worm burden. We have now done further exploration of our scRNAseq data (SFig 6A) as well as additional experiments in lab controls mice (Fig 5D) to show statistical support that an increased worm burden and worm prevalence in the C57BL/6 strain of mice might have direct correlation with the IFN γ levels in those strains of mice. These results are now described in Page 16, Line 425-433 and below.

For “Lab (Intercept)”, this notation simply reflects the structure of the Generalised Linear Model that we used to analyse the data. Here, the intercept of the model (i.e., the baseline for comparison with other groups) estimates the worm burden of C57BL/6 mice housed in the Lab; thus the full label at that point on the graph is C57BL/6-Lab (Intercept). All other groups are then compared against that baseline (e.g., 129S1 or PWK-PhJ strains vs C57BL/6; Rewilded vs Lab). We have now clarified this meaning in the figure legend (Page 32, Lines 684-690) and main text at Page 15-16, Lines 417-419.

*“Quantification of goblet cell count as a measure of effector Type 2 response (45-47) showed no significant difference between lab or rewilded environment in different strains of mice prior to exposure to *T. muris* (SFig. 8B). Furthermore, flow cytometric analysis of cytokine production in the three different strains of mice at day 14 post challenge with *T. muris* eggs from MLN cells following in vitro stimulation with cell activation cocktail (phorbol 12-myristate-13-acetate, ionomycin and protein transport inhibitor (Brefeldin A)) showed that increased levels of IFN γ , a Type 1 cytokine in the CD4 $^+$ T cells rather than differences in production of Type 2 cytokines, IL-4 and IL-13 (Fig. 5D) might explain variation in worm burden and prevalence, especially in the C57BL/6 strain of mice”.*

REFERENCE

Lin, J.-D., J. C. Devlin, F. Yeung, C. McCauley, J. M. Leung, Y.-H. Chen, A. Cronkite, C. Hansen, C. Drake-Dunn, K. V. Ruggles, K. Cadwell, A. L. Graham and P. n. Loke (2020). "Rewilding Nod2 and Atg16l1 Mutant Mice Uncovers Genetic and Environmental Contributions to Microbial Responses and Immune Cell Composition." Cell Host & Microbe **27**(5): 830-840.e834.

Yeung, F., Y.-H. Chen, J.-D. Lin, J. M. Leung, C. McCauley, J. C. Devlin, C. Hansen, A. Cronkite, Z. Stephens, C. Drake-Dunn, Y. Fulmer, B. Shopsis, K. V. Ruggles, J. L. Round, P. n. Loke, A. L. Graham and K. Cadwell (2020). "Altered Immunity of Laboratory Mice in the Natural Environment Is Associated with Fungal Colonization." Cell Host & Microbe **27**(5): 809-822.e806.

Leung, J. M., S. A. Budischak, H. Chung The, C. Hansen, R. Bowcutt, R. Neill, M. Shellman, P. n. Loke and A. L. Graham (2018). "Rapid environmental effects on gut nematode susceptibility in rewilded mice." PLOS Biology **16**(3): e2004108.

Artis, D., C. S. Potten, K. J. Else, F. D. Finkelman and R. K. Grencis (1999). "Trichuris muris: Host Intestinal Epithelial Cell Hyperproliferation during Chronic Infection Is Regulated by Interferon- γ ." Experimental Parasitology **92**(2): 144-153.

Cliffe, L. J., N. E. Humphreys, T. E. Lane, C. S. Potten, C. Booth and R. K. Grencis (2005). "Accelerated Intestinal Epithelial Cell Turnover: A New Mechanism of Parasite Expulsion." Science **308**(5727): 1463-1465.

Else, K. J., F. D. Finkelman, C. R. Maliszewski and R. K. Grencis (1994). "Cytokine-mediated regulation of chronic intestinal helminth infection." Journal of Experimental Medicine **179**(1): 347-351.

Hurst, R. J. M. and K. J. Else (2013). "Trichuris muris research revisited: a journey through time." Parasitology **140**(11): 1325-1339.

Chen, Y.-H., F. Yeung, K. A. Lacey, K. Zalana, J.-D. Lin, G. C. W. Bee, C. McCauley, R. S. Barre, S.-H. Liang, C. B. Hansen, A. E. Downie, K. Tio, J. N. Weiser, V. J. Torres, R. J. Bennett, P. n. Loke, A. L. Graham and K. Cadwell (2023). "Rewilding of laboratory mice enhances granulopoiesis and immunity through intestinal fungal colonization." Science Immunology **8**(84): eadd6910.

Decision Letter, first revision:

11th Mar 2024

Dear Dr. Oyesola,

Thank you for submitting your revised manuscript "Genetic and Environmental interactions contribute to immune variation in rewilded mice" (NI-A35790B). It has now been seen by the original referees and their comments are below. The reviewers find that the paper has improved in revision, and therefore we'll be happy in principle to publish it in Nature Immunology, pending minor revisions to satisfy the referees' final requests and to comply with our editorial and formatting guidelines.

We will now perform detailed checks on your paper and will send you a checklist detailing our editorial and formatting requirements in about a week. Please do not upload the final materials and make any revisions until you receive this additional information from us.

If you had not uploaded a Word file for the current version of the manuscript, we will need one before beginning the editing process; please email that to immunology@us.nature.com at your earliest convenience.

Thank you again for your interest in Nature Immunology. Please do not hesitate to contact me if you have any questions.

Kind regards,

Laurie

Laurie A. Dempsey, Ph.D.
Senior Editor
Nature Immunology
l.dempsey@us.nature.com
ORCID: 0000-0002-3304-796X

Reviewer #2 (Remarks to the Author):

I think that the amendments to and additional data have improved the manuscript and the authors have answered the points I raised. There are a couple of small points that the authors should consider below.

Line 85 – should "larva" be "eggs"?

Lines 187- 189 and Fig S2 regarding PBMC. Fig S2 is unclear. In S2 is there a descriptor missing? Should these plots also indicate if mice have been infected or not? The authors state in the text that a stronger Th1 response is seen in the infected C57BL/6 mice when rewilded compared to lab house. The % of Tbet + CD4+ are clearly different in C57BL/6 compared to other strains. Are the two plots representative of a non-infected and an infected animal respectively? If so it would suggest that

simply re-wilding C57BL/6 without *T. muris* infection appears to elevate the number of Tbet + CD4+ cells. So does the data more clearly say that rewilding C57BL/6 mice per se increases Th1 response much more dramatically in C57BL/6 than the other strains? I apologise if I have got this wrong.

Lines 427-436 . Fig 5D. Can the authors state in the legend that the data from day 14 was from mice infected and kept under laboratory conditions.

Also, to note, there is a clear trend in all strains for a type 2 signature (IL-13+, IL-4+) in infected mice. The % changes are small and statistically not significant but nevertheless these changes appear to be there. And of course, the low %/numbers of cells do not mean they are not potent in activity or less potent in activity than the higher levels/% IFN- γ + cells. I think this point helps explain the overall observations.

It is clear from the original worm burden data that for the majority of animals in all of the strains infected with 200 eggs under Lab or rewilded conditions have few/no worms on day 21 post infection, i.e have expelled/lost their worms. This is important for the readers to appreciate and the authors to stress.

Based upon a wealth of published data the most likely explanation is that the majority of worms had been expelled by a Th2/Type 2 cytokine protective response by this time point. Unless the authors think that the different strains of mice have different parasite establishment levels or that worm loss is not mediated by Type 2 responses. Thus, taken together, the hypothesis would be that Type 2 immunity operates to clear *T. muris* both in lab and rewilded conditions in all strains of mice. The effectiveness of this response varies between mouse strains and in rewilded conditions the increased propensity of the different strains to make IFN- γ , for whatever reason, influences the effectiveness of protective immunity generated. This supports the notion that IFN- γ may be influencing the balance of the Type2/Type 1 immunity generated, as the authors discuss. I think it is important to recognise that immunity is operating effectively in both conditions, lab and rewilded. IFN- γ modulates this response, and the stronger (and probably quicker) this is the greater effect on modulating the response that clears the parasites effectively.

Reviewer #3 (Remarks to the Author):

The authors have responded comprehensively to all the points raised by the Reviewers. I only have two general comments and a few specific corrections to raise.

1. I still find the abstract too general, and it will be difficult for those eg searching on PubMed, to gain information from it as written. Half of the abstract is general introduction or overview. The following sentence conveys little : "Importantly, variation in worm burden is associated with measures of immune variation, as well as genetics and environment". Even with the severe word limit imposed by the journal it should be possible to include more specific findings in the abstract.

2. Stepping back, would it be correct to generalise that genetics determine the quality of the immune response, and environment the quantity?

Corrections

In the discussion, lines 468-470 is duplicated by lines 509-511

Line 579, increased missing final d.

Author Rebuttal, first revision:

See inserted PDF

We thank all the reviewers for their feedback. We have now included a response here below.

Reviewer #2:

Remarks to the Author:

I think that the amendments to and additional data have improved the manuscript and the authors have answered the points I raised. There are a couple of small points that the authors should consider below.

We thank the Reviewer for their feedback.

Line 85 – should “larva” be “eggs”?

“larva” in Line 85 has been changed to “embryonated eggs.”

Lines 187- 189 and Fig S2 regarding PBMC. Fig S2 is unclear. In S2 is there a descriptor missing? Should these plots also indicate if mice have been infected or not?

We thank the reviewer for this observation, we have now included the descriptor – Uninfected and *T. muris* in Fig S2

The authors state in the text that a stronger Th1 response is seen in the infected C57BL/6 mice when rewilded compared to lab house. The % of Tbet + CD4+ are clearly different in C57BL/6 compared to other strains. Are the two plots representative of a non-infected and an infected animal respectively? If so, it would suggest that simply re-wilding C57BL/6 without *T. muris* infection appears to elevate the number of Tbet + CD4+ cells. So does the data more clearly say that rewilding C57BL/6 mice per se increases Th1 response much more dramatically in C57BL/6 than the other strains? I apologise if I have got this wrong.

We thank the reviewer for the clarification sought for this text. We agree with the reviewer that there is a trend to suggest that in the C57BL/6 mice, rewilding alone or exposure to *T. muris* eggs in the lab environment led to an increase in the proportion of Tbet+ CD4 T cells compared to other strains. However, this difference is not statistically significant. We have now replaced Figure 1F with a figure that shows the % of Tbet+ CD4 T cells of CD4 T cells in individual mice that was previously displayed in Fig. S2, which illustrates this point.

Lines 427-436 . Fig 5D. Can the authors state in the legend that the data from day 14 was from mice infected and kept under laboratory conditions.

We thank the reviewer for this observation. We have now included this in the information both in the text and in the legend.

Also, to note, there is a clear trend in all strains for a type 2 signature (IL-13+, IL-4+) in infected mice. The % changes are small and statistically not significant but nevertheless these changes appear to be there. And of course, the low %/numbers of cells do not mean they are not potent in activity or less potent in activity than the higher levels/% IFN- γ + cells. I think this point helps explain the overall observations.

It is clear from the original worm burden data that for most animals in all of the strains infected with 200 eggs under Lab or rewilded conditions have few/no worms on day 21 post infection, i.e

have expelled/lost their worms. This is important for the readers to appreciate and the authors to stress.

Based upon a wealth of published data the most likely explanation is that the majority of worms had been expelled by a Th2/Type 2 cytokine protective response by this time point. Unless the authors think that the different strains of mice have different parasite establishment levels or that worm loss is not mediated by Type 2 responses. Thus, taken together, the hypothesis would be that Type 2 immunity operates to clear *T. muris* both in lab and rewilded conditions in all strains of mice. The effectiveness of this response varies between mouse strains and in rewilded conditions the increased propensity of the different strains to make IFN- γ , for whatever reason, influences the effectiveness of protective immunity generated. This supports the notion that IFN- γ may be influencing the balance of the Type2/Type 1 immunity generated, as the authors discuss. I think it is important to recognize that immunity is operating effectively in both conditions, lab and rewilded. IFN- γ modulates this response, and the stronger (and probably quicker) this is the greater effect on modulating the response that clears the parasites effectively.

We thank the reviewer for these points, which have now been addressed in the manuscript within the limitations of the word limit for the discussion and the manuscript.

Reviewer #3:

Remarks to the Author:

The authors have responded comprehensively to all the points raised by the Reviewers. I only have two general comments and a few specific corrections to raise.

1. I still find the abstract too general, and it will be difficult for those eg searching on PubMed, to gain information from it as written. Half of the abstract is general introduction or overview. The following sentence conveys little : "Importantly, variation in worm burden is associated with measures of immune variation, as well as genetics and environment". Even with the severe word limit imposed by the journal it should be possible to include more specific findings in the abstract.

We appreciate the reviewers feedback on the abstract. The abstract has been edited to be more specific.

2. Stepping back, would it be correct to generalise that genetics determine the quality of the immune response, and environment the quantity?

We believe that the big picture conclusion is that genetics determines how the magnitude of the immune response is altered by the environment. Hence, both the quality and quantity of an individuals' immune response is determined by interactions between genetic and environmental factors.

Corrections

In the discussion, lines 468-470 is duplicated by lines 509-511

Line 579, increased missing final d.

We thank the reviewer for identifying these errors. We have edited and corrected them in the manuscript.

Final Decision Letter:

Dear Bola,

I am delighted to accept your manuscript entitled "Genetic and Environmental interactions contribute to immune variation in rewilded mice" for publication in an upcoming issue of Nature Immunology.

Over the next few weeks, your paper will be copyedited to ensure that it conforms to Nature Immunology style. Once your paper is typeset, you will receive an email with a link to choose the appropriate publishing options for your paper and our Author Services team will be in touch regarding any additional information that may be required.

Please note that *Nature Immunology* is a Transformative Journal (TJ). Authors may publish their research with us through the traditional subscription access route or make their paper immediately open access through payment of an article-processing charge (APC). Authors will not be required to make a final decision about access to their article until it has been accepted. Find out more about Transformative Journals.

Your paper will be published online soon after we receive your corrections and will appear in print in the next available issue.

Also, if you have any spectacular or outstanding figures or graphics associated with your manuscript - though not necessarily included with your submission - we'd be delighted to consider them as candidates for our cover. Simply send an electronic version (accompanied by a hard copy) to us with a possible cover caption enclosed.

If you have not already done so, we strongly recommend that you upload the step-by-step protocols used in this manuscript to the Protocol Exchange. Protocol Exchange is an open online resource that allows researchers to share their detailed experimental know-how. All uploaded protocols are made freely available, assigned DOIs for ease of citation and fully searchable through nature.com. Protocols can be linked to any publications in which they are used and will be linked to from your article. You can also establish a dedicated page to collect all your lab Protocols. By uploading your Protocols to Protocol Exchange, you are enabling researchers to more readily reproduce or adapt the methodology you use, as well as increasing the visibility of your protocols and papers. Upload your Protocols at www.nature.com/protocolexchange/. Further information can be found at www.nature.com/protocolexchange/about .

Please note that we encourage the authors to self-archive their manuscript (the accepted version before copy editing) in their institutional repository, and in their funders' archives, six months after publication. Nature Portfolio recognizes the efforts of funding bodies to increase access of the research they fund, and strongly encourages authors to participate in such efforts. For information about our editorial policy, including license agreement and author copyright, please visit www.nature.com/ni/about/ed_policies/index.html

Sincerely,

Laurie

Laurie A. Dempsey, Ph.D.
Senior Editor
Nature Immunology
l.dempsey@us.nature.com
ORCID: 0000-0002-3304-796X